# Intraspecific predator interference promotes biodiversity in ecosystems

**Ju Kang**[1†]**, Shijie Zhang**[2,3†]**, Yiyuan Niu**[1]**, Fan Zhong**[1]**, Xin Wang**[1]*****

[1]School of Physics, Sun Yat-sen University, Guangzhou, China; [2]School of Mathematics, Sun Yat-sen University, Guangzhou, China; [3]Department of Mechanical Engineering, Massachusetts Institute of Technology, Cambridge, United States

---

### eLife assessment

This manuscript is an **important** contribution, assessing the role of intraspecific consumer interference in maintaining diversity using a mathematical model. Consistent with long-standing ecological theory, the authors **convincingly** show that predator interference allows for the coexistence of multiple species on a single resource, beyond the competitive exclusion principle. Notably, the model matches observed rank-abundance curves in several natural ecosystems.

---

**\*For correspondence:**
wangxin36@mail.sysu.edu.cn

[†]These authors contributed equally to this work

**Competing interest:** The authors declare that no competing interests exist.

**Abstract** Explaining biodiversity is a fundamental issue in ecology. A long-standing puzzle lies in the paradox of the plankton: many species of plankton feeding on a limited variety of resources coexist, apparently flouting the competitive exclusion principle (CEP), which holds that the number of predator (consumer) species cannot exceed that of the resources at a steady state. Here, we present a mechanistic model and demonstrate that intraspecific interference among the consumers enables a plethora of consumer species to coexist at constant population densities with only one or a handful of resource species. This facilitated biodiversity is resistant to stochasticity, either with the stochastic simulation algorithm or individual-based modeling. Our model naturally explains the classical experiments that invalidate the CEP, quantitatively illustrates the universal S-shaped pattern of the rank-abundance curves across a wide range of ecological communities, and can be broadly used to resolve the mystery of biodiversity in many natural ecosystems.

## Introduction

The most prominent feature of life on Earth is its remarkable species diversity: countless macro- and micro-species fill every corner on land and in the water (**Pennisi, 2005**; **Hoorn et al., 2010**; **de Vargas et al., 2015**; **Daniel, 2005**). In tropical forests, thousands of plant and vertebrate species coexist (**Hoorn et al., 2010**). Within a gram of soil, the number of microbial species is estimated to be 2000–18,000 (**Daniel, 2005**). In the photic zone of the world ocean, there are roughly 150,000 eukaryotic plankton species (**de Vargas et al., 2015**). Explaining this astonishing biodiversity is a major focus in ecology (**Pennisi, 2005**). A great challenge stems from the well-known competitive exclusion principle (CEP): two species competing for a single type of resources cannot coexist at constant population densities (**Gause, 1934**; **Hardin, 1960**), or generically, in the framework of consumer-resource models, the number of consumer species cannot exceed that of resources at a steady state (**MacArthur and Levins, 1964**; **Levin, 1970**; **McGehee, 1977**). On the contrary, in the paradox of plankton, a limited variety of resources supports hundreds or more coexisting species of phytoplankton (**Hutchinson, 1961**). Then, how can plankton and many other organisms somehow liberate the constraint of CEP?

Ever since MacArthur and Levin proposed the classical mathematical proof for CEP in the 1960s (**MacArthur and Levins, 1964**), various mechanisms have been put forward to overcome the limits

**eLife digest** The surface waters of the ocean are teeming with microscopic creatures known as plankton, which get carried across vast distances by the currents. In a single ecosystem, thousands of plankton species may coexist, all competing for very few types of food sources. According to the principle of competitive exclusion, this should not be the case. Indeed, this theory states that the population levels of two species competing for the same resource cannot remain steady over time – or more generally, that the number of consumer species in an ecosystem cannot be higher than the number of resource types on which they rely. And yet, the Earth abounds with examples where a limited variety of resources supports a large number of competing yet coexisting consumer species. This is known as the paradox of the plankton.

Many models have been proposed to explain how the limitations set by the competitive exclusion principle can be overcome, yet it is still unknown how to resolve the paradox of the plankton in a steady environment. In response, Kang et al. set out to test whether a phenomenon known as predator interference, which emerges when two or more individuals of the same species compete for the same resources, could help address the paradox of the plankton.

To test this idea, Kang et al. developed a mathematical model of predator interference for multiple species of plankton feeding on a limited variety of food sources. The model put predators of the same species into encountering pairs to simulate predator interference. In this scenario, a wide range of predator species were able to live alongside each other with the numbers of each type of predator remaining steady over time.

These results can be understood as follows: as a species becomes more successful at extracting resources from its environment, its population grows – and with it, the number of individuals engaged in intraspecific interference. Locked in interference, these species become less effective at getting food, creating a negative feedback loop that slows down the expansion of the species, allowing others to occupy the same niche.

These findings may benefit ecologists and conservationists by offering insights into how to maintain biodiversity in ecosystems and protect endangered species. Further work is needed to test how well the model applies to other ecosystems.

set by CEP (*Chesson, 2000*). Some suggest that the system never approaches a steady state where the CEP applies, due to temporal variations (*Hutchinson, 1961*; *Levins, 1979*), spatial heterogeneity (*Levin, 1974*), or species' self-organized dynamics (*Koch, 1974*; *Huisman and Weissing, 1999*). Others consider factors such as toxins (*Czárán et al., 2002*), cross-feeding (*Goyal and Maslov, 2018*; *Goldford et al., 2018*; *Niehaus et al., 2019*), spatial circulation (*Villa Martín et al., 2020*; *Gupta et al., 2021*), 'kill the winner' (*Thingstad, 2000*), pack hunting (*Wang and Liu, 2020*), collective behavior (*Dalziel et al., 2021*), metabolic trade-offs (*Posfai et al., 2017*; *Weiner et al., 2019*), co-evolution (*Xue and Goldenfeld, 2017*), and other complex interactions among the species (*Beddington, 1975*; *DeAngelis et al., 1975*; *Arditi and Ginzburg, 1989*; *Kelsic et al., 2015*; *Grilli et al., 2017*; *Ratzke et al., 2020*). However, questions remain as to what determines species diversity in nature, especially for quasi-well-mixed systems such as that of plankton (*Pennisi, 2005*; *Sunagawa et al., 2020*).

Among the proposed mechanisms, predator interference, specifically the pairwise encounters among consumer individuals, emerges as a potential solution to this issue. Predator interference is commonly described by the classical Beddington-DeAngelis (B-D) phenomenological model (*Beddington, 1975*; *DeAngelis et al., 1975*). Through the application of the B-D model, several studies (*Cantrell et al., 2004*; *Hsu et al., 2013*) have shown that intraspecific predator interference can break CEP and facilitate species coexistence. However, from a mechanistic perspective, the functional response of the B-D model can be formally derived from a scenario solely involving chasing pairs, representing the consumption process between consumers and resources, without accounting for pairwise encounters among consumer individuals (*Wang and Liu, 2020*; *Huisman and De Boer, 1997*). Disturbingly, it has been established that the scenario involving only chasing pairs is subject to the constraint of CEP (*Wang and Liu, 2020*), raising doubt regarding the validity of applying the B-D model to overcome the CEP.

In this work, building upon MacArthur's consumer-resource model framework (**Arthur, 1969**; **MacArthur, 1970**; **Chesson, 1990**), and drawing on concepts from chemical reaction kinetics (**Ruxton et al., 1992**; **Huisman and De Boer, 1997**; **Wang and Liu, 2020**), we present a mechanistic model of predator interference that extends the B-D phenomenological model (**Beddington, 1975**; **DeAngelis et al., 1975**) by providing a detailed consideration of pairwise encounters. The intraspecific interference among consumer individuals effectively constitutes a negative feedback loop, enabling a wide range of consumer species to coexist with only one or a few types of resources. The coexistence state is resistant to stochasticity and can hence be realized in practice. Our model is broadly applicable and can be used to explain biodiversity in many ecosystems. In particular, it naturally explains species coexistence in classical experiments that invalidate CEP (**Ayala, 1969**; **Park, 1954**) and quantitatively illustrates the S-shaped pattern of the rank-abundance curves in an extensive spectrum of ecological communities, ranging from the communities of ocean plankton worldwide (**Fuhrman et al., 2008**; **Ser-Giacomi et al., 2018**), tropical river fishes from Argentina (**Cody and Smallwood, 1996**), forest bats of Trinidad (**Clarke et al., 2005**), rainforest trees (**Hubbell, 2001**), birds (**Terborgh et al., 1990**; **Martínez et al., 2023**), butterflies (**De Vries, 1997**) in Amazonia, to those of desert bees (**Hubbell, 2001**) in Utah and lizards from South Australia (**Cody and Smallwood, 1996**).

## Results
### A generic model of pairwise encounters

Here we present a mechanistic model of pairwise encounters (see **Figure 1A**), where $S_C$ consumer species $\{C_1, \cdots, C_{S_C}\}$ compete for $S_R$ resource species $\{R_1, \cdots, R_{S_R}\}$. The consumers are biotic, while the resources can be either biotic or abiotic. For simplicity, we assume that all species are motile and move at certain speeds, namely, $v_{C_i}$ for consumer species $C_i$ and $v_{R_l}$ for resource species $R_l$. For abiotic resources, they cannot propel themselves but may passively drift due to environmental factors. Each consumer is free to feed on one or multiple types of resources, while consumers do not directly interact with one another except through pairwise encounters.

Then, we explicitly consider the population structure of consumers and resources: some wander around freely, undergoing Brownian motions; others encounter one another, forming ephemeral entangled pairs. Specifically, when a consumer individual $C_i$ and a resource $R_l$ come close within a distance of $r_{il}^{(C)}$ (see **Figure 1A**), the consumer can chase the resource and form a chasing pair: $C_i^{(P)} \bigvee R_l^{(P)}$ (see **Figure 1B**), where the superscript '(P)' represents 'pair'. The resource can either escape at rate $d_{il}$ or be caught and consumed by the consumer with rate $k_{il}$. Meanwhile, when a $C_i$ individual encounters another consumer $C_j$ within a distance of $r_{ij}^{(I)}$ (see **Figure 1A**), they can stare at, fight against, or play with each other, thus forming an interference pair: $C_i^{(P)} \bigvee C_j^{(P)}$ (see **Figure 1B**). This paired state is evanescent, with consumers separating at rate $d_{ij}'$. For simplicity, we assume that all $r_{il}^{(C)}$ and $r_{ij}^{(I)}$ are identical, respectively, that is $\forall i,j,l$, $r_{il}^{(C)} = r^{(C)}$ and $r_{ij}^{(I)} = r^{(I)}$.

In a well-mixed system of size $L^2$, the encounter rates among individuals, $a_{il}$ and $a_{ij}'$ (see **Figure 1B**), can be derived using the mean-field approximation: $a_{il} = 2r^{(C)}L^{-2}\sqrt{v_{C_i}^2 + v_{R_l}^2}$ and $a_{ij}' = 2r^{(I)}L^{-2}\sqrt{v_{C_i}^2 + v_{C_j}^2}$ (see Materials and methods, and **Appendix 1—figure 1**). Then, we proceed to analyze scenarios involving different types of pairwise encounters (see **Figure 1B**). For the scenario involving only chasing pairs, the population dynamics can be described as follows:

$$
\begin{cases}
\dot{x}_{il} = a_{il}C_i^{(F)}R_l^{(F)} - (d_{il} + k_{il})x_{il}, \\
\dot{C}_i = \sum_{l=1}^{S_R} w_{il}k_{il}x_{il} - D_iC_i, i = 1, \cdots, S_C, \\
\dot{R}_l = g_l(\{R_l\}, \{x_i\}, \{C_i\}), l = 1, \cdots, S_R,
\end{cases}
\tag{1}
$$

where $x_{il} \equiv C_i^{(P)} \bigvee R_l^{(P)}$, $g_l$ is an unspecified function, the superscript '(F)' represents the freely wandering population, $D_i$ denotes the mortality rate of $C_i$, and $w_{il}$ is the mass conversion ratio (**Wang and Liu, 2020**) from resource $R_l$ to consumer $C_i$. With the integration of intraspecific predator interference, we combine **Equation 1** and the following equation:

$$
\dot{y}_i = a_i'[C_i^{(F)}]^2 - d_i'y_i,
\tag{2}
$$

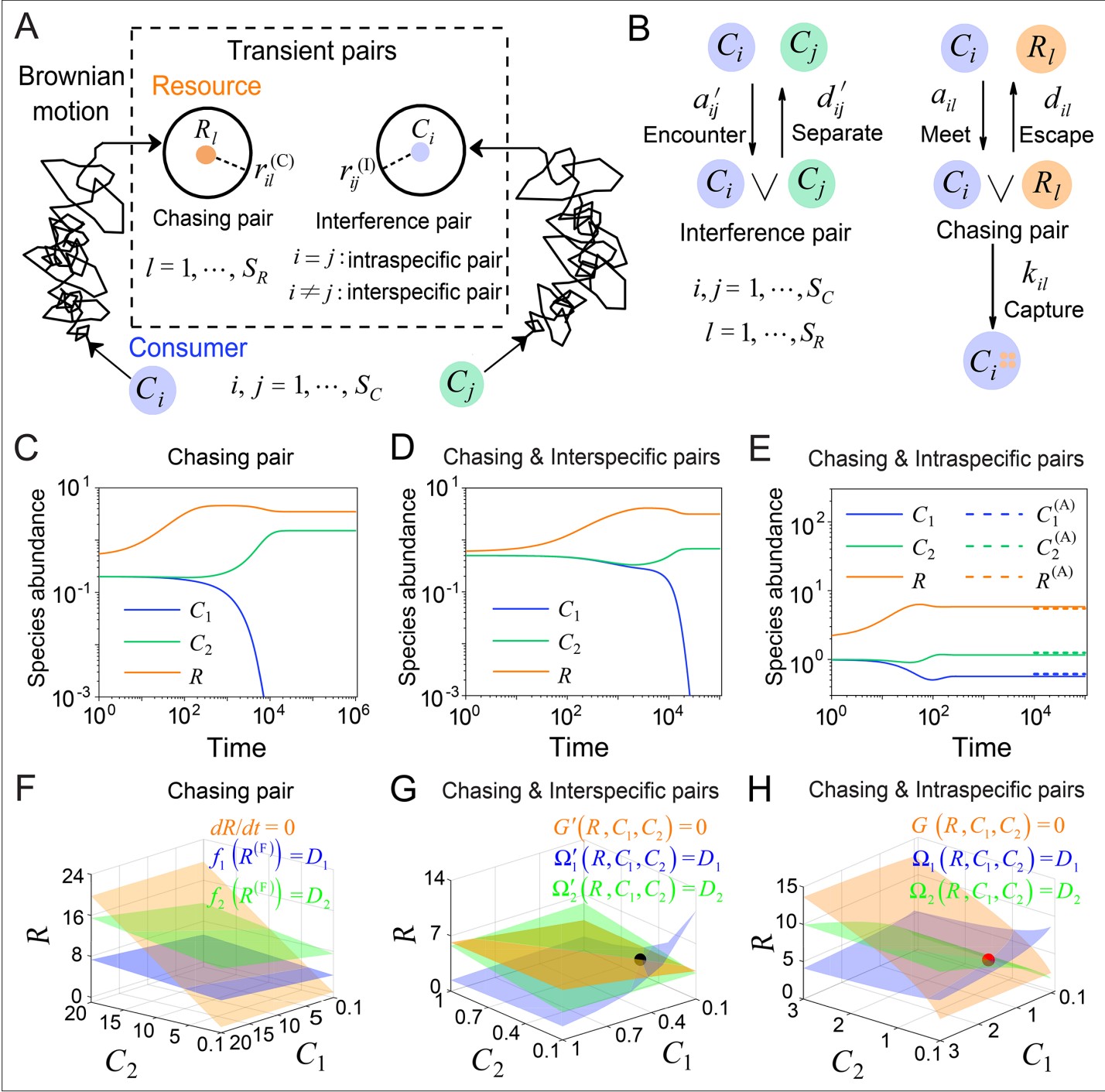

**Figure 1.** A model of pairwise encounters may naturally break CEP. (**A**) A generic model of pairwise encounters involving $S_C$ consumer species and $S_R$ resource species. (**B**) The well-mixed model of (**A**). (**C–E**) Time courses of two consumer species competing for one resource species. (**F–H**) Positive solutions to the steady-state equations (see ***Equations S38 and S65***): $\dot{R} = 0$ (orange surface), $\dot{C}_1 = 0$ (blue surface), $\dot{C}_2 = 0$ (green surface), that is the zero-growth isoclines. The black/red dot represents the unstable/stable fixed point, while the dotted lines in (**E**) are the analytical solutions of the steady-state abundances (marked with superscript '(A)'). See ***Appendix 1—tables 1 and 2*** for the definitions of symbols. See Appendix 9 for simulation details.

where $a_i' = a_{ii}'$, $d_i' = d_{ii}'$, and $y_i \equiv C_i^{(P)} \bigvee C_i^{(P)}$ represents the intraspecific interference pair (see **Figure 1B**). For the scenario involving chasing pairs and interspecific interference, we combine **Equation 1** with the following equation:

$$\dot{z}_{ij} = a_{ij}' C_i^{(F)} C_j^{(F)} - d_{ij}' z_{ij}, \; i \neq j, \tag{3}$$

where $z_{ij} \equiv C_i^{(P)} \bigvee C_j^{(P)}$ stands for the interspecific interference pair (see **Figure 1B**). In the scenario where chasing pairs and both intra- and interspecific interference are relevant, we combine **Equations 1–3**, and the populations of consumers and resources are given by $C_i = C_i^{(F)} + \sum_l x_{il} + 2y_i + \sum_{j \neq i} z_{ij}$ and $R_l = R_l^{(F)} + \sum_i x_{il}$, respectively.

Generically, the consumption and interference processes are much quicker compared to the birth and death processes. Thus, in the derivation of the functional response, $\mathcal{F}(R_l, C_i) \equiv k_{il} x_{il}/C_i$, the consumption and interference processes are supposed to be in fast equilibrium. In all scenarios involving different types of pairwise encounters, the functional response in the B-D model is a good approximation only for a special case with $d_{il} \approx 0$ and $R_l \gg \sum_{i=1}^{S_C} C_i$ (see **Appendix 1—figure 2** and Appendix 3 for details).

To facilitate further analysis, we assume that the population dynamics of the resources follows the same construction rule as that in MacArthur's consumer-resource model (**Arthur, 1969**; **MacArthur, 1970**; **Chesson, 1990**). Then,

$$g_l(\{R_l\}, \{x_i\}, \{C_i\}) = \begin{cases} \eta_l R_l (1 - R_l/\kappa_l) - \sum_{i=1}^{S_C} k_{il} x_{il} & \text{(for biotic resources);} \\ \zeta_l (1 - R_l/\kappa_l) - \sum_{i=1}^{S_C} k_{il} x_{il} & \text{(for abiotic resources).} \end{cases} \tag{4}$$

In the absence of consumers, biotic resources exhibit logistic growth. Here, $\eta_l$ and $\kappa_l$ represent the intrinsic growth rate and the carrying capacity of species $R_l$. For abiotic resources, $\zeta_l$ stands for the external resource supply rate of $R_l$, and $\kappa_l$ is the abundance of $R_l$ at a steady state without consumers. For simplicity, we focus our analysis on abiotic resources, although all results generally apply to biotic resources as well. By applying dimensional analysis, we render all parameters dimensionless (see Appendix 7). For convenience, we retain the same notations below, with all parameters considered dimensionless unless otherwise specified.

## Intraspecific predator interference facilitates species coexistence and breaks CEP

To clarify the specific mechanisms that can facilitate species coexistence, we systematically investigate scenarios involving different forms of pairwise encounters in a simple case with $S_C = 2$ and $S_R = 1$. To simplify the notations, we omit the subscript/superscript '$l$' since $S_R = 1$. For clarity, we assign each consumer species of unique competitiveness by setting that the mortality rate $D_i$ is the only parameter that varies with the consumer species.

First, we conduct the analysis within a deterministic framework using ordinary differential equations (ODEs). In the scenario involving only chasing pairs, consumer species cannot coexist at a steady state except for special parameter settings (sets of measure zero) (**Wang and Liu, 2020**). In practice, if all species coexist, the steady-state equations of the consumer species ($\dot{C}_i = 0$, i.e. the zero-growth isolines) yield $f_i(R^{(F)}) = D_i$ ($i = 1, 2$), where $f_i(R^{(F)})$ is defined as $f_i(R^{(F)}) \equiv R^{(F)}/(R^{(F)} + K_i)$ and $K_i \equiv (d_i + k_i)/a_i$. These equations form two parallel surfaces in the $(C_1, C_2, R)$ coordinates, making steady coexistence impossible (**Wang and Liu, 2020**; see **Figure 1C and F** and **Appendix 1—figure 3A–C**).

Meanwhile, in the scenario involving chasing pairs and interspecific interference, if all species coexist, the zero-growth isolines of the three species (see **Equation S65**) correspond to three non-parallel surfaces $\Omega_i'(R, C_1, C_2) = D_i$ ($i = 1, 2$), $G'(R, C_1, C_2) = 0$ (see **Figure 1G** and **Appendix 1—figure 3D**; refer to Appendix 5 for definitions of $\Omega_i'$ and $G'$), which can intersect at a common point (fixed point). However, this fixed point is unstable (see **Figure 1G** and **Appendix 1—figure 3D, F**), and thus one of the consumer species is doomed to extinction (see **Figure 1D**).

Next, we turn to the scenario involving chasing pairs and intraspecific interference. Likewise, steady coexistence requires (see *Equation S38*) that three non-parallel surfaces $\Omega_i(R, C_1, C_2) = D_i$ ($i = 1, 2$), $G(R, C_1, C_2) = 0$ cross at a common point (see *Figure 1H* and *Appendix 1—figure 3G*; refer to Appendix 4 for definitions of $\Omega_i$ and $G$). Indeed, this naturally happens, and encouragingly the fixed point can be stable. Therefore, two consumer species may stably coexist at a steady state with only one type of resources, which obviously breaks CEP (see *Figure 1E* and *Appendix 1—figure 4A*). In fact, the coexisting state is globally attractive (see *Appendix 1—figure 4A*), and there exists a non-zero volume of parameter space where the two consumer species stably coexist at constant population densities (see *Appendix 1—figure 4B, C*), demonstrating that the violation of CEP does not depend on special parameter settings. We further consider the scenario involving chasing pairs and both intra- and interspecific interference (see *Appendix 1—figure 5*). Much as expected, the species coexistence behavior is very similar to that without interspecific interference.

## Intraspecific interference promotes biodiversity in the presence of stochasticity

Stochasticity is ubiquitous in nature. However, it is prone to jeopardize species coexistence (*Xue and Goldenfeld, 2017*). Influential mechanisms such as 'kill the winner' fail when stochasticity is incorporated (*Xue and Goldenfeld, 2017*). Consistent with this, we observe that two notable cases of oscillating coexistence (*Koch, 1974*; *Huisman and Weissing, 1999*) turn into species extinction when stochasticity is introduced (see *Appendix 1—figure 6A, B*), where we simulate the models with stochastic simulation algorithm (SSA; *Gillespie, 2007*) and adopt the same parameters as those in the original references (*Koch, 1974*; *Huisman and Weissing, 1999*).

Then, we proceed to investigate the impact of stochasticity on our model using SSA (*Gillespie, 2007*). In the scenario involving chasing pairs and intraspecific interference, species may coexist indefinitely in the SSA simulations (see *Figure 2A* and *Appendix 1—figure 4D*). In fact, the parameter region for species coexistence in this scenario is rather similar between the SSA and ODEs studies (see *Appendix 1—figure 6C, D*). Similarly, in the scenario involving chasing pairs and both inter- and intraspecific interference, all species may coexist indefinitely in company with stochasticity (see *Appendix 1—figure 5D*).

To further mimic a real ecosystem, we resort to individual-based modeling (IBM; *Grimm, 2013*; *Vetsigian, 2017*), an essentially stochastic simulation method. In the simple case of $S_C = 2$ and $S_R = 1$, we simulate the time evolution of a 2-D square system in a size of $L^2$ with periodic boundary conditions (see Materials and methods for details). In the scenario involving chasing pairs and intraspecific interference, two consumer species coexist for long with only one type of resources in the IBM simulations (see *Figure 2B and C*). Together with the SSA simulation studies, it is obvious that intraspecific interference still robustly promotes species coexistence when stochasticity is considered.

## Comparison with experimental studies that reject CEP

In practice, two classical studies (*Ayala, 1969*; *Park, 1954*) reported that, in their respective laboratory systems, two species of insects coexisted for roughly years or more with only one type of resources. Evidently, these two experiments (*Ayala, 1969*; *Park, 1954*) are incompatible with CEP, while factors such as temporal variations, spatial heterogeneity, cross-feeding, etc. are clearly not involved in such systems. As intraspecific fighting is prevalent among insects (*Boomsma et al., 2005*; *Dankert et al., 2009*; *Chen et al., 2002*), we apply the model involving chasing pairs and intraspecific interference to simulate the two systems. Overall, our SSA results show good consistency with those of the experiments (see *Figure 2D and E* and *Appendix 1—figure 7*). The fluctuations in experimental time series can be mainly accounted by stochasticity.

## A handful of resource species can support a wide range of consumer species regardless of stochasticity

To resolve the puzzle stated in the paradox of the plankton, we analyze the generic case where $S_C$ consumer species compete for $S_R$ resource species (with $S_C > S_R$) within the scenario involving chasing pairs and intraspecific interference. The population dynamics is described by equations combining *Equations 1, 2 and 4*. As with the cases above, each consumer species is assigned a unique competitiveness through a distinctive $D_i$.

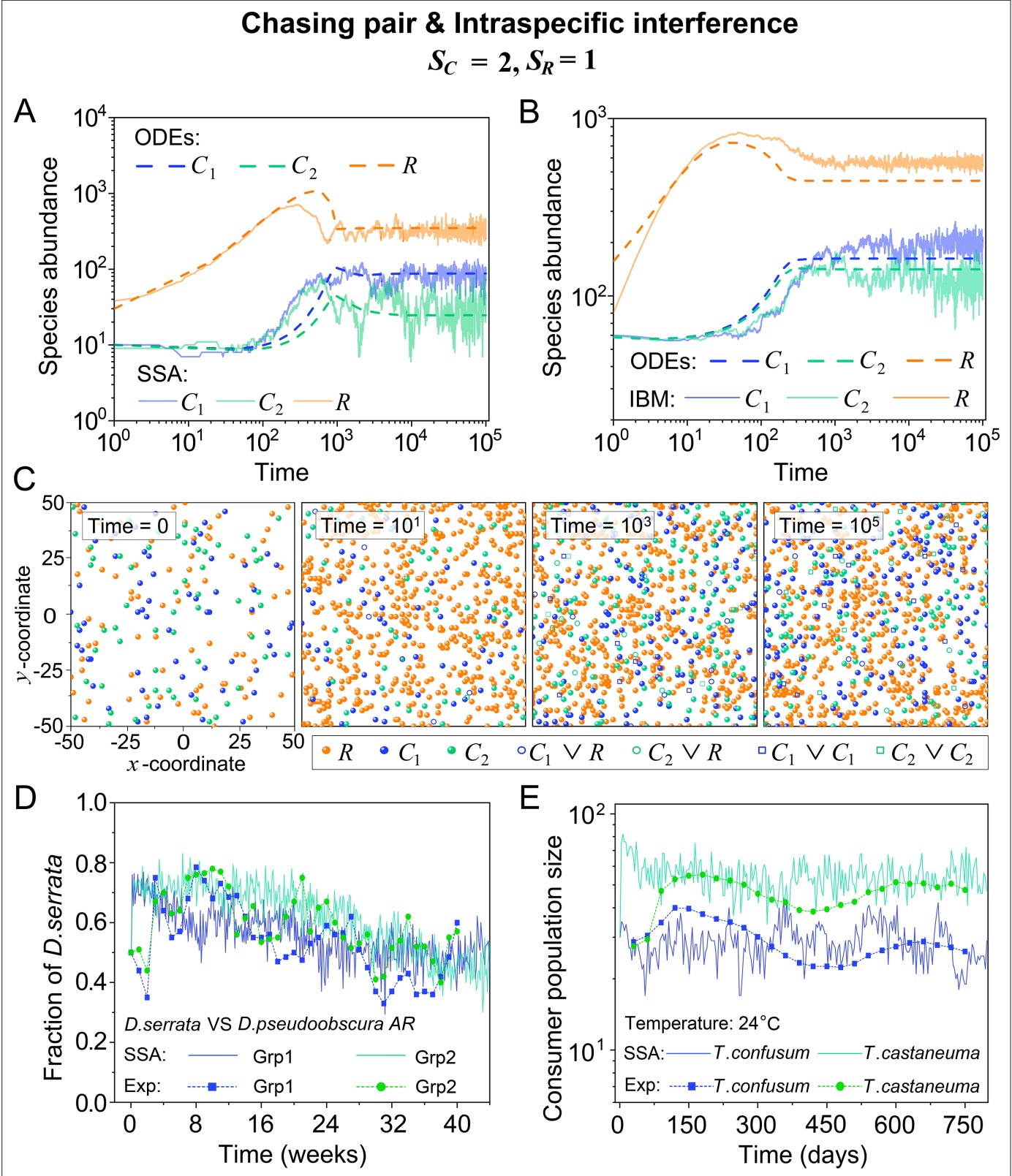

**Figure 2.** Intraspecific predator interference facilitates species coexistence regardless of stochasticity. (**A, B**) Time courses of the species abundances simulated with ODEs, SSA, or IBM. (**C**) Snapshots of the IBM simulations. (**D, E**) A model of intraspecific predator interference explains two classical laboratory experiments that invalidate CEP. (**D**) In Ayala's experiment, two *Drosophila* species coexist with one type of resources within a laboratory bottle (***Ayala, 1969***). (**E**) In Park's experiment, two *Tribolium* species coexist for 2 years with one type of food (flour) within a lab (***Park, 1954***). See ***Appendix 1—figure 7C, D*** for the comparison between model results and experimental data using Shannon entropies.

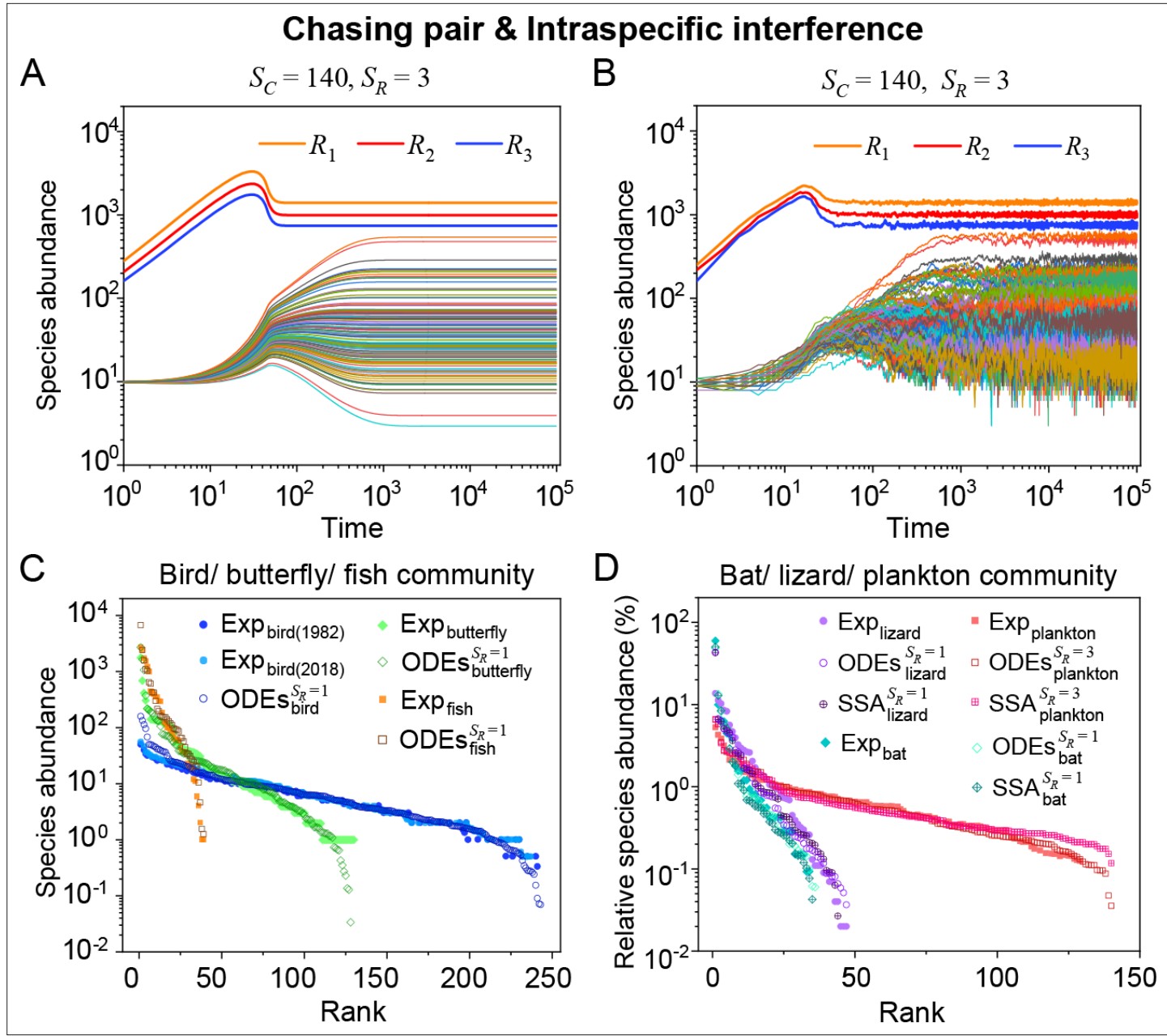

**Figure 3.** Intraspecific interference enables a wide range of consumer species to coexist with only one or a handful of resource species. (**A, B**) Representative time courses simulated with ODEs and SSA. (**C, D**) A model of intraspecific predator interference illustrates the S-shaped pattern of the species' rank-abundance curves across different ecological communities. The solid icons represent the experimental data (marked with 'Exp') reported in existing studies (*Fuhrman et al., 2008*; *Cody and Smallwood, 1996*; *Terborgh et al., 1990*; *Martínez et al., 2023*; *Clarke et al., 2005*; *De Vries, 1997*; *Hubbell, 2001*), where the bird community data were collected longitudinally in 1982 and 2018 (*Terborgh et al., 1990*; *Martínez et al., 2023*). The ODEs and SSA results were constructed from timestamp $t = 1.0 \times 10^5$ in the time series. In the K-S test, the probabilities ($p$-values) that the simulation results and the corresponding experimental data come from the same distributions are: $p_{\text{ODEs}}^{\text{bird}(1982)} = 0.17$, $p_{\text{ODEs}}^{\text{bird}(2018)} = 0.26$, $p_{\text{ODEs}}^{\text{butterfly}} = 0.70$, $p_{\text{ODEs}}^{\text{fish}} = 0.88$, $p_{\text{ODEs}}^{\text{bat}} = 0.42$, $p_{\text{SSA}}^{\text{bat}} = 0.48$, $p_{\text{ODEs}}^{\text{lizard}} = 0.96$, $p_{\text{SSA}}^{\text{lizard}} = 0.54$, $p_{\text{ODEs}}^{\text{plankton}} = 0.20$, $p_{\text{SSA}}^{\text{plankton}} = 0.06$. See Appendix 9 for simulation details and the Shannon entropies.

Strikingly, a plethora of consumer species may coexist at a steady state with only one resource species ($S_C \gg S_R$, $S_R = 1$) in the ODEs simulations, and crucially, the facilitated biodiversity can still be maintained in the SSA simulations. The long-term coexistence behavior is exemplified in *Figure 3* and *Appendix 1—figures 8–10*, involving simulations with or without stochasticity. The number of consumer species in long-term coexistence can be up to hundreds or more (see *Figure 3* and

*Appendix 1—figure 8*). To mimic real ecosystems, we further analyze cases with more than one type of resources, such as systems with $S_R = 3$ ($S_C \gg S_R$). Just like the case of $S_R = 1$ ($S_C \gg S_R$), an extensive range of consumer species may coexist indefinitely regardless of stochasticity (see *Figure 3* and *Appendix 1—figures 11–14*).

We further analyze the scenario involving chasing pairs and both intra- and interspecific interference, where multiple consumer species compete for one resource species. Similar to the scenario involving chasing pairs and intraspecific interference, all species coexist indefinitely in either ODEs or SSA simulation studies (see *Appendix 1—figure 5F–H* for the cases of $S_C = 6, 20$ and $S_R = 1$).

## Intuitive understanding: an underlying negative feedback loop

For the case with only one resource species ($S_R = 1$), if the total population size of the resources is much larger than that of the consumers (i.e. $R \gg \sum_{i=1}^{S_C} C_i$), the functional response $\mathcal{F} \equiv k_i x_i / C_i$ and the steady-state population of each consumer and resource species can be obtained analytically (see Appendix 4.B-C for details). In fact, the functional response of a consumer species (e.g. $C_i$) is negatively correlated with its own population size:

$$\mathcal{F}(R, C_i) \approx \frac{2R}{\sqrt{(R + K_i)^2 + 8\beta_i K_i^2 C_i} + R + K_i}, \tag{5}$$

where $\beta_i \equiv a_i'/d_i'$. The analytical steady-state solutions are highly consistent with the numerical results (see *Figure 1E* and *Appendix 1—figure 3H, I*) and can even quantitatively predict the coexistence region of the parameter space (see *Appendix 1—figure 3I*).

Intuitively, the mechanisms of how intraspecific interference facilitates species coexistence can be understood from the underlying negative feedback loop. Specifically, for consumer species of higher competitiveness (e.g. $C_i$) in an ecological community, as the population size of $C_i$ increases during competition, a larger portion of $C_i$ individuals are then engaged in intraspecific interference pairs which are temporarily absent from hunting (see *Equation S59* and *Appendix 1—figure 15A, B*). Consequently, the fraction of $C_i$ individuals within chasing pairs decreases (see *Equation S59* and *Appendix 1—figure 15A, B*) and thus form a self-inhibiting negative feedback loop through the functional response (see *Equation 5* and *Appendix 1—figure 15C*). This negative feedback loop prevents further increases in $C_i$ populations, results in an overall balance among the consumer species, and thus promotes biodiversity (see Appendix 4.C for details).

## The S shape pattern of the rank-abundance curves in a broad range of ecological communities

As mentioned above, a prominent feature of biodiversity is that the species' rank-abundance curves follow a universal S-shaped pattern in the linear-log plot across a broad spectrum of ecological communities (*Fuhrman et al., 2008*; *Ser-Giacomi et al., 2018*; *Cody and Smallwood, 1996*; *Terborgh et al., 1990*; *Martínez et al., 2023*; *Clarke et al., 2005*; *Hubbell, 2001*; *De Vries, 1997*). Previously, this pattern was mostly explained by the neutral theory (*Hubbell, 2001*), which requires special parameter settings that all consumer species share identical fitness. To resolve this issue, we apply the model involving chasing pairs and intraspecific interference to simulate the ecological communities, where one or three types of resources support a large number of consumer species ($S_C \gg S_R$). In each model system, the mortality rates of consumer species follow a Gaussian distribution where the coefficient of variation was taken around 0.3 (*Menon et al., 2003*; see Appendix 9 for details). For a broad array of the ecological communities, the rank-abundance curves obtained from the long-term coexisting state of both the ODEs and SSA simulation studies agree quantitatively with those of experiments (see *Figure 3C and D* and *Appendix 1—figures 8–14*), sharing roughly equal Shannon entropies and mostly being regarded as identical distributions in the Kolmogorov-Smirnov (K-S) statistical test (with a significance threshold of 0.05). Still, there is a noticeable discrepancy between the experimental data and SSA studies in terms of the species' absolute abundances (e.g. see *Appendix 1—figure 8C*): those with experimental abundances less than 10 tend to be extinct in the SSA simulations. This is due to the fact that the recorded individuals in an experimental sample are just a tiny portion of that in the real ecological system, whereas the species population size in a natural community is certainly much larger than 10.

## Discussion

The conflict between the CEP and biodiversity, exemplified by the paradox of the plankton (*Hutchinson, 1961*), is a long-standing puzzle in ecology. Although many mechanisms have been proposed to overcome the limit set by CEP (*Hutchinson, 1961*; *Chesson, 2000*; *Levins, 1979*; *Levin, 1974*; *Koch, 1974*; *Huisman and Weissing, 1999*; *Czárán et al., 2002*; *Goyal and Maslov, 2018*; *Goldford et al., 2018*; *Villa Martín et al., 2020*; *Gupta et al., 2021*; *Thingstad, 2000*; *Wang and Liu, 2020*; *Dalziel et al., 2021*; *Posfai et al., 2017*; *Weiner et al., 2019*; *Xue and Goldenfeld, 2017*; *Beddington, 1975*; *DeAngelis et al., 1975*; *Arditi and Ginzburg, 1989*; *Kelsic et al., 2015*; *Grilli et al., 2017*; *Ratzke et al., 2020*), it is still unclear how plankton and many other organisms can flout CEP and maintain biodiversity in quasi-well-mixed natural ecosystems. To address this issue, we investigate a mechanistic model with detailed consideration of pairwise encounters. Using numerical simulations combined with mathematical analysis, we identify that the intraspecific interference among the consumer individuals can promote a wide range of consumer species to coexist indefinitely with only one or a handful of resource species through the underlying negative feedback loop. By applying the above analysis to real ecological systems, our model naturally explains two classical experiments that reject CEP (*Ayala, 1969*; *Park, 1954*), and quantitatively illustrates the universal S-shaped pattern of the rank-abundance curves for a broad range of ecological communities (*Fuhrman et al., 2008*; *Ser-Giacomi et al., 2018*; *Cody and Smallwood, 1996*; *Terborgh et al., 1990*; *Martínez et al., 2023*; *Clarke et al., 2005*; *Hubbell, 2001*; *De Vries, 1997*).

In fact, predator interference has been introduced long ago by the classical B-D phenomenological model (*Beddington, 1975*; *DeAngelis et al., 1975*). However, the functional response of the B-D model involving intraspecific interference can be formally derived from the scenario involving only chasing pairs without consideration of pairwise encounters between consumer individuals (*Wang and Liu, 2020*; *Huisman and De Boer, 1997*; see *Equations S8a and S24a*). Yet, it has been demonstrated that the scenario involving only chasing pairs is under the constraint of CEP (*Wang and Liu, 2020*; see *Appendix 1—figure 3A–C*). Therefore, it is questionable regarding the validity of applying the B-D model to break CEP (*Cantrell et al., 2004*; *Hsu et al., 2013*). From a mechanistic perspective, we resolve these issues and show that the B-D model corresponds to a special case of our mechanistic model yet without the escape rate (see *Appendix 1—figure 2* and Appendix 3 for details).

Our model is broadly applicable to explain biodiversity in many ecosystems. In practice, many more factors are potentially involved, and special attention is required to disentangle confounding factors. In microbial systems, complex interactions are commonly involved (*Goyal and Maslov, 2018*; *Goldford et al., 2018*; *Hu et al., 2022*), and species' preference for food is shaped by the evolutionary course and environmental history (*Wang et al., 2019*). It is still highly challenging to fully explain how organisms evolve and maintain biodiversity in diverse ecosystems.

## Materials and methods

### Derivation of the encounter rates with the mean-field approximation

In the model depicted in *Figure 1A*, consumers and resources move randomly in space, which can be regarded as Brownian motions. At moment $t$, a consumer individual of species $C_i$ moves at speed $v_{C_i}$ with velocity $\boldsymbol{v}_{C_i}(t)$, while a resource individual of species $R_l$ moves at speed $v_{R_l}$ with velocity $\boldsymbol{v}_{R_l}(t)$. Here, $v_{C_i}$ and $v_{R_l}$ are two invariants, while the directions of $\boldsymbol{v}_{C_i}(t)$ and $\boldsymbol{v}_{R_l}(t)$ change constantly. The relative velocity between the two individuals is $\boldsymbol{u}_{C_i-R_l}(t) \equiv \boldsymbol{v}_{R_l}(t) - \boldsymbol{v}_{C_i}(t)$, with a relative speed of $u_{C_i-R_l}(t)$. Then, $u_{C_i-R_l}(t)^2 = v_{C_i}^2 + v_{R_l}^2 - 2v_{C_i} \cdot v_{R_l} \cdot \cos\theta_{C_i-R_l}(t)$, where $\theta_{C_i-R_l}(t)$ represents the angle between $\boldsymbol{v}_{C_i}(t)$ and $\boldsymbol{v}_{R_l}(t)$. This system is homogeneous, thus, $\overline{\cos\theta_{C_i-R_l}} = 0$, where the overline stands for the temporal average. Then, we obtain the average relative speed between the $C_i$ and $R_l$ individuals: $\overline{u_{C_i-R_l}} = \sqrt{v_{C_i}^2 + v_{R_l}^2}$. Likewise, the average relative speed between the $C_i$ and $C_j$ individuals is $\overline{u_{C_i-C_j}} = \sqrt{v_{C_i}^2 + v_{C_j}^2}$. Evidently, $\overline{u_{C_i-C_i}} = \sqrt{2}v_{C_i}$. Meanwhile, the concentrations of species $C_i$ and $R_l$ in a 2-D square system with a length of $L$ are $n_{C_i} = C_i/L^2$ and $n_{R_l} = R_l/L^2$, while those of the freely wandering $C_i$ and $R_l$ are $n_{C_i^{(F)}} = C_i^{(F)}/L^2$ and $n_{R_l^{(F)}} = R_l^{(F)}/L^2$.

Then, we use the mean-field approximation to calculate the encounter rates $a_{il}$ and $a_{ij}'$ in the well-mixed system. In particular, we estimate $a_{il}$ by tracking a randomly chosen consumer individual from species $C_i$ and counting its encounter frequency with the freely wandering individuals

from resource species $R_l$ (see *Appendix 1—figure 1*). At any moment, the consumer individual may form a chasing pair with a $R_l$ individual within a radius of $r_{il}^{(C)}$ (see *Figure 1A*). Over a time interval of $\Delta t$, the number of encounters between the consumer individual and $R_l$ individuals can be estimated by the encounter area and the concentration $n_{R_l}$, which takes the value of $2r_{il}^{(C)} n_{R^{(F)}} \overline{u_{C_i - R}} \Delta t$ (see *Appendix 1—figure 1*). Combined with $n_{R_l^{(F)}} = R_l^{(F)}/L^2$, for all freely wandering $C_i$ individuals, the number of their encounters with $R^{(F)}$ during interval $\Delta t$ is $\frac{2r_{il}^{(C)} \overline{u_{C_i - R}} C_i^{(F)} R^{(F)}}{L^2} \Delta t$. Meanwhile, in the ODEs, this corresponds to $a_i C_i^{(F)} R^{(F)} \Delta t$. Comparing both terms above, for chasing pairs, we have $a_{il} = 2r_{il}^{(C)} L^{-2} \overline{u_{C_i - R_l}} = 2r_{il}^{(C)} L^{-2} \sqrt{v_{C_i}^2 + v_{R_l}^2}$. Likewise, for interference pairs, we obtain $a'_{ij} = 2r_{ij}^{(I)} L^{-2} \overline{u_{C_i - C_j}} = 2r_{ij}^{(I)} L^{-2} \sqrt{v_{C_i}^2 + v_{C_j}^2}$. In particular, $a'_{ii} = 2\sqrt{2} v_{C_i} r_{ii}^{(I)} L^{-2}$.

## Stochastic simulations

To investigate the impact of stochasticity on species coexistence, we use the stochastic simulation algorithm (SSA; *Gillespie, 2007*) and individual-based modeling (IBM; *Vetsigian, 2017*; *Grimm, 2013*) in simulating the stochastic process. In the SSA studies, we follow the standard Gillespie algorithm and simulation procedures (*Gillespie, 2007*).

In the IBM studies, we consider a 2D square system with a length of $L$ and periodic boundary conditions. In the case of $S_C = 2$ and $S_R = 1$, consumers of species $C_i$ move at speed $v_{C_i}$, while the resources move at speed $v_R$. The unit length is $\Delta l = 1$, and all individuals move probabilistically. Specifically, when $\Delta t$ is small so that $v_{C_i} \Delta t \ll 1$, $C_i$ individuals jump a unit length with the probability $v_{C_i} \Delta t$. In practice, we simulate the temporal evolution of the model system following the procedures below.

### Initialization

We choose the initial position for each individual randomly from a uniform distribution in the square space, which rounds to the nearest integer point in the $x$-$y$ coordinates.

### Moving

We choose the destination of a movement randomly from four directions ($x$-positive, $x$-negative, $y$-positive, $y$-negative) following a uniform distribution. The consumers and resources jump a unit length with probabilities $v_{C_i} \Delta t$ and $v_R \Delta t$, respectively.

### Forming pairs

When a $C_i$ individual and a resource individual get close in space within a distance of $r^{(C)}$, they form a chasing pair. Meanwhile, when two consumer individuals $C_i$ and $C_j$ stand within a distance of $r^{(I)}$, they form an interference pair.

### Dissociation

We update the system with a small time step $\Delta t$ so that $d_i \Delta t, k_i \Delta t, d'_{ij} \Delta t \ll 1$. In practice, a random number $\varsigma$ is sampled from a uniform distribution between 0 and 1, that is $\mathcal{U}(0, 1)$. If $\varsigma < d_i \Delta t$ or $\varsigma < d'_{ij} \Delta t$, then the chasing pair or interference pair dissociates into two separated individuals. One occupies the original position, while the other individual moves just out of the encounter radius in a uniformly distributed random angle. For a chasing pair, if $d_i \Delta t < \varsigma < (d_i + k_i) \Delta t$, then, the consumer catches the resource, and the biomass of the resource flows into the consumer populations (updated according to the birth procedure), while the consumer individual occupies the original position. Finally, if $\varsigma > (d_i + k_i) \Delta t$ or $\varsigma > d'_{ij} \Delta t$, the chasing pair or interference pair maintains the current status.

### Birth and death

For each species, we use two separate counters with decimal precision to record the contributions of the birth and death processes, both of which accumulate in each time step. The incremental integer part of the counter will trigger updates in this run. Specifically, a newborn would join the system following the initialization procedure in a birth action, while an unfortunate target would be randomly chosen from the living population in a death action.

## Acknowledgements

We thank Roy Kishony, Eric D Kelsic and Yang-Yu Liu for helpful discussions. This work was supported by National Natural Science Foundation of China (Grant No.12004443), Guangzhou Municipal Innovation Fund (Grant No. 202102020284) and the Hundred Talents Program of Sun Yat-sen University.

## Additional information

### Funding

| Funder | Grant reference number | Author |
|---|---|---|
| National Natural Science Foundation of China | 12004443 | Xin Wang |
| Guangzhou Municipal Science and Technology Bureau | 202102020284 | Xin Wang |
| Sun Yat-sen University | The Hundred Talents Program | Xin Wang |

The funders had no role in study design, data collection and interpretation, or the decision to submit the work for publication.

### Author contributions
Ju Kang, Data curation, Software, Formal analysis, Validation, Investigation, Visualization, Writing – original draft; Shijie Zhang, Software, Investigation, Visualization; Yiyuan Niu, Data curation, Software, Validation, Investigation, Visualization; Fan Zhong, Investigation; Xin Wang, Conceptualization, Data curation, Formal analysis, Supervision, Funding acquisition, Validation, Investigation, Writing – original draft, Project administration, Writing – review and editing

### Author ORCIDs
Ju Kang http://orcid.org/0000-0003-3862-4065
Yiyuan Niu https://orcid.org/0009-0008-6078-9265
Xin Wang https://orcid.org/0000-0001-6479-395X

Reviewer #1 (Public Review): https://doi.org/10.7554/eLife.93115.3.sa1
Reviewer #3 (Public Review): https://doi.org/10.7554/eLife.93115.3.sa2
Author response https://doi.org/10.7554/eLife.93115.3.sa3

## Additional files

### Supplementary files
• MDAR checklist

### Data availability
All data and codes for this paper are available on GitHub (copy archived at *SchordK, 2024*).

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

# Appendix 1

## Appendix tables and figures

**Appendix 1—table 1.** Illustrations of symbols in our generic model of pairwise encounters.

| Symbols | Illustrations |
| --- | --- |
| $C_i$ | The total population of consumer species $C_i$. |
| $R_l$ | The total population of resource species $R_l$. |
| $C_i^{(F)}$ | The freely wandering population of consumer species $C_i$. |
| $R_l^{(F)}$ | The freely wandering population of resource species $R_l$. |
| $x_{il}$ | Chasing pairs formed between individuals from species $C_i$ and $R_l$, i.e. $C_i^{(P)} \bigvee R_l^{(P)}$. |
| $y_i$ | Intraspecific interference pairs formed between individuals from species $C_i$, i.e. $C_i^{(P)} \bigvee C_i^{(P)}$. |
| $z_{ij}$ | Interspecific interference pairs formed between individuals from species $C_i$ and $C_j$, i.e., $C_i^{(P)} \bigvee C_j^{(P)}$. |
| $r_{il}^{(C)}$ | The upper distance criterion for forming a chasing pair. |
| $r_{ij}^{(I)}$ | The upper distance criterion for forming an interference pair. |
| $v_{C_i}$ | The motility speed of consumer species $C_i$. |
| $v_{R_l}$ | The motility speed of resource species $R_l$. |
| $S_C$ | The number of consumer species. |
| $S_R$ | The number of resource species. |
| $a_{il}$ | The encounter rate between a consumers and a resource. |
| $d_{il}$ | The escape rate within a chasing pair. |
| $k_{il}$ | The capture rate within a chasing pair. |
| $a'_{ij}$ | The encounter rate among consumer individuals. |
| $d'_{ij}$ | The separation rate within an interference pair. |
| $w_{il}$ | The mass conversion ratio from resource $R_l$ to $C_i$. |
| $D_i$ | The mortality rate of species $C_i$. |
| $\kappa_l$ | The steady-state population abundance of resources species $R_l$ in the absence of consumers. |
| $\zeta_l$ | The external resource supply rate of species $R_l$. |
| $\eta_l$ | The intrinsic growth rate of species $R_l$ for biotic resources (unused in all analyses). |
| $g_l$ | The function describing the population dynamics of resource species $R_l$. |
| $L$ | The length of the 2-D square system where species coexist. |
| $C_i^{(F)}(+)$ | We count $C^{(F)}(+)$ as $C^{(F)}$, where "(+)" signifies gaining biomass from resources. |
| $\boldsymbol{v}_{C_i}$ | The velocity of an individual of species $C_i$. |
| $\boldsymbol{v}_{R_l}$ | The velocity of an individual of species $R_l$. |
| $\theta_{C_i - R_l}$ | The angle between $\boldsymbol{v}_{C_i}$ and $\boldsymbol{v}_{R_l}$. |
| $\boldsymbol{u}_{C_i - R_l}$ | The relative velocity between a consumer and a resource. |

*Appendix 1—table 1 Continued on next page*

*Appendix 1—table 1 Continued*

| Symbols | Illustrations |
| --- | --- |
| $u_{C_i - R_l}$ | The relative speed between a consumer and a resource. |
| $n_{C_i}$ | The concentration of species $C_i$. |
| $n_{R_l}$ | The concentration of species $R_l$. |
| $n_{C_i^{(F)}}$ | The concentration of the freely wandering $C_i$. |
| $n_{R_l^{(F)}}$ | The concentration of the freely wandering $R_l$. |

For all the symbols in **Appendix 1—tables 1 and 2**, the subscript '$l$' is omitted if $S_R = 1$, and the subscript '$i$' is omitted if $S_C = 1$.

**Appendix 1—table 2.** Illustrations of other symbols used in our manuscript.

| Symbols | Illustrations/Definitions |
| --- | --- |
| $\mathcal{F}$ | The functional response. |
| $\Xi$ | The searching efficiency. |
| $\varsigma$ | A random number sampled from a uniform distribution. |
| $\mathcal{U}$ | The uniform distribution. |
| $\mathcal{N}(\mu, \sigma)$ | A Gaussian distribution with a mean of μ and a standard deviation of σ. |
| $\equiv$ | An equal sign for equations defining the symbol on the left-hand side. |
| $\Delta$ | The competitive difference between two consumer species, defined as $\Delta \equiv (D_1 - D_2)/D_2$. |
| $\widehat{\Delta}$ | The supremum of the competitive difference tolerated for species coexistence. |
| $\Delta t$ | A short time interval. |
| $\mathcal{P}_i$ | The probability that a consumer individual of the ecological community belongs to species $C_i$. |
| $p_{\text{ODEs}}$, $p_{\text{SSA}}$ | The *p*-value assessing the similarity of simulation results and experimental data. |
| $H$ | The Shannon entropy: $$H = -\sum_{i=1}^{S_C} \mathcal{P}_i \log_2(\mathcal{P}_i).$$ |
| $\overline{X}$ | The time average of an arbitrary quantity $X$. |
| $\delta X$ | The standard deviation of an arbitrary quantity $X$. |
| $\langle X \rangle$ | The expectation of a random variable $X$. |
| $\tau$ | The parameter that sets the time scale of a system. |
| $K_{il}$ | $K_{il} \equiv \frac{k_{il} + d_{il}}{a_{il}}$. |
| $\alpha_{il}$ | $\alpha_{il} \equiv \frac{D_i}{w_{il} k_{il}}$. |
| $\beta_i$ | $\beta_i \equiv \frac{a_i'}{d_i''}$. |
| $\gamma$ | $\gamma \equiv \frac{a_{12}'}{d_{12}'}$. |
| $f_i(R^{(F)})$ | $f_i(R^{(F)}) \equiv R^{(F)}/(R^{(F)} + K_i)$. |
| $\phi_0, \phi_1, \phi_2$ | $\phi_0 \equiv -CR^2$, $\phi_1 \equiv 2CR + KR + R^2$, $\phi_2 \equiv 2\beta K^2 - K - C - 2R$. |
| $\Lambda$ | The discriminant of **Equation S13**. |
| $\psi, \varphi$ | $\psi \equiv \phi_1 - (\phi_2)^2/3$, $\varphi \equiv \phi_0 - \phi_1\phi_2/3 + 2(\phi_2)^3/27$. |

*Appendix 1—table 2 Continued on next page*

*Appendix 1—table 2 Continued*

| Symbols | Illustrations/Definitions |
| --- | --- |
| $\omega, \theta_1, \theta_2$ | $\omega \equiv -1/2 + i\sqrt{3}/2$, $\theta_1 \equiv (-\varphi/2 + \sqrt{-\Lambda/108})^{1/3}$, $\theta_2 \equiv (-\varphi/2 - \sqrt{-\Lambda/108})^{1/3}$. |
| $\psi', \varphi'$ | $\psi' \equiv (-4\psi/3)^{1/2}$, $\varphi' \equiv \arccos(-(-\psi/3)^{-3/2}\varphi/2)/3$. |
| $t_h$ | The handling time in the B-D model. |
| $t_w$ | The wasting time in the B-D model. |
| $u_i(R, C_1, C_2)$ | Expression of $x_i$ using $C_1$, $C_2$, and $R$ in **Equation S33** involving intraspecific interference, see **Equation S37**. |
| $\Omega_i(R, C_1, C_2)$ | $\Omega_i(R, C_1, C_2) \equiv \frac{w_i k_i}{C_i} u_i(R, C_1, C_2)$. |
| $G(R, C_1, C_2)$ | $G(R, C_1, C_2) \equiv g(R, u_1(R, C_1, C_2), u_2(R, C_1, C_2), C_1, C_2)$, see **Equations S33 and S38**. |
| $o_1, o_2$ | $o_1 \equiv \frac{\zeta}{\kappa} - \frac{k_1}{2\beta_1 K_1} - \frac{k_2}{2\beta_2 K_2}$, $o_2 \equiv \frac{k_1(1-\alpha_1)}{2\beta_1\alpha_1(K_1)^2} + \frac{k_2(1-\alpha_2)}{2\beta_2\alpha_2(K_2)^2}$. |
| $\varpi$ | $\varpi \equiv \frac{1}{2}(\frac{1}{\kappa} - \frac{k_2}{2\zeta\beta_2 K_2}) + \frac{1}{2}\sqrt{(\frac{1}{\kappa} - \frac{k_2}{2\zeta\beta_2 K_2})^2 + 2\frac{k_2(1-\alpha_2)}{\zeta\beta_2\alpha_2(K_2)^2}}$. |
| $\iota_1, \iota_2$ | $\iota_1 \equiv \frac{\zeta}{\kappa} - \sum\limits_{i=1}^{S_C} \frac{k_i}{2\beta_i K_i}$, $\iota_2 \equiv \sum\limits_{i=1}^{S_C} \frac{k_i(1-\alpha_i)}{2\beta_i\alpha_i(K_i)^2}$. |
| $u'_i(R, C_1, C_2)$ | Expression of $x_i$ using $C_1$, $C_2$, and $R$ in **Equation S61** involving interspecific interference, see **Equation S64**. |
| $\Omega'_i(R, C_1, C_2)$ | $\Omega'_i(R, C_1, C_2) \equiv \frac{w_i k_i}{C_i} u'_i(R, C_1, C_2)$. |
| $G'(R, C_1, C_2)$ | $G'(R, C_1, C_2) \equiv g(R, u'_1(R, C_1, C_2), u'_2(R, C_1, C_2), C_1, C_2)$, see **Equation S61 and S65**. |
| $\varrho_1, \varrho_2$ | $\varrho_1 \equiv \frac{\zeta}{\kappa} - \frac{k_1}{\gamma K_1} - \frac{k_2}{\gamma K_2}$, $\varrho_2 \equiv \frac{k_1(1-\alpha_2)}{\gamma K_1 K_2 \alpha_2} + \frac{k_2(1-\alpha_1)}{\gamma K_1 K_2 \alpha_1}$. |
| $\chi_1$ | $\chi_1 \equiv \frac{k_2\gamma(\alpha_1-1)}{K_1 K_2 \alpha_1(4\beta_1\beta_2-\gamma^2)} + \frac{k_1\gamma(\alpha_2-1)}{K_1 K_2 \alpha_2(4\beta_1\beta_2-\gamma^2)} - \frac{k_1 2\beta_2(\alpha_1-1)}{K_1^2\alpha_1(4\beta_1\beta_2-\gamma^2)} - \frac{k_2 2\beta_1(\alpha_2-1)}{K_2^2\alpha_2(4\beta_1\beta_2-\gamma^2)}$. |
| $\chi_2$ | $\chi_2 \equiv \frac{k_1(\gamma-2\beta_2)}{K_1(4\beta_1\beta_2-\gamma^2)} + \frac{k_2(\gamma-2\beta_1)}{K_2(4\beta_1\beta_2-\gamma^2)} + \frac{\zeta}{\kappa}$. |

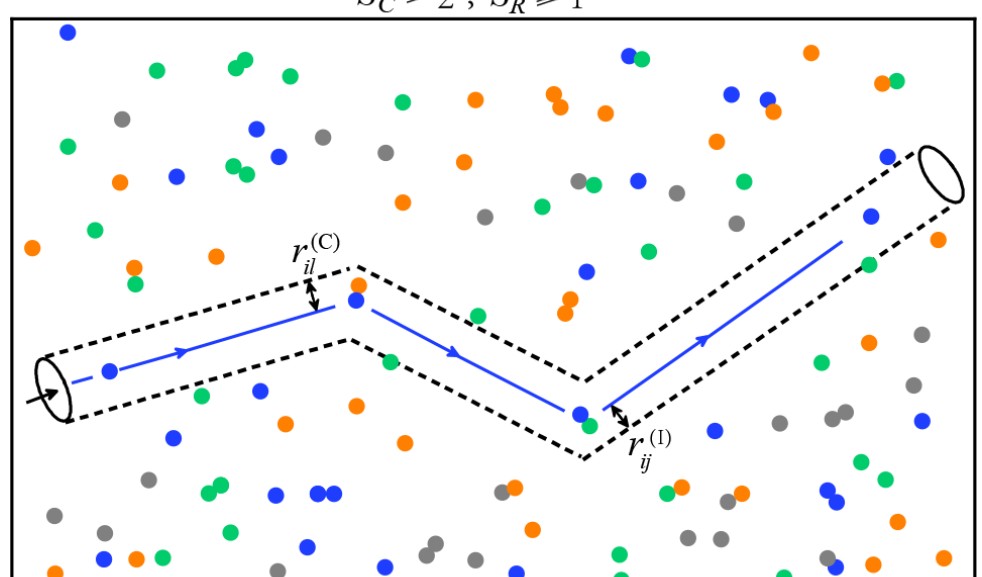

**Appendix 1—figure 1.** Estimation of the encounter rates with the mean-field approximations. To calculate $a_{il}$ in the chasing pair, we suppose that all individuals of species $R_l$ stand still while a $C_i$ individual moves at the speed of $\overline{u_{C_i - R_l}}$ (the relative speed). Over a time interval of $\Delta t$, the length of zigzag trajectory of the $C_i$ individual is approximately $\overline{u_{C_i - R_l}} \Delta t$, while the encounter area (marked with dashed lines) is estimated to be $2 r_{il}^{(C)} \overline{u_{C_i - R}} \Delta t$. Then, we can estimate the encounter rate $a_{il}$ using the encounter area and concentrations of the species (see Materials and methods for details). Similarly, we can estimate $a'_{ij}$ in the interference pair.

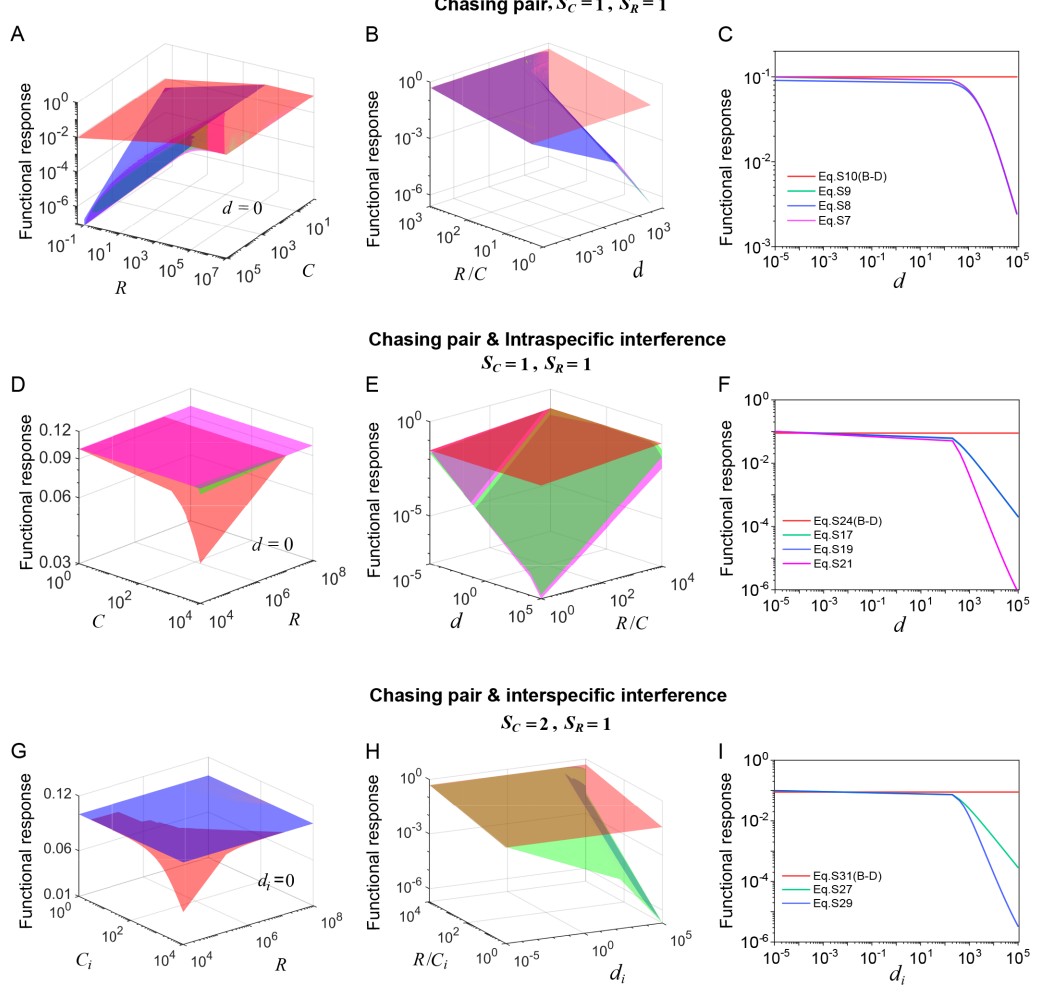

**Appendix 1—figure 2.** Functional response in scenarios involving different types of pairwise encounter. (**A–C**) In the scenario involving only chasing pair, the red surface/line corresponds to the B-D model (calculated from *Equation S10b*), while the green surface/line represents the exact solutions to our mechanistic model (calculated from *Equation S7b*). The magenta (calculated from *Equation S8b*) and blue (calculated from *Equation S9b*) surfaces/lines represent the approximate solutions to our model (see Appendix 3.B). (**D–F**) In the scenario involving chasing pairs and intraspecific interference, the red surface/line corresponds to the B-D model (calculated from *Equation S24b*), while the green/line surface represents the exact solutions to our mechanistic model (calculated from *Equation S17b*). The blue surface/line (calculated from *Equation S19b*) and the magenta surface/line (calculated from *Equation S21b*) represent the quasi-rigorous and the approximate solutions to our model, respectively (see Appendix 3.C). (**G–I**) In the scenario involving chasing pairs and interspecific interference, the red surface/line corresponds to the B-D model (calculated from *Equation S31b*), while the green surface/line represents the quasi-rigorous solutions to our mechanistic model (calculated from *Equation S27d*). The blue surface/line (calculated from *Equation S29d*) represents the approximate solutions to our model (see Appendix 3.D). In (**A–C**): $k = 0.1$, $a = 0.25$. In (**A**): $d = 0$. In (**C**): $R = 10^4$, $C = 10^3$. In (**D–F**): $a = 0.1$, $k = 0.1$, $a' = 0.12$, $d' = 0.1$. In (**D**): $d = 0$. In (**F**): $R = 10^6$, $C = 10^5$. In (**G–I**): $a_1 = a_2 = 0.1$, $k_1 = k_2 = 0.1$, $a'_{12} = 0.12$, $d'_{12} = 0.1$. In (**G**): $d_i = 0$. In (**I**): $R = 10^6$, $C_i = 10^5$.

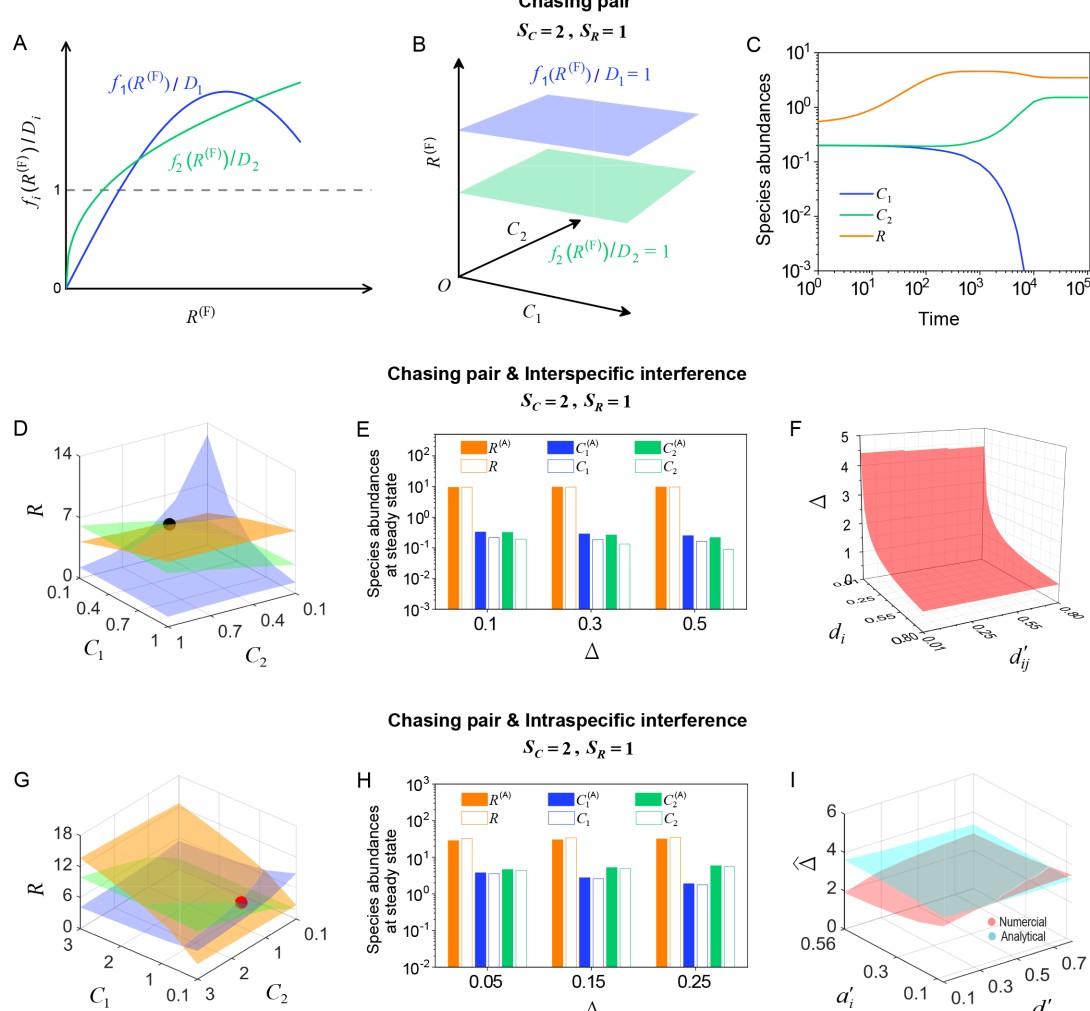

**Appendix 1—figure 3.** Numerical solutions in scenarios involving chasing pairs and different types of predator interference. Here, $S_C = 2$ and $S_R = 1$. $D_i (i = 1, 2)$ is the only parameter varying with the consumer species (with $D_1 > D_2$), and $\Delta \equiv (D_1 - D_2)/D_2$ represents the competitive difference between the two species. (**A–C**) Scenario involving only chasing pairs. (**A**) If all consumer species coexist at steady state, then $f_i(R^{(F)})/D_i = 1$, where $f_i(R^{(F)}) = R^{(F)}/(R^{(F)} + K_i)$ and $K_i = (d_i + k_i)/a_i$. This requires that the three lines $y = f_i(R)/D_i$ and $y = 1$ share a common point, which is generally impossible. (**B**) The blue plane is parallel to the green one, and hence they do not have a common point. (**C**) Time courses of the species abundances in the scenario involving only chasing pairs. The two consumer species cannot coexist at steady state. (**D–F**) Scenario involving chasing pairs and interspecific interference. (**G–I**) Scenario involving chasing pairs and intraspecific interference. (**D, G**) Positive solutions to the steady-state equations: $\dot{R} = 0$ (orange surface), $\dot{C}_1 = 0$ (blue surface), $\dot{C}_2 = 0$ (green surface). The intersection point marked by black/red dots is an unstable/stable fixed point. (**E, H**) Comparisons between the numerical results and analytical solutions of the species abundances at fixed points. Color bars are analytical solutions while hollow bars are numerical results. The analytical solutions in (**E**) and (**H**) (marked with superscript '(A)') were calculated from *Equations S68 and S70* and *Equation S41*, *Equation S43*, respectively. (**F**) In this scenario, there is no parameter region for stable coexistence. The region below the red surface and above $\Delta = 0$ represents unstable fixed points. (**I**) Comparisons between the numerical results and analytical solutions of the coexistence region. Here, $\widehat{\Delta}$ represents the maximum competitive difference tolerated for species coexistence. The red and cyan surfaces represent the analytical solutions (calculated from *Equation S46*) and numerical results, respectively. The numerical results in (**C**), (**D–F**) and (**G–I**) were calculated from *Equations 1 and 4*, *Equation S42 and S61* and *Equation S33 and S42*, respectively. In (**C**): $a_i = 0.1$, $k_i = 0.1$, $w_i = 0.1$, $d_i = 0.5$, $(i = 1, 2)$, $D_1 = 0.002$, $D_2 = 0.001$, $\kappa = 5$, $\zeta = 0.05$. In (**D**): $a_i = 0.05$, $d_i = 0.05$, $k_i = 0.02$, $w_i = 0.08$, $(i = 1, 2)$, $D_1 = 0.0011$, $D_2 = 0.001$, $\kappa = 20$, $\zeta = 0.01$, $a'_{12} = 0.06$, $d'_{12} = 0.01$. In (**E**): $a_i = 0.04$, $d_i = 0.2$, $k_i = 0.1$, $w_i = 0.3$, $(i = 1, 2)$, $D_2 = 0.0008$, $\kappa = 10$, $\zeta = 0.2$, $a'_{12} = 0.048$, $d'_{12} = 0.001$. In (**F**): $a_i = 0.1$, $k_i = 0.1$, $w_i = 0.1$, $(i = 1, 2)$, $D_2 = 0.001$, $\kappa = 100$, $\zeta = 0.05$, $a'_{12} = 0.12$. In (**G**): $a_i = 0.5$, $a'_i = 0.625$, $d_i = 0.5$, $d'_i = 0.5$, $k_i = 0.4$, $w_i = 0.5$,

*Appendix 1—figure 3 continued on next page*

*Appendix 1—figure 3 continued*

$(i = 1, 2)$, $D_2 = 0.02$, $D_1 = 1.2D_2$, $\kappa = 10$, $\zeta = 0.1$. In (**H**): $a_i = 0.1$, $a_i' = 0.12$, $k_i = 0.12$, $w_i = 0.3$, $d_i = 0.5$, $d_i' = 0.05$, $(i = 1, 2)$, $D_2 = 0.02$, $\kappa = 100$, $\zeta = 0.8$. In (**I**): $a_i = 0.5$, $k_i = 0.2$, $d_i = 0.8$, $w_i = 0.2$, $(i = 1, 2)$, $D_2 = 0.008$, $\kappa = 60$, $\zeta = 0.8$.

**Chasing pair & Intraspecific interference**

$S_C = 2$, $S_R = 1$

**Appendix 1—figure 4.** Intraspecific predator interference facilitates species coexistence regardless of stochasticity. Here, we consider the case of $S_C = 2$, $S_R = 1$. (**A**) A representative trajectory of species coexistence in the phase space simulated with ODEs. The fixed point (shown in red) is stable and globally attractive. (**B, C**) 3D phase diagrams in the ODEs studies. Here, $D_i$ is the only parameter that varies with the two consumer species, and $\Delta \equiv (D_1 - D_2)/D_2$ measures the competitive difference between the two species. The parameter region below the blue surface yet above the red surface represents stable coexistence. The region below the red surface and above $\Delta = 0$ represents unstable fixed points (an empty set). (**D**) An exemplified transection corresponding to the $\Delta = 0.3$ plane in (**C**). (**E**) Time courses of the species abundances simulated with ODEs or SSA. (**F**) Representative trajectories of species coexistence in the phase space simulated with SSA. The coexistence state is stable and globally attractive (see (**E**) for time courses, SSA results). (**A–F**) were calculated or simulated from *Equations 1, 2 and 4*. In (**A**): $a_i = 0.1$, $a_i' = 0.125$, $d_i = 0.1$, $d_i' = 0.05$, $w_i = 0.1$, $k_i = 0.1$, $(i = 1, 2)$, $D_1 = 0.0035$, $D_2 = 0.0038$, $\kappa = 100$, $\zeta = 0.3$. In (**B**): $a_i = 0.1$, $d_i = 0.1$, $w_i = 0.1$, $k_i = 0.1$, $(i = 1, 2)$, $D_2 = 0.001$, $\Delta \equiv (D_1 - D_2)/D_2$, $\kappa = 100$, $\zeta = 0.1$. In (**C, D**): $a_i = 0.05$, $a_i' = 0.065$, $w_i = 0.1$, $k_i = 0.1$, $(i = 1, 2)$, $D_2 = 0.002$, $\Delta \equiv (D_1 - D_2)/D_2$, $\kappa = 10$, $s = 0.1$. In (**D**): $\Delta = 0.3$. In (**E, F**): $a_i = 0.02$, $a_i' = 0.025$, $d_i = 0.7$, $d_i' = 0.7$, $w_i = 0.4$, $k_i = 0.05$, $(i = 1, 2)$, $D_1 = 0.0160$, $D_2 = 0.0171$, $\kappa = 2000$, $\zeta = 5.5$.

**Chasing pair & Interspecific and intraspecific interference**

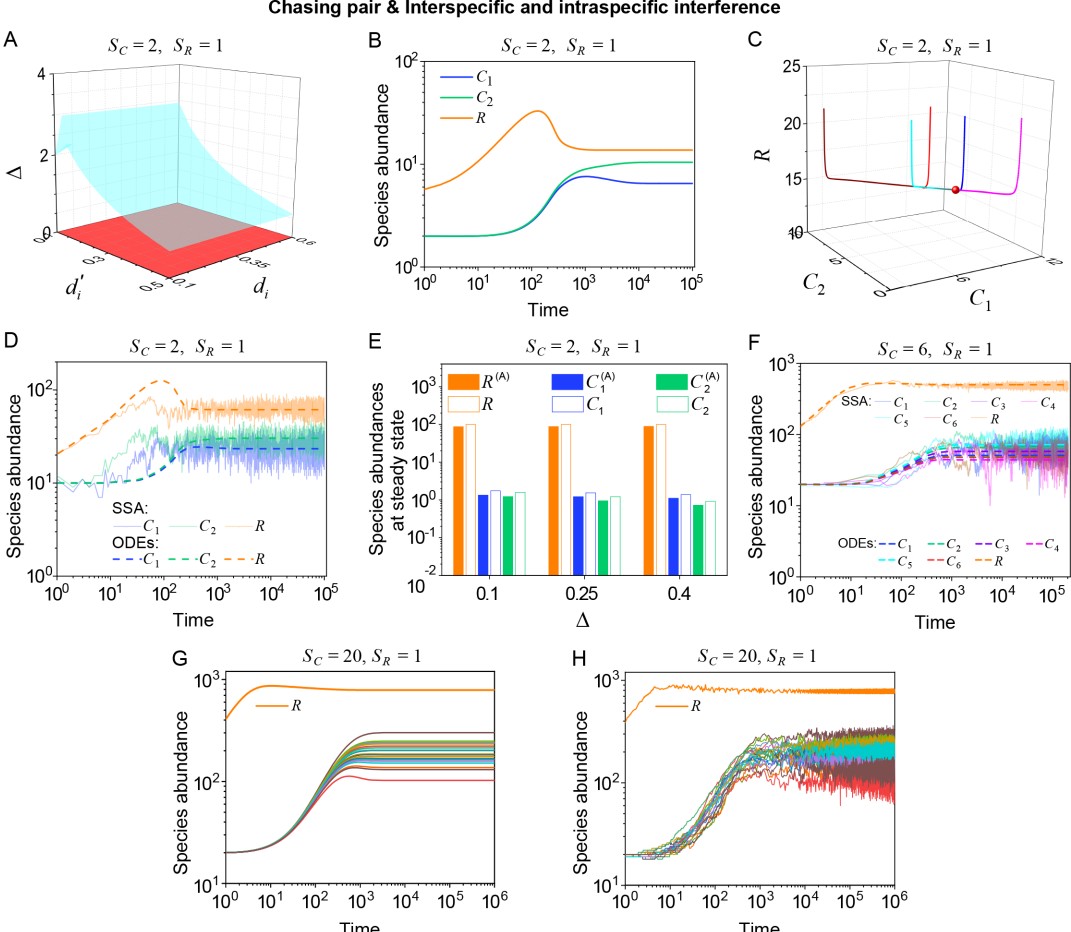

**Appendix 1—figure 5.** Outcomes of multiple consumers species competing for one resource species involving chasing pairs and intra- and inter-specific interference. (**A–E**) The case involving two consumer species and one resource species ($S_C = 2$, $S_R = 1$). Here, $D_i$ is the only parameter that varies with the consumer species (with $D_1 > D_2$), and $\Delta \equiv (D_1 - D_2)/D_2$ measures the competitive difference between the two species. (**A**) A 3D phase diagram. The parameter region below the blue surface yet above the red surface represents stable coexistence, while that below the red surface and above $\Delta = 0$ represents unstable fixed points (an empty set). (**B**) Time courses of the species abundances. Two consumer species may coexist with one type of resources at steady state. (**C**) Representative trajectories of species coexistence in the phase space. The fixed point (shown in red) is stable and globally attractive. (**D**) Consumer species may coexist indefinitely with the resources regardless of stochasticity. (**E**) Comparisons between numerical results and analytical solutions of the species abundances at fixed points. Color bars are analytical solutions while hollow bars are numerical results. The analytical solutions (marked with superscript '(A)') were calculated from *Equation S74 and S75*. (**F–H**) Time courses of species abundances in cases involving 6 or 20 consumer species and one type of resources ($S_C = 6$ or 20, $S_R = 1$). All consumer species may coexist with one type of resource at a steady state, and this coexisting state is robust to stochasticity. (**A–C, E, G**) ODEs results. (**D, F**) ODEs and SSA results. (**H**) SSA results. The numerical results in (**A–H**) were calculated or simulated from *Equations 1-4*. In (**A–C**): $a_i = 0.1$, $a_i' = 0.12$, $k_i = 0.1$, $w_i = 0.1$, $(i = 1, 2)$, $D_2 = 0.004$, $\kappa = 100$, $\zeta = 0.8$, $a_{12}' = 0.12$, $d_{12}' = 0.5$. In (**B–C**): $d_i = 0.3$, $d_i' = 0.5$, $(i = 1, 2)$. In (**D**): $a_i = 0.1$, $a_i' = 0.14$, $k_i = 0.12$, $w_i = 0.15$, $d_i = 0.3$, $d_i' = 0.5$, $(i = 1, 2)$, $D_1 = 0.0125$, $D_2 = 0.012$, $\kappa = 300$, $\zeta = 5.5$, $a_{12}' = 0.14$, $d_{12}' = 5$. In (**E**): $a_i = 0.05$, $a_i' = 0.06$, $k_i = 0.1$, $w_i = 0.2$, $d_i = 0.5$, $d_i' = 0.15$, $(i = 1, 2)$, $D_2 = 0.008$, $\kappa = 100$, $\zeta = 0.8$, $a_{12}' = 0.06$, $d_{12}' = 0.0005$. In (**F**): $a_i = 0.1$, $a_i' = 0.14$, $k_i = 0.15$, $w_i = 0.18$, $d_i = 2.8$, $d_i' = 0.02$, $a_{ij}' = 0.14$, $d_{ij}' = 0.15$, $(i, j = 1, ..., 6, i \neq j)$, $\kappa = 600$, $s = 100$, $D_1 = 0.0091$, $D_2 = 0.0084$, $D_3 = 0.0088$, $D_4 = 0.0096$, $D_5 = 0.0082$, $D_6 = 0.0093$. In (**G–H**): $a_i = 0.1$, $a_i' = 0.14$, $k_i = 0.2$, $w_i = 0.18$, $d_i = 2.8$, $d_i' = 0.02$, $a_{ij}' = 0.14$, $d_{ij}' = 0.8$, $D_i = \mathcal{N}(1, 0.1) \times 0.005$, $(i, j = 1, ..., 20, i \neq j)$, $\kappa = 1000$, $s = 500$.

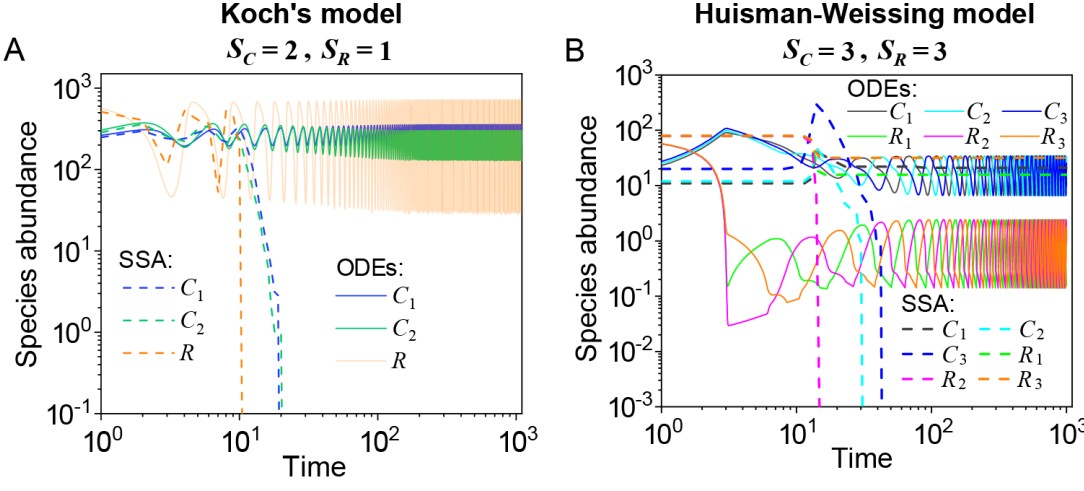

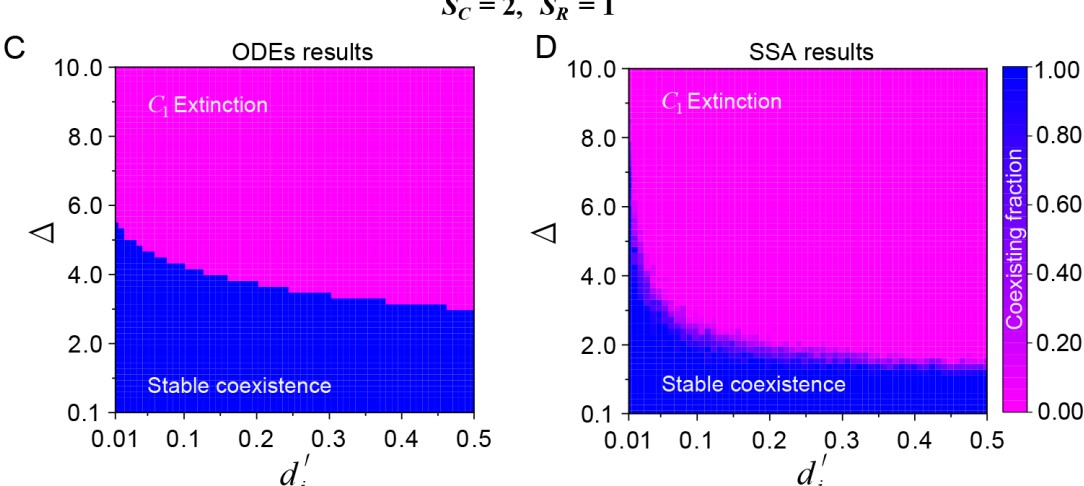

**Appendix 1—figure 6.** The influence of stochasticity on species coexistence. (**A, B**) Stochasticity jeopardizes species coexistence. Koch's model (***Koch, 1974***) and Huisman-Weissing model (***Huisman and Weissing, 1999***) were simulated with SSA using the same parameter settings as their deterministic model. Nevertheless, both cases of oscillating coexistence are vulnerable to stochasticity. See (***Koch, 1974***) and (***Huisman and Weissing, 1999***) for the parameters. (**C, D**) Phase diagrams in the scenario involving chasing pairs and intraspecific interference. Here, $S_C = 2$ and $S_R = 1$. $D_i (i = 1, 2)$ is the only parameter varying with the consumer species (with $D_1 > D_2$), and $\Delta \equiv (D_1 - D_2)/D_2$ represents the competitive difference between the two species. (**C**) The ODEs results. (**D**) The SSA results (with the same parameter region as (**C**)). The species' coexisting fraction in each pixel was calculated from 16 random repeats. (**C**) and (**D**) were calculated from ***Equations 1, 2 and 4***. In (**C, D**): $a_i = 0.1$, $a_i' = 0.125$, $d_i = 0.5$, $w_i = 0.1$, $k_i = 0.1$, $(i = 1, 2)$, $\kappa = 100$, $\zeta = 5$, $D_2 = 0.0014$.

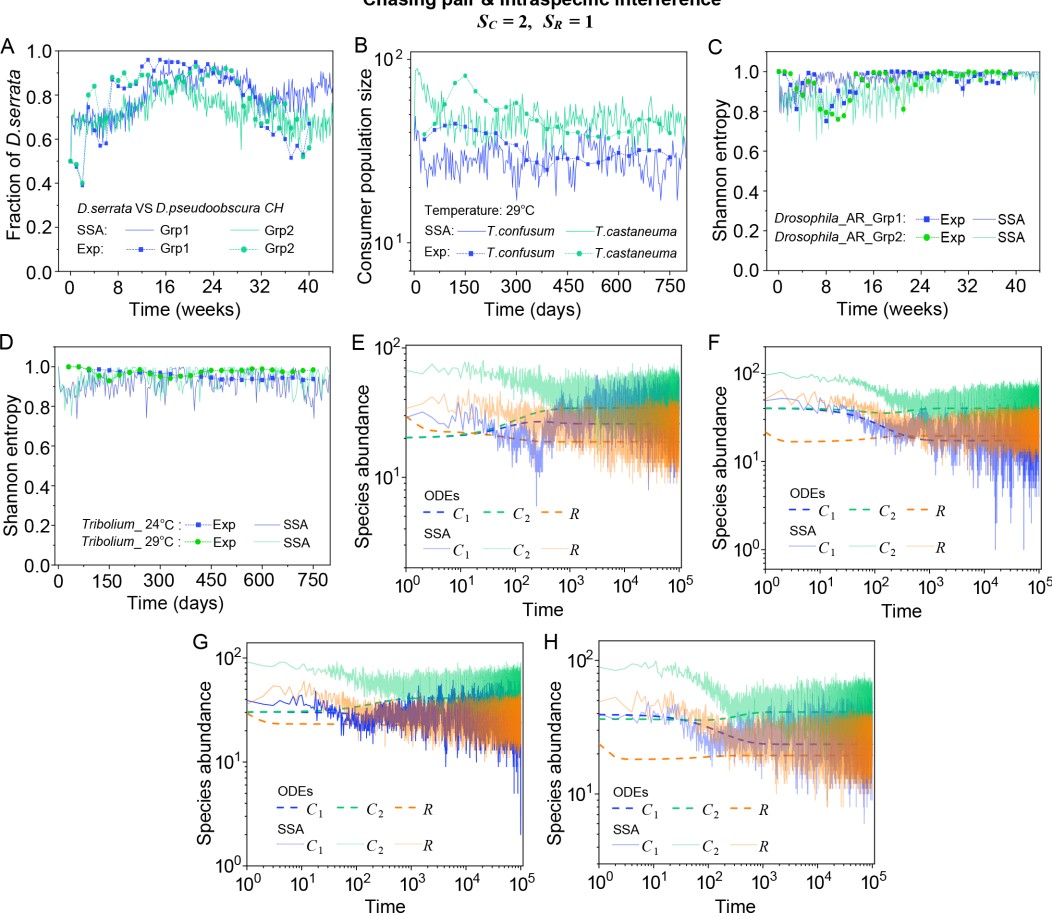

**Appendix 1—figure 7.** A model of intraspecific predator interference explains two classical experiments that invalidate CEP. (**A**) In Ayala's experiment (*Ayala, 1969*), two *Drosophila* species (consumers) coexisted for 40 weeks with the same type of abiotic resources within a laboratory bottle. The time averages ($\bar{C}_i$) and standard deviation ($\delta C_i$) of the species' relative abundances for the experimental data or SSA results are: $\overline{{}^{(R)}C}{}^{\text{Exp(SSA)}}_{D.serrata\_CH\_Grp1} = 0.77\,(0.80)$, $\delta^{(R)}C^{\text{Exp(SSA)}}_{D.serrata\_CH\_Grp1} = 0.17\,(0.08)$, $\overline{{}^{(R)}C}{}^{\text{Exp(SSA)}}_{D.serrata\_CH\_Grp2} = 0.78\,(0.73)$, $\delta^{(R)}C^{\text{Exp(SSA)}}_{D.serrata\_CH\_Grp2} = 0.14\,(0.07)$, where the superscript '(R)' represents relative abundances. (**B**) In Park's experiment (*Park, 1954*), two *Tribolium* species coexisted for 750 days with the same food (flour). The time averages ($\bar{C}_i$) and standard deviations ($\delta C_i$) of the species' abundances are: $\overline{C}{}^{\text{Exp(SSA)}}_{T.confusum\_29°C} = 33.4\,(28.8)$, $\delta C^{\text{Exp(SSA)}}_{T.confusum\_29°C} = 6.0\,(5.4)$, $\overline{C}{}^{\text{Exp(SSA)}}_{T.castaneuma\_29°C} = 48.8\,(47.7)$, $\delta C^{\text{Exp(SSA)}}_{T.castaneuma\_29°C} = 11.9\,(9.9)$. (**A, B**) The solid icons represent the experimental data, which are connected by the dotted lines for the sake of visibility. The solid lines stand for the SSA simulation results. (**C, D**) The Shannon entropies of each time point for the experimental or model-simulated communities shown in (**B**) and ***Figure 2D and E***. Here, we calculated the Shannon entropies with $H(t) = -\sum_{i=1}^{S_C} P_i(t)\log_2\left(P_i(t)\right)$, where $P_i(t)$ is the probability that a consumer individual belongs to species $C_i$ at the time stamp of $t$. The time averages ($\bar{H}$) and standard deviations ($\delta H$) of the Shannon entropies are: $\bar{H}^{Drosophila\_AR\_Grp1}_{\text{Exp(SSA)}} = 0.95\,(0.97)$, $\delta H^{Drosophila\_AR\_Grp1}_{\text{Exp(SSA)}} = 0.06\,(0.04)$, $\bar{H}^{Drosophila\_AR\_Grp2}_{\text{Exp(SSA)}} = 0.94\,(0.92)$, $\delta H^{Drosophila\_AR\_Grp2}_{\text{Exp(SSA)}} = 0.07\,(0.07)$, $\bar{H}^{Tribolium\_24°C}_{\text{Exp(SSA)}} = 0.96\,(0.92)$, $\delta H^{Tribolium\_24°C}_{\text{Exp(SSA)}} = 0.02\,(0.05)$, $\bar{H}^{Tribolium\_29°C}_{\text{Exp(SSA)}} = 0.97\,(0.94)$, $\delta H^{Tribolium\_29°C}_{\text{Exp(SSA)}} = 0.02\,(0.05)$. (**E–H**) Time courses of the species abundances in the scenario involving chasing pairs and intraspecific interference. The time series in (**E–H**) correspond to the long-term version of that shown in ***Figure 2D***, ***Appendix 1—figure 7A***, ***Figure 2E***, ***Appendix 1—figure 7B***, respectively. (**A, B, F–I**) were simulated from ***Equations 1, 2 and 4***. In (**A**): $a_i = 0.3$, $a'_i = 0.33$, $w_i = 0.018$, $k_i = 4.8$, $d'_i = 5$, $d_i = 5.5$, $(i = 1, 2)$, $D_1 = 0.0132$, $D_2 = 0.010$, $\zeta = 35$, $\kappa = 10000$. In (**B**): $a_i = 0.3$, $a'_i = 0.36$, $w_i = 0.02$, $k_i = 4.5$, $d'_i = 4$, $d_i = 4.5$, $(i = 1, 2)$, $D_1 = 0.0122$, $D_2 = 0.010$, $\zeta = 35$, $\kappa = 10000$. In(A-B): $\tau = 0.4$ Day (see Appendix 7).

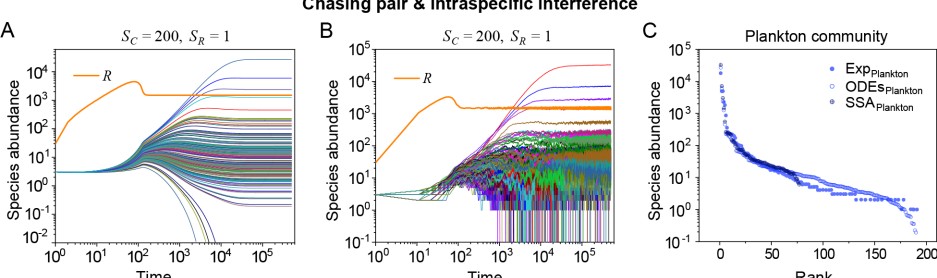

**Appendix 1—figure 8.** A model of intraspecific interference semi-quantitatively illustrates the rank-abundance curve of a plankton community ($S_C \gg S_R$). (**A, B**) Intraspecific interference enables a wide range of consumers species to coexist with one type of resources. (**A**) Time courses of the species abundances simulated with ODEs. (**B**) Time series of the species abundances simulated with SSA (with theh same as parameter settings as (**A**)). (**C**) The rank-abundance curve of a plankton community. The solid dots represent the experimental data (marked with 'Exp') reported in a recent study (**Ser-Giacomi et al., 2018**) (TARA_139.SUR.180.2000.DNA), while the hollow dots and those with '+' center are the ODEs and SSA results constructed from timestamp $t = 5.0 \times 10^5$ in the time series (see (**A**) and (**B**)), respectively. The Shannon entropies of the experimental data and simulation results for the plankton community are: $H_{\text{Exp(ODEs,SSA)}}^{\text{Plankton}} = 2.85(2.18, 2.00)$. In the model settings, $S_C = 200$ and $S_R = 1$. $D_i(i = 1, \ldots, S_C)$ is the only parameter that varies with the consumer species, which was randomly drawn from a Gaussian distribution $\mathcal{N}(\mu, \sigma)$. Here, $\mu$ and $\sigma$ are the mean and standard deviation of the distribution. The numerical results in (**A–C**) were simulated from **Equations 1, 2 and 4**. In (**A–C**): $a_i = 0.1$, $a_i' = 0.125$, $d_i = 0.2$, $d_i' = 0.5$, $w_i = 0.2$, $k_i = 0.1$, $D_i = 0.008 \times \mathcal{N}(1, 0.38)$, $(i = 1, \cdots, 200)$, $\kappa = 10^5$, $\zeta = 150$. $a_i = 0.1, a_i' = 0.125, d_i' = 0.5, d_i = 0.2, w_i = 0.2, k_i = 0.1, D_i = \mathcal{N}(1, 0.38) \times 0.008$

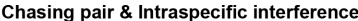

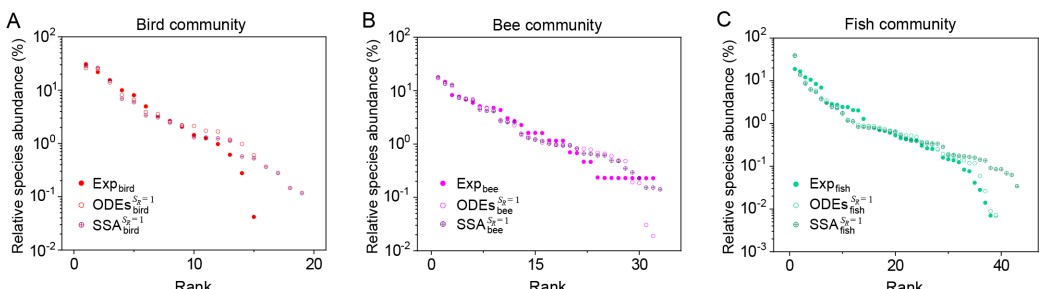

**Appendix 1—figure 9.** A model of intraspecific interference illustrates the rank-abundance curves across different ecological communities ($S_C \gg S_R$). The solid dots represent the experimental data (marked with 'Exp') reported in existing studies (**Hubbell, 2001**; **Holmes et al., 1986**; **Cody and Smallwood, 1996**), while the hollow dots and those with '+' center are the ODEs and SSA results constructed from timestamp $t = 1.0 \times 10^5$ in the time series (see **Appendix 1—figure 10A–G**), respectively. In the model settings, $S_R = 1$, $S_C = 20$ (in (**A**)), 35 (in (**B**)) or 45 (in (**C**)). $D_i$ is the only parameter varying with the consumer species, which was randomly drawn from a Gaussian distribution. The Shannon entropies of the experimental data and simulation results for each ecological community are: $H_{\text{Exp(ODEs,SSA)}}^{\text{bird}} = 2.98(3.06, 2.98)$, $H_{\text{Exp(ODEs,SSA)}}^{\text{bee}} = 4.04(4.02, 4.02)$, $H_{\text{Exp(ODEs,SSA)}}^{\text{fish}} = 3.78(3.40, 3.41)$. In the Kolmogorov-Smirnov (K-S) test, the probabilities ($p$-values) that the simulation results and the corresponding experimental data come from identical distributions are: $p_{\text{ODEs}}^{\text{bird}} = 0.89$, $p_{\text{SSA}}^{\text{bird}} = 0.88$, $p_{\text{ODEs}}^{\text{bee}} = 0.47$, $p_{\text{SSA}}^{\text{bee}} = 0.75$, $p_{\text{SSA}}^{\text{fish}} = 0.77$. With a significance threshold of 0.05, none of the $p$-values suggest there exists a statistically significant difference. The numerical results in (**A–C**) were simulated from **Equations 1, 2 and 4**. In (**A**): $a_i = 0.1$, $a_i' = 0.125$, $d_i = 0.5$, $d_i' = 0.6$, $w_i = 0.22$, $k_i = 0.1$, $D_i = 0.016 \times \mathcal{N}(1, 0.35)$, $(i = 1, \cdots, 20)$, $\zeta = 350$, $\kappa = 10^6$. In (**B**): $a_i = 0.1$, $a_i' = 0.125$, $d_i = 0.5$, $d_i' = 0.6$, $w_i = 0.22$, $k_i = 0.1$, $D_i = 0.012 \times \mathcal{N}(1, 0.35)$, $(i = 1, \cdots, 35)$, $\zeta = 350$, $\kappa = 10^6$. In (**C**): $a_i = 0.1$, $a_i' = 0.14$, $d_i = 0.5$, $d_i' = 0.5$, $w_i = 0.2$, $k_i = 0.1$, $D_i = 0.015 \times \mathcal{N}(1, 0.32)$, $(i = 1, \cdots, 45)$, $\zeta = 550$, $\kappa = 10^6$.

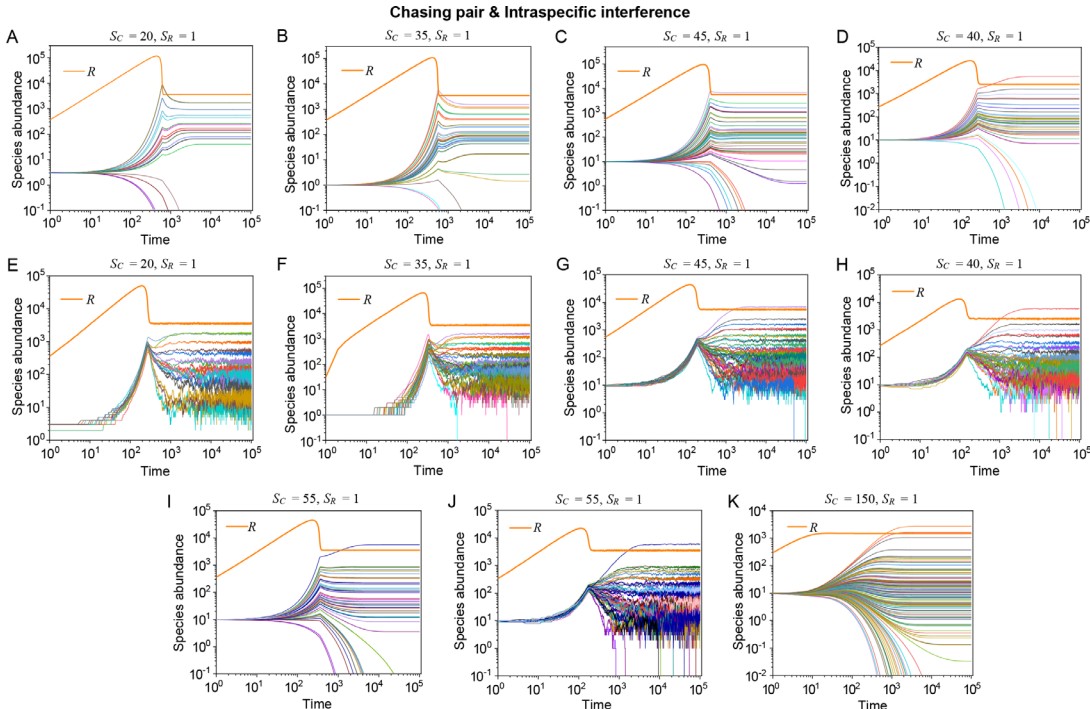

**Appendix 1—figure 10.** Time courses of the species abundances in the scenario involving chasing pairs and intraspecific interference. The time series in (**A, E**), (**B, F**), (**C, G**), (**D, H**), (**I, J**) and (**K**) correspond to that shown in *Appendix 1—figure 9A–C*, *Figure 3D* (bat), *Figure 3D* (lizard) and *Figure 3C* (butterfly), respectively.

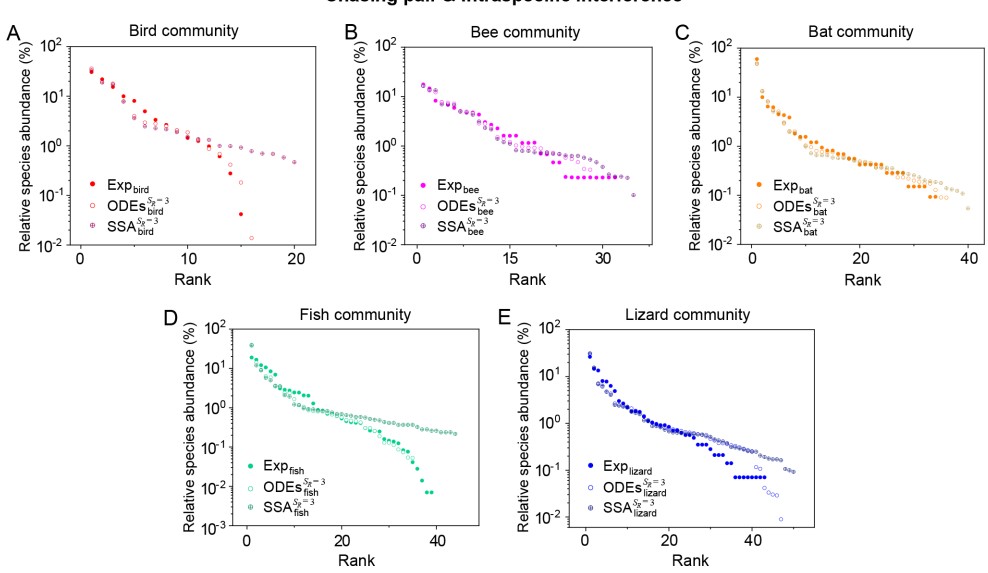

**Appendix 1—figure 11.** A model of intraspecific interference illustrates the rank-abundance curves across different ecological communities ($S_C \gg S_R$). The solid dots represent the experimental data (marked with 'Exp') reported in existing studies (*Hubbell, 2001*; *Holmes et al., 1986*; *Cody and Smallwood, 1996*; *Clarke et al., 2005*), while the hollow dots and those with '+' center are the ODEs and SSA results constructed from timestamp $t = 1.0 \times 10^5$ in the time series (see *Appendix 1—figure 13*), respectively. In the model settings, $S_R = 3$, $S_C = 20$ (in (**A**)), 35 (in (**B**)), 40 (in (**C**)), 45 (in (**D**)) or 50 (in (**E**)). $D_i$ is the only parameter varying with the consumer species, which was randomly drawn from a Gaussian distribution. The Shannon entropies of the experimental data and simulation results for each ecological community are: $H_{\text{Exp(ODEs,SSA)}}^{\text{Bird}} = 2.98(2.98, 3.26)$, $H_{\text{Exp(ODEs,SSA)}}^{\text{Bee}} = 4.04(4.34, 4.35)$, $H_{\text{Exp(ODEs,SSA)}}^{\text{Bat}} = 3.00(3.00, 3.00)$, $H_{\text{Exp(ODEs,SSA)}}^{\text{Fish}} = 3.78(3.28, 3.64)$, *Appendix 1—figure 11 continued on next page*

*Appendix 1—figure 11 continued*

$H_{\text{Exp(ODEs,SSA)}}^{\text{lizard}} = 4.05(3.94, 3.94)$. In the K-S test, the $p$-values that the simulation results and the corresponding experimental data come from identical distributions are: $p_{\text{ODEs}}^{\text{Bird}} = 0.59$, $p_{\text{SSA}}^{\text{Bird}} = 0.43$, $p_{\text{ODEs}}^{\text{Bee}} = 0.47$, $p_{\text{SSA}}^{\text{Bee}} = 0.33$, $p_{\text{ODEs}}^{\text{Bat}} = 0.42$, $p_{\text{SSA}}^{\text{Bat}} = 0.27$, $p_{\text{ODEs}}^{\text{Fish}} = 0.22$, $p_{\text{SSA}}^{\text{Fish}} = 0.06$, $p_{\text{ODEs}}^{\text{lizard}} = 0.56$, $p_{\text{SSA}}^{\text{lizard}} = 0.36$. With a significance threshold of 0.05, none of the $p$-values suggest there exists a statistically significant difference. The numerical results in (**A–E**) were simulated from ***Equations 1, 2 and 4***. In (**A–E**): $a_{il} = 0.1$, $a_i' = 0.125$, $d_{il} = 0.5$. In (**A**): $d_i' = 0.3$, $w_{il} = 0.2$, $k_{il} = 0.12$, $\kappa_1 = 8 \times 10^4$, $\kappa_2 = 5 \times 10^4$, $\kappa_3 = 3 \times 10^4$, $D_i = 0.021 \times \mathcal{N}(1, 0.28)$, $(i = 1, \cdots, 20, l = 1, 2, 3)$, $\zeta_1 = 180$, $\zeta_2 = 160$, $\zeta_3 = 140$. In (**B**): $d_i' = 0.6$, $w_{il} = 0.2$, $k_{il} = 0.12$, $\kappa_1 = 8 \times 10^4$, $\kappa_2 = 5 \times 10^4$, $\kappa_3 = 3 \times 10^4$, $D_i = 0.017 \times \mathcal{N}(1, 0.3)$, $(i = 1, \cdots, 35, l = 1, 2, 3)$, $\zeta_1 = 180$, $\zeta_2 = 160$, $\zeta_3 = 110$. In (**C**): $d_i' = 0.4$, $w_{il} = 0.3$, $k_{il} = 0.12$, $\kappa_1 = 10^5$, $\kappa_2 = 5 \times 10^4$, $\kappa_3 = 3 \times 10^4$, $D_i = 0.023 \times \mathcal{N}(1, 0.34)$, $(i = 1, \cdots, 40, l = 1, 2, 3)$, $\zeta_1 = 180$, $\zeta_2 = 120$, $\zeta_3 = 40$. In (**D**): $d_i' = 0.3$, $w_{il} = 0.3$, $k_{il} = 0.12$, $\kappa_1 = 8 \times 10^4$, $\kappa_2 = 5 \times 10^4$, $\kappa_3 = 3 \times 10^4$, $D_i = 0.027 \times \mathcal{N}(1, 0.32)$, $(i = 1, \cdots, 45, l = 1, 2, 3)$, $\zeta_1 = 80$, $\zeta_2 = 60$, $\zeta_3 = 40$. In (**E**): $d_i' = 0.3$, $w_{il} = 0.3$, $k_{il} = 0.2$, $\kappa_1 = 3 \times 10^5$, $\kappa_2 = 10^5$, $\kappa_3 = 3 \times 10^4$, $D_i = 0.034 \times \mathcal{N}(1, 0.34)$, $(i = 1, \cdots, 50, l = 1, 2, 3)$, $\zeta_1 = 380$, $\zeta_2 = 260$, $\zeta_3 = 140$.

**Chasing pair & Intraspecific interference**

**Appendix 1—figure 12.** A model of intraspecific interference illustrates the rank-abundance curves across different plankton communities ($S_C \gg S_R$). The solid dots represent the experimental data (marked with 'Exp') reported in a recent study (***Fuhrman et al., 2008***), while the hollow dots and those with '+' center are the ODEs and SSA results constructed from timestamp $t = 1.0 \times 10^5$ in the time series (see ***Appendix 1—figure 13***), respectively. The plankton community data were obtained separately from the Norwegian Sea (NS) and Pacific Station (PS). In the model settings, $S_R = 1$ (in (**B, C**)), 3 (in (**A**)); $S_C = 50$ (in (**A, C**)), 150 (in (**B**)). $D_i$ is the only parameter varying with the consumer species, which was randomly drawn from a Gaussian distribution. The Shannon entropies of the experimental data and simulation results for each plankton community are: $H_{\text{Exp(ODEs,SSA)}}^{\text{plankton(NS)}} = 4.67(4.85, 4.90)$ for $S_R = 3$, $H_{\text{Exp(ODEs,SSA)}}^{\text{plankton(NS)}} = 4.67(4.74, 4.64)$ for $S_R = 1$. In the K-S test, the $p$-values that the simulation results and the corresponding experimental data come from identical distributions are: $p_{\text{ODEs}}^{\text{plankton(NS)}} = 0.31$, $p_{\text{SSA}}^{\text{plankton(NS)}} = 0.14$ for $S_R = 3$, $p_{\text{ODEs}}^{\text{plankton(NS)}} = 0.46$, $p_{\text{SSA}}^{\text{plankton(NS)}} = 0.37$ for $S_R = 1$. With a significance threshold of 0.05, none of the $p$-values suggest there exists a statistically significant difference. The numerical results in (**A–C**) were simulated from ***Equations 1, 2 and 4***. In (**A**): $a_{il} = 0.1$, $a_i' = 0.125$, $d_{il} = 0.5$, $d_i' = 0.2$, $w_{il} = 0.3$, $k_{il} = 0.2$, $\kappa_1 = 8 \times 10^4$, $\kappa_2 = 5 \times 10^4$, $\kappa_3 = 3 \times 10^4$, $\zeta_1 = 280$, $\zeta_2 = 200$, $\zeta_3 = 150$, $D_i = 0.035 \times \mathcal{N}(1, 0.25)$, $(i = 1, \cdots, 50, l = 1, 2, 3)$. In (**B**): $a_i = 0.1$, $a_i' = 0.125$, $d_i = 0.3$, $d_i' = 0.3$, $w_i = 0.3$, $k_i = 0.2$, $D_i = 0.025 \times \mathcal{N}(1, 0.25)$, $(i = 1, \cdots, 150, l = 1, 2, 3)$, $\zeta = 350$, $\kappa = 10^4$. In (**C**): $a_i = 0.1$, $a_i' = 0.125$, $d_i = 0.3$, $d_i' = 0.3$, $w_i = 0.3$, $k_i = 0.2$, $\zeta = 350$, $\kappa = 10^4$, $D_i = 0.03 \times \mathcal{N}(1, 0.27)$, $(i = 1, \cdots, S_C)$.

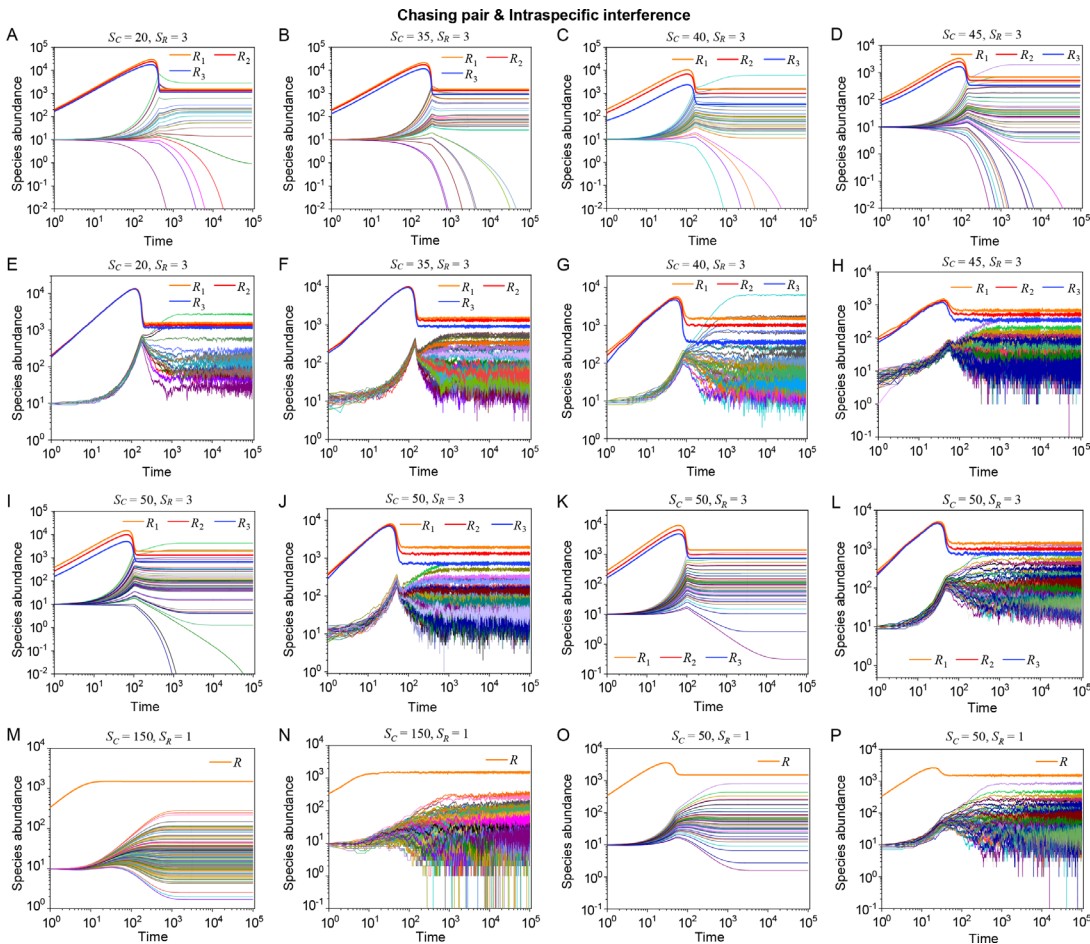

**Appendix 1—figure 13.** Time courses of the species abundances in the scenario involving chasing pairs and intraspecific interference. The time series in (**A, E**), (**B, F**), (**C, G**), (**D, H**), (**I, J**), (**K, L**), (**M, N**) and (**O, P**) correspond to that shown in *Appendix 1—figure 11A–E* and *Appendix 1—figure 12A–C*, respectively.

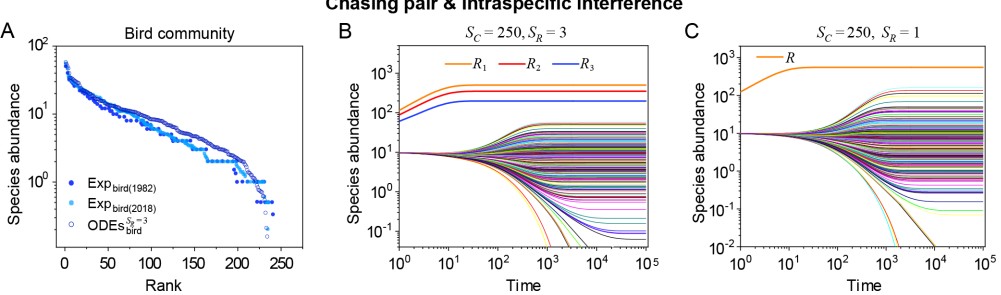

**Appendix 1—figure 14.** A model of intraspecific interference illustrates the rank-abundance curve of a bird community ($S_C \gg S_R$). (**A**) The rank-abundance curve. The solid dots represent the bird community data (marked with 'Exp') collected longitudinally within the same Amazonian region in 1982 (blue) and 2018 (cyan) (*Terborgh et al., 1990*; *Martínez et al., 2023*). The hollow dots are the ODEs results constructed from timestamp $t = 1.0 \times 10^5$ in the time series (see (**B**)). (**B**) Time courses of the species abundances simulated with ODEs. In the model settings, $S_R = 3$ and $S_C = 250$ is the only parameter varying with the consumer species, which was randomly drawn from a Gaussian distribution. The Shannon entropies of the experimental data and simulation results for the bird community are $H_{\text{Exp(ODEs)}}^{\text{bird(1982/2018)}} = 5.67/6.63(7.31)$. In the K-S test, the $p$-values that the simulation results and the corresponding experimental data come from identical distributions are: $p_{\text{ODEs}}^{\text{bird(1982)}} = 0.28$, $p_{\text{ODEs}}^{\text{bird(2018)}} = 0.46$. With a significance threshold of 0.05, none of the $p$-values suggest there exists a statistically significant difference. (**C**) Time courses of the species abundances simulated with ODEs
*Appendix 1—figure 14 continued on next page*

*Appendix 1—figure 14 continued*

corresponding to *Figure 3C* (bird), and the simulation parameters are the same as *Figure 3C* (bird). The numerical results in (**A–C**) were simulated from *Equations 1, 2 and 4*. In (**A, B**): $a_{il} = 0.1$, $a'_i = 0.125$, $d_{il} = 0.5$, $d'_i = 0.6$, $w_{il} = 0.3$, $k_{il} = 0.2$, $D_i = 0.032 \times \mathcal{N}(1, 0.17)$, $(i = 1, \cdots, 250, l = 1, 2, 3)$, $\kappa_1 = 5 \times 10^4$, $\kappa_2 = 3 \times 10^4$, $\kappa_3 = 10^4$, $\zeta_1 = 100$, $\zeta_2 = 70$, $\zeta_3 = 40$.

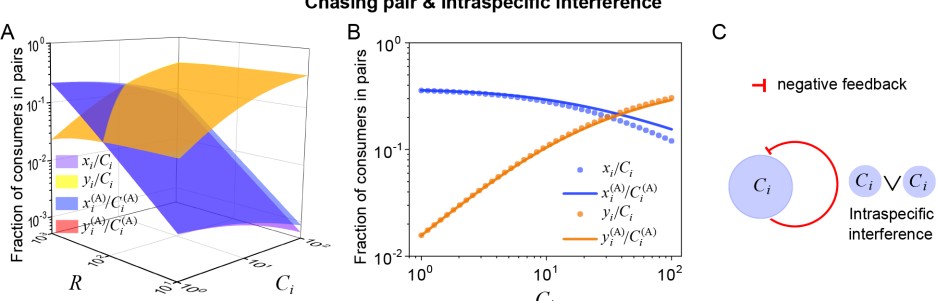

**Appendix 1—figure 15.** Intraspecific interference results in an underlying negative feedback loop and thus promotes biodiversity. (**A, B**) The fraction of consumer individuals engaged in pairwise encounter. Here, $S_C = 40$ and $S_R = 1$. $x_i$ represents $C_i^{(P)} \vee R^{(P)}$, and $y_i$ represents $C_i^{(P)} \vee C_i^{(P)}$ and $y_i/C_i$ stand for the fractions of consumer individuals within a chasing pair and an interference pair, respectively. The numerical results were calculated from *Equation S55*, while the analytical solutions (marked with superscript '(A)') were calculated from *Equation S59*. The orange surface in (**A**) is an overlap of the red and yellow surfaces. (**C**) The formation of intraspecific interference results in a self-inhibiting negative feedback loop. In (**A, B**): $a_i = 0.0015$, $a'_i = 0.0021$, $d_i = 0.1$, $d'_i = 0.05$, $k_i = 5$. In (**B**): $R = 2000$.

## Appendix 2

### The classical proof of Competitive Exclusion Principle (CEP)

In the 1960s, MacArthur (*MacArthur and Levins, 1964*) and Levin (*Levin, 1970*) put forward the classical mathematical proof of CEP. We rephrase their idea in the simple case of $S_C = 2$ and $S_R = 1$, that is two consumer species $C_1$ and $C_2$ competing for one resource species $R$. In practice, this proof can be generalized into higher dimensions with several consumer and resource species. The population dynamics of the system can be described as follows:

$$\begin{cases} \dot{C}_i = C_i(f_i(R) - D_i), \ i = 1, 2; \\ \dot{R} = g(R, C_1, C_2). \end{cases} \tag{S1}$$

Here, $C_i$ and $R$ represent the population abundances of consumers and resources, respectively, while the functional forms of $f_i(R)$ and $g(R, C_1, C_2)$ are unspecific. $D_i$ stands for the mortality rate of the species $C_i$. If all consumer species can coexist at steady state, then $f_i(R)/D_i = 1$ ($i = 1, 2$). In a 2-D representation, this requires that three lines $f_i(R)/D_i = 1$ and $y = 1$ share a common point, which is commonly impossible unless the model parameters satisfy special constraint (sets of Lebesgue measure zero). In a 3-D representation, the two planes corresponding to $f_i(R)/D_i = 1$ ($i = 1, 2$) are parallel, and hence do not share a common point (see *Wang and Liu, 2020* for details).

## Appendix 3

## Comparison of the functional response with Beddington-DeAngelis (B-D) model

### A B-D model

In 1975, Beddington proposed a mathematical model (*Beddington, 1975*) to describe the influence of predator interference on the functional response with hand-waving derivations. In the same year, DeAngelis and his colleagues considered a related question and put forward a similar model (*DeAngelis et al., 1975*). Essentially, both models are phenomenological, and they were called B-D model in the subsequent studies. In practice, the B-D model can be extended into scenarios involving different types of pairwise encounters with Beddington's modelling method. In this section, we systematically compare the functional response in B-D model with that of our mechanistic model in all the relevant scenarios.

Recalling Beddington's analysis, the model (*Beddington, 1975*) consists of one consumer species $C$ and one resource species $R$ ($S_C = 1$, $S_R = 1$). In a well-mixed system, an individual consumer meets a resource with rate $a$, while encounters another consumer with rate $a'$. There are two other phenomenological parameters in this model, namely, the handling time $t_h$ and the wasting time $t_w$. Both can be determined by specifying the scenario and using statistical physics modeling analysis. In fact, Beddington analyzed the searching efficiency $\Xi_{\text{B-D}}$ rather than the functional response $\mathcal{F}_{\text{B-D}}$, yet both can be reciprocally derived with $\Xi_{\text{B-D}} \equiv \mathcal{F}_{\text{B-D}}/R$. Here $R$ stands for the population abundance of the resources, and the specific form of $\Xi_{\text{B-D}}$ is (*Beddington, 1975*):

$$\Xi_{\text{B-D}}(R, C) = \frac{a}{1 + at_h R + a't_w C'}, \tag{S2}$$

where $C' = C - 1$, and $C$ stands for the population abundance of the consumes. Generally, $C \gg 1$, and thus $C' \approx C$.

### B Scenario involving only chasing pairs

Here, we consider the scenario involving only chasing pair for the simple case with one consumer species $C$ and one resource species $R$ ($S_C = 1, S_R = 1$). When an individual consumer is chasing a resource, they form a chasing pair:

$$C^{(\text{F})} + R^{(\text{F})} \underset{d}{\overset{a}{\rightleftharpoons}} C^{(\text{P})} \vee R^{(\text{P})} \xrightarrow{k} C^{(\text{F})}(+),$$

where the superscript '(F)' stands for populations that are freely wandering, and '(+)' signifies gaining biomass (we count $C^{(\text{F})}(+)$ as $C^{(\text{F})}$). $C^{(\text{P})} \vee R^{(\text{P})}$ represents chasing pair (where '(P)' signifies pair), denoted as $x$. $a$, $d$ and $k$ stand for encounter rate, escape rate and capture rate, respectively. Hence, the total number of consumers and resources are $C \equiv C^{(\text{F})} + x$ and $R \equiv R^{(\text{F})} + x$. Then, the population dynamics of the system follows:

$$\begin{cases} \dot{x} = aC^{(\text{F})}R^{(\text{F})} - (k + d)x, \\ \dot{C} = wkx - DC, \\ \dot{R} = g(R, x, C). \end{cases} \tag{S3}$$

Here, the functional form of $g(R, x, C)$ is unspecific, while $D$ and $w$ represent the mortality rate of the consumer species and biomass conversion ratio (*Wang and Liu, 2020*), respectively. Since consumption process is generically much faster than the birth/death process, in deriving the functional response, the consumption process is supposed to be in fast equilibrium (i.e. $\dot{x} = 0$). Then, we can solve for $x$ with:

$$x^2 - (R + C + K)x + RC = 0, \tag{S4}$$

where $K = \frac{k+d}{a}$, and then,

$$x = \frac{2RC}{(R + C + K)} \frac{1}{(1 + \sqrt{1 - \frac{4RC}{(R + C + K)^2}})}. \tag{S5}$$

By definition, the functional response and searching efficiency are:

$$\mathcal{F}_{\text{CP}}(R, C) = \frac{kx}{C}, \tag{S6a}$$

$$\Xi_{\text{CP}}(R, C) = \frac{kx}{RC}. \tag{S6b}$$

Hence, we obtain the functional response and searching efficiency in this chasing-pair scenario:

$$\mathcal{F}_{\text{CP}}(R, C)_{(1)} = k\frac{(R + C + K)}{2C} \left(1 - \sqrt{1 - \frac{4RC}{(R + C + K)^2}}\right), \tag{S7a}$$

$$\Xi_{\text{CP}}(R, C)_{(1)} = k\frac{(R + C + K)}{2RC} \left(1 - \sqrt{1 - \frac{4RC}{(R + C + K)^2}}\right). \tag{S7b}$$

Since $\frac{4RC}{(R+C+K)^2} < 4\frac{C}{R} \ll 1$, using first-order approximations in **Equation S7a**, **Equation S7b**, we obtain $\sqrt{1 - \frac{4RC}{(R+C+K)^2}} \approx 1 - \frac{2RC}{(R+C+K)^2}$. Then the functional response and searching efficiency are:

$$\mathcal{F}_{\text{CP}}(R, C)_{(2)} = k\frac{R}{R + C + K}, \tag{S8a}$$

$$\Xi_{\text{CP}}(R, C)_{(2)} = \frac{k}{R + C + K}. \tag{S8b}$$

Evidently, there is no predator interference within the chasing-pair scenario, yet the functional response form is identical to the B-D model involving intraspecific interference (see **Equation S2**). Meanwhile, using first-order approximations in the denominator of **Equation S5**, we have $x \approx \frac{RC}{(R+C+K) - \frac{RC}{(R+C+K)}}$. Hence,

$$\mathcal{F}_{\text{CP}}(R, C)_{(3)} = k\frac{R}{(R + C + K) - \frac{RC}{(R + C + K)}}, \tag{S9a}$$

$$\Xi_{\text{CP}}(R, C)_{(3)} = \frac{k}{(R + C + K) - \frac{RC}{(R + C + K)}}. \tag{S9b}$$

In the case that $R \gg C$, then $R \gg C > x = R - R^{(\text{F})}$. By applying $R \approx R^{(\text{F})}$ in **Equation S3**, we obtain $x \approx \frac{RC}{R+K}$. Then,

$$\mathcal{F}_{\text{CP}}(R, C)_{(4)} = k\frac{R}{R + K}, \tag{S10a}$$

$$\Xi_{\text{CP}}(R, C)_{(4)} = \frac{k}{R + K}. \tag{S10b}$$

To compare these functional responses with that of the B-D model, we determine the parameters $t_h$ and $t_w$ in the B-D model by calculating their ensemble average values in a stochastic framework. Using the properties of waiting time distribution in the Poisson process, we obtain $\langle t_h \rangle = \frac{1}{k}$ and $\langle t_w \rangle = \frac{1}{d'}$ (in the chasing-pair scenario, $a' = 0$). By substituting these calculations into **Equation S2**, we have:

$$\Xi_{\text{CP}}^{\text{B-D}}(R, C) = \frac{a}{1 + Ra/k} = \frac{k}{k/a + R}, \tag{S11a}$$

$$\mathcal{F}_{\text{CP}}^{\text{B-D}}(R, C) = \frac{kR}{k/a + R}. \tag{S11b}$$

In the special case with $d = 0$ and $R \gg C$, the B-D model is consistent with our mechanistic model: $\Xi_{\text{B-D}}(R, C) = \Xi_{\text{CP}}(R, C)_{(4)}$. Outside the special region, however, the discrepancy can be considerably large (see *Appendix 1—figure 2A–C* for the comparison).

## C Scenario involving chasing pairs and intraspecific interference

Here, we consider the scenario with additional involvement of intraspecific interference in the simple case of $S_C = 1$ and $S_R = 1$:

$$C^{(\text{F})} + R^{(\text{F})} \underset{d}{\overset{a}{\rightleftharpoons}} C^{(\text{P})} \vee R^{(\text{P})} \overset{k}{\to} C^{(\text{F})}(+),$$

$$C^{(\text{F})} + C^{(\text{F})} \underset{d'}{\overset{a'}{\rightleftharpoons}} C^{(\text{P})} \vee C^{(\text{P})}.$$

Here, $C^{(\text{P})} \vee C^{(\text{P})}$ stands for the intraspecific predator interference pair, denoted as $y$; $a'$ and $d'$ represent the encounter rate and separation rate of the interference pair, respectively. Then, the total population of consumers and resources are $C \equiv C^{(\text{F})} + x + 2y$ and $R \equiv R^{(\text{F})} + x$. Hence the population dynamics of the consumers and resources can be described as follows:

$$\begin{cases} \dot{x} = aC^{(\text{F})}R^{(\text{F})} - (k + d)x, \\ \dot{y} = a'[C^{(\text{F})}]^2 - d'y, \\ \dot{C} = wkx - DC, \\ \dot{R} = g(R, x, C). \end{cases} \tag{S12}$$

The consumption process and interference process are supposed to be in fast equilibrium (i.e., $\dot{x} = 0, \dot{y} = 0$), then we can solve for $x$ with:

$$x^3 + \phi_2 x^2 + \phi_1 x + \phi_0 = 0, \tag{S13}$$

where $\phi_0 = -CR^2$, $\phi_1 = 2CR + KR + R^2$, $\phi_2 = 2\beta K^2 - K - C - 2R$ with $\beta = a'/d'$. The discriminant of *Equation S13* (denoted as $\Lambda$) is:

$$\Lambda = -4\psi^3 - 27\varphi^2, \tag{S14}$$

with $\psi = \phi_1 - (\phi_2)^2/3$ and $\varphi = \phi_0 - \phi_1\phi_2/3 + 2(\phi_2)^3/27$. When $\Lambda < 0$, there are one real solution $x_{(1)}$ and two complex solutions $x_{(2)}, x_{(3)}$, which are:

$$x_{(1)} = \theta_1 + \theta_2 - \phi_2/3, x_{(2)} = \omega\theta_1 + \omega^2\theta_2 - \phi_2/3, x_{(3)} = \omega^2\theta_1 + \omega\theta_2 - \phi_2/3, \tag{S15}$$

where $\omega = -1/2 + i\sqrt{3}/2$ (i stands for the imaginary unit), $\theta_1 = (-\varphi/2 + \sqrt{-\Lambda/108})^{1/3}$, and $\theta_2 = (-\varphi/2 - \sqrt{-\Lambda/108})^{1/3}$. On the other hand, when $\Lambda > 0$, there are three real solutions $x_{(1)}, x_{(2)}$, and $x_{(3)}$, which are:

$$x_{(1)} = \psi'\cos\varphi' - \phi_2/3, x_{(2)} = \psi'\cos(\varphi' + \frac{2\pi}{3}) - \phi_2/3, x_{(3)} = \psi'\cos(\varphi' + \frac{4\pi}{3}) - \phi_2/3, \tag{S16}$$

where $\psi' = (-4\psi/3)^{1/2}$ and $\varphi' = \arccos(-(-\psi/3)^{-3/2}\varphi/2)/3$. Note that $x \in [0, \min(R, C)]$, then we obtain the exact feasible solution of $x$ (denoted as $x_{\text{ext}}$), and hence the functional response and searching efficiency are:

$$\mathcal{F}_{\text{intra}}(R, C)_{(1)} = \frac{kx_{\text{ext}}}{C}, \tag{S17a}$$

$$\Xi_{\text{intra}}(R, C)_{(1)} = \frac{kx_{\text{ext}}}{RC}. \tag{S17b}$$

In the case of $R \gg C$, then $R - R^{(F)} = x < C \ll R$, and thus $R^{(F)} \approx R$. Still, the consumption process is supposed to be in fast equilibrium (i.e. $\dot{x} = 0, \dot{y} = 0$), and then we obtain:

$$x \approx \frac{RC}{\sqrt{[\frac{1}{2}(K+R)]^2 + 2C\beta K^2} + \frac{1}{2}(K+R)}. \tag{S18}$$

Consequently,

$$\mathcal{F}_{\text{intra}}(R, C)_{(2)} = k \frac{R}{\sqrt{[\frac{1}{2}(K+R)]^2 + 2C\beta K^2} + \frac{1}{2}(K+R)}, \tag{S19a}$$

$$\Xi_{\text{intra}}(R, C)_{(2)} = k \frac{1}{\sqrt{[\frac{1}{2}(K+R)]^2 + 2C\beta K^2} + \frac{1}{2}(K+R)}. \tag{S19b}$$

When $\beta \ll \frac{1}{8C}$ or $8\beta C/(1 + R/K)^2 \ll 1$, using first-order approximations in the denominator of **Equation S18**, we have:

$$x \approx \frac{RC}{(K+R) + \frac{2K}{(1+R/K)}\beta C}, \tag{S20}$$

and then,

$$\mathcal{F}_{\text{intra}}(R, C)_{(3)} = k \frac{R}{(K+R) + \frac{2K}{(1+R/K)}\beta C}, \tag{S21a}$$

$$\Xi_{\text{intra}}(R, C)_{(3)} = k \frac{1}{(K+R) + \frac{2K}{(1+R/K)}\beta C}. \tag{S21b}$$

In the case that $8\beta C/(1 + R/K)^2 \gg 1$, using first-order approximations in **Equation S18**, we obtain:

$$x \approx \frac{RC}{K\sqrt{2C\beta} + \frac{(K+R)^2}{8K\sqrt{2C\beta}} + \frac{1}{2}(K+R)}, \tag{S22}$$

and thus,

$$\mathcal{F}_{\text{intra}}(R, C)_{(4)} = k \frac{R}{K\sqrt{2C\beta} + \frac{(K+R)^2}{8K\sqrt{2C\beta}} + \frac{1}{2}(K+R)}, \tag{S23a}$$

$$\Xi_{\text{intra}}(R, C)_{(4)} = k \frac{1}{K\sqrt{2C\beta} + \frac{(K+R)^2}{8K\sqrt{2C\beta}} + \frac{1}{2}(K+R)}. \tag{S23b}$$

Meanwhile, the B-D model only fits to the cases with $d = 0$. By calculating the average values of $t_h$ and $t_w$ in the stochastic framework, we have $\langle t_h \rangle = \frac{1}{k}, \langle t_w \rangle = \frac{1}{d'}$. Thus, we obtain the searching efficiency and functional response in the B-D model:

$$\Xi_{\text{intra}}^{\text{B-D}}(R, C) = \frac{a}{1 + \frac{a}{k}R + \frac{a'}{d'}C} = \frac{a}{1 + R/K \mid_{d=0} + \beta C}, \tag{S24a}$$

$$\mathcal{F}_{\text{intra}}^{\text{B-D}}(R, C) = \frac{aR}{1 + R/K \mid_{d=0} + \beta C}. \tag{S24b}$$

Overall, the searching efficiency (and the functional response) of the B-D model is quite different from either the rigorous form $\Xi_{\text{intra}}(R, C)_{(1)}$, the quasi rigorous form $\Xi_{\text{intra}}(R, C)_{(2)}$, or the more simplified forms $\Xi_{\text{intra}}(R, C)_{(3)}$ and $\Xi_{\text{intra}}(R, C)_{(4)}$ (*Appendix 1—figure 2D–F*). Still, there is a region where the discrepancies can be small, namely $d \approx 0$ and $R \gg C$ (*Appendix 1—figure 2D–F*). Intuitively, when $\beta \ll \frac{1}{8C}$ and $d = 0$, then $\Xi_{\text{intra}}(R, C)_{(3)} = \frac{a}{1 + \frac{a}{k}R + \frac{2}{(1+R/K)}\beta C}$. Consequently, if $R/K = x/C^{(F)} < 1$, then $\frac{2}{(1+R/K)} \in [1, 2]$. In this case, the difference between $\Xi_{\text{intra}}^{\text{B-D}}(R, C)$ and $\Xi_{\text{intra}}(R, C)_{(3)}$ is small.

In fact, the above analysis also applies to cases with more than one types of consumer species (i.e., for cases with $S_C > 1$).

## D Scenario involving chasing pairs and interspecific interference

Next, we consider the scenario involving chasing pairs and interspecific interference in the case of $S_C = 2$ and $S_R = 1$:

$$C_i^{(F)} + R^{(F)} \underset{d_i}{\overset{a_i}{\rightleftharpoons}} C_i^{(P)} \vee R^{(P)} \xrightarrow{k_i} C_i^{(F)}(+), \, i = 1, 2;$$

$$C_1^{(F)} + C_2^{(F)} \underset{d'_{12}}{\overset{a'_{12}}{\rightleftharpoons}} C_1^{(P)} \vee C_2^{(P)}.$$

Here $C_1^{(P)} \vee C_2^{(P)}$ stands for the interspecific interference pair, denoted as $z$; $a'_{12}$ and $d'_{12}$ represent the encounter rate and separation rate of the interference pair, respectively. Then, the total population of consumers and resources are $C_i \equiv C_i^{(F)} + x_i + z$ and $R \equiv R^{(F)} + x_1 + x_2$. The population dynamics of the consumers and resources follows:

$$
\begin{cases}
\dot{x}_i = a_i C_i^{(F)} R^{(F)} - (k_i + d_i)x_i, i = 1, 2; \\
\dot{z} = a'_{12} C_1^{(F)} C_2^{(F)} - d'_{12}z, \\
\dot{C}_i = w_i k_i x_i - D_i C_i, \\
\dot{R} = g(R, x_1, x_2, C_1, C_2).
\end{cases}
\tag{S25}
$$

where the functional form of $g(R, x_1, x_2, C_1, C_2)$ is unspecific, while $D_i$ and $w_i$ represents the mortality rates of the two consumers species and biomass conversion ratios. Still, the consumption/interference process is supposed to be in fast equilibrium, that is $\dot{x}_i = 0, \dot{z} = 0$. In the case that $R \gg C_1 + C_2 > x_1 + x_2$, by applying $R^{(F)} \approx R$, we obtain:

$$x_1 \approx \frac{2C_1(R/K_2 + 1)R/K_1}{\sqrt{[\gamma(C_2 - C_1) + (\frac{R}{K_1} + 1)(\frac{R}{K_2} + 1)]^2 + 4\gamma C_1(\frac{R}{K_1} + 1)(\frac{R}{K_2} + 1)} + \gamma(C_2 - C_1) + (\frac{R}{K_1} + 1)(\frac{R}{K_2} + 1)}, \tag{S26a}$$

$$x_2 \approx \frac{2C_2(R/K_1 + 1)R/K_2}{\sqrt{[\gamma(C_1 - C_2) + (\frac{R}{K_1} + 1)(\frac{R}{K_2} + 1)]^2 + 4\gamma C_2(\frac{R}{K_1} + 1)(\frac{R}{K_2} + 1)} + \gamma(C_1 - C_2) + (\frac{R}{K_1} + 1)(\frac{R}{K_2} + 1)}. \tag{S26b}$$

Then, the searching efficiencies and functional responses are:

$$\Xi_1^{\text{inter}}(R, C_1, C_2)_{(1)} = \frac{2k_2(R/K_1 + 1)R/K_2}{\gamma(C_1 - C_2) + (\frac{R}{K_1} + 1)(\frac{R}{K_2} + 1) + \sqrt{[\gamma(C_1 - C_2) + (\frac{R}{K_1} + 1)(\frac{R}{K_2} + 1)]^2 + 4\gamma C_2(\frac{R}{K_1} + 1)(\frac{R}{K_2} + 1)}}, \tag{S27a}$$

$$\Xi_2^{\text{inter}}(R, C_1, C_2)_{(1)} = \frac{2k_2(R/K_1 + 1)/K_2}{\gamma(C_1 - C_2) + (\frac{R}{K_1} + 1)(\frac{R}{K_2} + 1) + \sqrt{[\gamma(C_1 - C_2) + (\frac{R}{K_1} + 1)(\frac{R}{K_2} + 1)]^2 + 4\gamma C_2(\frac{R}{K_1} + 1)(\frac{R}{K_2} + 1)}}, \tag{S27b}$$

$$\mathcal{F}_1^{\text{inter}}(R, C_1, C_2)_{(1)} = \frac{2k_1(R/K_2 + 1)R/K_1}{\gamma(C_2 - C_1) + (\frac{R}{K_1} + 1)(\frac{R}{K_2} + 1) + \sqrt{[\gamma(C_2 - C_1) + (\frac{R}{K_1} + 1)(\frac{R}{K_2} + 1)]^2 + 4\gamma C_1(\frac{R}{K_1} + 1)(\frac{R}{K_2} + 1)}}, \tag{S27c}$$

$$\mathcal{F}_2^{\text{inter}}(R, C_1, C_2)_{(1)} = \frac{2k_2(R/K_1 + 1)R/K_2}{\gamma(C_1 - C_2) + (\frac{R}{K_1} + 1)(\frac{R}{K_2} + 1) + \sqrt{[\gamma(C_1 - C_2) + (\frac{R}{K_1} + 1)(\frac{R}{K_2} + 1)]^2 + 4\gamma C_2(\frac{R}{K_1} + 1)(\frac{R}{K_2} + 1)}}. \tag{S27d}$$

Since $\frac{4\gamma^2 C_1 C_2}{[\gamma C_1 + \gamma C_2 + (\frac{R}{K_1} + 1)(\frac{R}{K_2} + 1)]^2} < 1$, by applying first-order approximation to the denominator of *Equation S26b*, we obtain:

$$x_1 \approx \frac{C_1 R}{(R + K_1) + \frac{\gamma K_1 K_2 C_2}{(R + K_2)} - \frac{\gamma^2 K_1 K_2 C_1 C_2}{[\gamma(C_1 + C_2) + (\frac{R}{K_1} + 1)(\frac{R}{K_2} + 1)](R + K_2)}}, \tag{S28a}$$

$$x_2 \approx \frac{C_2 R}{(R + K_2) + \frac{\gamma K_1 K_2 C_1}{(R + K_1)} - \frac{\gamma^2 K_1 K_2 C_1 C_2}{[\gamma(C_1 + C_2) + (\frac{R}{K_1} + 1)(\frac{R}{K_2} + 1)](R + K_1)}}, \tag{S28b}$$

and the searching efficiencies and functional responses are:

$$\Xi_1^{\text{inter}}(R, C_1, C_2)_{(2)} = \frac{k_1}{(R + K_1) + \frac{\gamma K_1 K_2 C_2}{(R + K_2)} - \frac{\gamma^2 K_1 K_2 C_1 C_2}{[\gamma(C_1 + C_2) + (\frac{R}{K_1} + 1)(\frac{R}{K_2} + 1)](R + K_2)}}, \tag{S29a}$$

$$\Xi_2^{\text{inter}}(R, C_1, C_2)_{(2)} = \frac{k_2}{(R + K_2) + \frac{\gamma K_1 K_2 C_1}{(R + K_1)} - \frac{\gamma^2 K_1 K_2 C_1 C_2}{[\gamma(C_1 + C_2) + (\frac{R}{K_1} + 1)(\frac{R}{K_2} + 1)](R + K_1)}}, \tag{S29b}$$

$$\mathcal{F}_1^{\text{inter}}(R, C_1, C_2)_{(2)} = \frac{k_1 R}{(R + K_1) + \frac{\gamma K_1 K_2 C_2}{(R + K_2)} - \frac{\gamma^2 K_1 K_2 C_1 C_2}{[\gamma(C_1 + C_2) + (\frac{R}{K_1} + 1)(\frac{R}{K_2} + 1)](R + K_2)}}, \tag{S29c}$$

$$\mathcal{F}_2^{\text{inter}}(R, C_1, C_2)_{(2)} = \frac{k_2 R}{(R + K_2) + \frac{\gamma K_1 K_2 C_1}{(R + K_1)} - \frac{\gamma^2 K_1 K_2 C_1 C_2}{[\gamma(C_1 + C_2) + (\frac{R}{K_1} + 1)(\frac{R}{K_2} + 1)](R + K_1)}}. \tag{S29d}$$

Likewise, the B-D model only fits to cases with $d = 0$. By calculating the average values in a stochastic framework, we obtain $\langle t_h^i \rangle = \frac{1}{k_i}$, $\langle t_w^i \rangle = \frac{1}{d'_{12}}$ ($i = 1, 2$). Then, we obtain the searching efficiencies in the B-D model:

$$\Xi_1^{\text{B-D (inter)}}(R, C_1, C_2) = \frac{a_1}{1 + \frac{a_1}{k_1}R + \frac{a'_{12}}{d'_{12}}C_2} = \frac{a_1}{1 + R/K_1 \mid_{d=0} + \gamma C_2}, \tag{S30a}$$

$$\Xi_2^{\text{B-D (inter)}}(R, C_1, C_2) = \frac{a_2}{1 + \frac{a_2}{k_2}R + \frac{a'_{12}}{d'_{12}}C_1} = \frac{a_2}{1 + R/K_2 \mid_{d=0} + \gamma C_1}. \tag{S30b}$$

Consequently, the functional responses in the B-D model are:

$$\mathcal{F}_1^{\text{B-D (inter)}}(R, C_1, C_2) = \frac{a_1 R}{1 + R/K_1 \mid_{d=0} + \gamma C_2}, \tag{S31a}$$

$$\mathcal{F}_2^{\text{B-D (inter)}}(R, C_1, C_2) = \frac{a_2 R}{1 + R/K_2 \mid_{d=0} + \gamma C_1}. \tag{S31b}$$

Evidently, the searching efficiencies in the B-D model are overall different from either the quasi rigorous form $\Xi_i(R, C_1, C_2)_1$, or the simplified form $\Xi_i(R, C_1, C_2)_2$ (**Appendix 1—figure 2G–I**). Still, the discrepancy can be small when $d \approx 0$ and $R \gg C$ (**Appendix 1—figure 2G–I**). Intuitively, when $\gamma \ll \min(C_1^{-1}, C_2^{-1})$, we have:

$$\Xi_1^{\text{inter}}(R, C_1, C_2)_{(2)} \approx \frac{a_1}{(1 + \frac{a_1}{k_1}R) + \frac{\gamma C_2}{R/K_2 + 1}}, \tag{S32a}$$

$$\Xi_2^{\text{inter}}(R, C_1, C_2)_{(2)} \approx \frac{a_2}{(1 + \frac{a_2}{k_2}R) + \frac{\gamma C_1}{R/K_1 + 1}}. \tag{S32b}$$

Thus, if $R/K_i = x_i/C_i^{(\text{F})} < 1$ $(i = 1, 2)$, then $\frac{1}{1+R/K_i} \in [0.5, 1]$. In this case, the difference between $\Xi_i^{\text{B-D (inter)}}(R, C_1, C_2)$ and $\Xi_i^{\text{inter}}(R, C_1, C_2)_{(2)}$ is small.

## Appendix 4

### Scenario involving chasing pairs and intraspecific interference

#### A Two consumers species competing for one resource species

We consider the scenario involving chasing pairs and intraspecific interference in the simple case of $S_C = 2$ and $S_R = 1$:

$$C_i^{(F)} + R^{(F)} \underset{d_i}{\overset{a_i}{\rightleftharpoons}} C_i^{(P)} \vee R^{(P)} \overset{k_i}{\rightarrow} C_i^{(F)}(+),$$

$$C_i^{(F)} + C_i^{(F)} \underset{d_i'}{\overset{a_i'}{\rightleftharpoons}} C_i^{(P)} \vee C_i^{(P)}, \ i = 1, 2.$$

Here, the variables and parameters are just extended from the case of $S_C = 1$ and $S_R = 1$ (see Appendix 3.C). The total number of consumers and resources are $C_i \equiv C_i^{(F)} + x_i + 2y_i$ and $R \equiv R^{(F)} + \sum_{i=1}^{2} x_i$. Then, the population dynamics of the consumers and resources can be described as follows:

$$\begin{cases} \dot{x}_i = a_i C_i^{(F)} R^{(F)} - (k_i + d_i)x_i, i = 1, 2; \\ \dot{y}_i = a_i'[C_i^{(F)}]^2 - d_i'y_i, \\ \dot{C}_i = w_i k_i x_i - D_i C_i, \\ \dot{R} = g(R, x_1, x_2, C_1, C_2). \end{cases} \tag{S33}$$

The functional form of $g(R, x_1, x_2, C_1, C_2)$ is unspecified. For simplicity, we limit our analysis to abiotic resources, while all results generically apply to biotic resources. Besides, we define $K_i \equiv (d_i + k_i)/a_i$, $\alpha_i \equiv D_i/(w_i k_i)$ and $\beta_i \equiv a_i'/d_i'$ $(i = 1, 2)$. At steady state, from $\dot{x}_i = 0, \dot{y}_i = 0$, we have:

$$\begin{cases} x_i = C_i^{(F)} R^{(F)}/K_i, \ i = 1, 2; \\ y_i = \beta_i[C_i^{(F)}]^2. \end{cases} \tag{S34}$$

Note that $C_i \equiv C_i^{(F)} + x_i + 2y_i$, and $R \equiv R^{(F)} + \sum_{i=1}^{2} x_i$. Then,

$$R^{(F)} = R/(1 + C_1^{(F)}/K_1 + C_2^{(F)}/K_2), \tag{S35a}$$

$$C_i = C_i^{(F)} + R^{(F)} C_i^{(F)}/K_i + 2\beta_i[C_i^{(F)}]^2, \ i = 1, 2. \tag{S35b}$$

By substituting *Equation S35a* into *Equation S35b*, we have:

$$C_2^{(F)} = \frac{K_2}{K_1}[\frac{RC_1^{(F)}}{C_1 - C_1^{(F)} - 2\beta_1[C_1^{(F)}]^2} - K_1 - C_1^{(F)}], \tag{S36a}$$

$$(C_2 - C_2^{(F)} - 2\beta_2[C_2^{(F)}]^2)(1 + C_1^{(F)}/K_1 + C_2^{(F)}/K_2) = RC_2^{(F)}/K_2. \tag{S36b}$$

Then, we can present $C_i^{(F)}$ with $C_1, C_2$ and $R$ $(i = 1, 2)$. By further combining with *Equation S34*, *Equation S35a* and *Equation S36a*, we express $R^{(F)}, x_i$ and $y_i$ using $C_1, C_2$ and $R$. In particular, for $x_i$, we have:

$$x_i = u_i(R, C_1, C_2), i = 1, 2. \tag{S37}$$

If all species coexist, then the steady-state equations of $\dot{C}_i = 0$ and $\dot{R} = 0$ are:

$$\begin{cases} \Omega_1(R, C_1, C_2) - D_1 = 0, \\ \Omega_2(R, C_1, C_2) - D_2 = 0, \\ G(R, C_1, C_2) = 0, \end{cases} \tag{S38}$$

where $G(R, C_1, C_2) \equiv g(R, u_1(R, C_1, C_2), u_2(R, C_1, C_2), C_1, C_2)$, and $\Omega_i(R, C_1, C_2) \equiv \frac{w_i k_i}{C_i} u_i(R, C_1, C_2)$. In practice, *Equation S38* corresponds to three unparallel surfaces, which share a common point (*Figure 1H* and *Appendix 1—figure 3G*). Importantly, the fixed point can be stable, and hence two consumer species may coexist at constant population densities.

## 1 Stability analysis of the fixed-point solution

We use linear stability analysis to study the local stability of the fixed point. Specifically, for an arbitrary fixed point $E(x_1, x_2, y_1, y_2, C_1, C_2, R)$, only when all the eigenvalues (defined as $\lambda_i, i = 1, \cdots, 7$) of the Jacobian matrix at point $E$ own negative real parts would the point be locally stable.

To investigate whether there exists a non-zero measure parameter region for species coexistence, we set $D_i$ $(i = 1, 2)$ to be the only parameter that varies with species $C_1$ and $C_2$, and then $\Delta \equiv (D_1 - D_2)/D_2$ reflects the completive difference between the two consumer species. As shown in *Appendix 1—figure 4B*, the region below the blue surface and above the red surface corresponds to stable coexistence. Thus, there exists a non-zero measure parameter region to promote species coexistence, which breaks CEP.

## 2 Analytical solutions of the species abundances at steady state

At steady state, since $\dot{x}_i = \dot{y}_i = \dot{C}_i = 0$ $(i = 1, 2)$, then,

$$\begin{cases} x_i = \alpha_i C_i, \\ C_i^{(\mathrm{F})} = K_i \alpha_i C_i / R^{(\mathrm{F})}, \\ y_i = \beta_i (K_i \alpha_i C_i)^2 [R^{(\mathrm{F})}]^{-2}. \end{cases} \tag{S39}$$

Meanwhile, $C_i = C_i^{(\mathrm{F})} + x_i + 2y_i$, and $C_i, R > 0$. Then, we have:

$$C_i = \frac{(1 - \alpha_i)[R^{(\mathrm{F})}]^2 - K_i \alpha_i R^{(\mathrm{F})}}{2\beta_i (K_i \alpha_i)^2}. \tag{S40}$$

If the resource species owns a much larger population abundance than the consumers (i.e. $R \gg C_1 + C_2$), then $R \gg x_1 + x_2$, and $R^{(\mathrm{F})} \approx R$. Thus,

$$C_i = \frac{(1 - \alpha_i)R^2 - K_i \alpha_i R}{2\beta_i (K_i \alpha_i)^2}. \tag{S41}$$

By further assuming that the population dynamics of the resources follow identical construction rule as the MacArthur's consumer-resource model (*MacArthur, 1970*), we have:

$$g(R, x_1, x_2, C_1, C_2) = \zeta(1 - R/\kappa) - (k_1 x_1 + k_2 x_2), \tag{S42}$$

Since $\dot{R} = 0$, then

$$R = \frac{-o_1 + \sqrt{o_1^2 + 4o_2\zeta}}{2o_2}, \tag{S43}$$

where $o_1 \equiv \frac{\zeta}{\kappa} - \frac{k_1}{2\beta_1 K_1} - \frac{k_2}{2\beta_2 K_2}$ and $o_2 \equiv \frac{k_1(1 - \alpha_1)}{2\beta_1 \alpha_1 (K_1)^2} + \frac{k_2(1 - \alpha_2)}{2\beta_2 \alpha_2 (K_2)^2}$

*Equations. S41, S43* are the analytical solutions of species abundances at steady state when $R \gg C_1 + C_2$. As shown in *Figure 1E*, the analytical solutions agree well with the numerical results (the exact solutions). To conduct a systematic comparison for different model parameters, we assign $D_i$ to be the only parameter varying with species $C_1$ and $C_2$ $(D_1 > D_2)$, and define $\Delta \equiv (D_1 - D_2)/D_2$ as the competitive difference between the two consumer species. The comparison between analytical

solutions and numerical results is shown in *Appendix 1—figure 3H*. Clearly, they are close to each other, exhibiting very good consistency.

Furthermore, we test if the parameter region for species coexistence is predictable using the analytical solutions. Since $D_i$ is the only parameter that varies with the two-consumer species, the supremum of the competitive difference tolerated for species coexistence (defined as $\widehat{\Delta}$) corresponds to the steady-state solutions that satisfy $R, C_2 > 0$ and $C_1 = 0^+$, where $0^+$ stands for the infinitesimal positive number. To calculate the analytical solutions at the upper surface of the coexistence region, where $\Delta = \widehat{\Delta}$ and $C_1 = 0^+$, we further combine *Equation S41* and then obtain (note that $R > 0$):

$$R = \frac{K_1 \alpha_1}{1 - \alpha_1}. \tag{S44}$$

Meanwhile, $\alpha_1 = \alpha_2(\Delta + 1)$. Thus, for the upper surface of the coexistence region:

$$\alpha_1 = \alpha_2(\widehat{\Delta} + 1). \tag{S45}$$

Combining *Equations S43-S45*, we have:

$$\widehat{\Delta} = \frac{1}{\alpha_2(\kappa_1 \varpi + 1)} - 1, \tag{S46}$$

where $\varpi \equiv \frac{1}{2}(\frac{1}{\kappa} - \frac{k_2}{2\zeta\beta_2 K_2}) + \frac{1}{2}\sqrt{(\frac{1}{\kappa} - \frac{k_2}{2\zeta\beta_2 K_2})^2 + 2\frac{k_2(1-\alpha_2)}{\zeta\beta_2\alpha_2(K_2)^2}}$. When $R \gg C_1 + C_2$, the comparison of $\widehat{\Delta}$ obtained from analytical solutions with that from numerical results (the exact solutions) are shown in *Appendix 1—figure 3I*, which overall exhibits good consistency.

## B $S_C$ consumers species competing for $S_R$ resources species

Here, we consider the scenario involving chasing pairs and intraspecific interference for the generic case with $S_C$ types of consumers and $S_R$ types of resources. Then, the population dynamics of the system can be described as follows:

$$\begin{cases} \dot{x}_{il} = a_{il}C_i^{(F)}R_l^{(F)} - (k_{il} + d_{il})x_{il}, \\ \dot{y}_i = a'_{ii}[C_i^{(F)}]^2 - d'_{ii}y_i, \\ \dot{C}_i = \sum\limits_{l=1}^{S_R} w_{il}k_{il}x_{il} - D_iC_i, \\ \dot{R}_l = g_l(\{R_l\}, \{x_i\}, \{C_i\}), i = 1, \cdots, S_C, l = 1, \cdots, S_R. \end{cases} \tag{S47}$$

Note that *Equation S47* is identical with *Equations 1-2*, and we use the same variables and parameters as that in the main text. Then, the populations of the consumers and resources are $C_i = C_i^{(F)} + \sum\limits_{l=1}^{S_R} x_{il} + 2y_i$ and $R_l = R_l^{(F)} + \sum\limits_{i=1}^{S_C} x_{il}$. For convenience, we define $K_{il} \equiv (d_{il} + k_{il})/a_{il}$, $\alpha_{il} \equiv D_{il}/(k_{il}w_{il})$ and $\beta_i \equiv a'_{ii}/d'_{ii}$ ($i = 1, \cdots, S_C$, $l = 1, \cdots, S_R$).

### 1 Analytical solutions of species abundances at steady state

At steady state, from $\dot{x}_{il} = 0, \dot{y}_i = 0$, and $\dot{C}_i = 0$, we have:

$$\begin{cases} x_{il} = C_i^{(F)}R_l^{(F)}/K_{il}, \\ y_i = \beta_i[C_i^{(F)}]^2, \\ C_i = \sum\limits_{l=1}^{S_R} x_{il}/\alpha_{il} = \sum\limits_{l=1}^{S_R} C_i^{(F)}R_l^{(F)}/(K_{il}\alpha_{il}). \end{cases} \tag{S48}$$

Meanwhile $C_i = C_i^{(F)} + \sum\limits_{l=1}^{S_R} x_{il} + 2y_i$, and note that $C_i > 0$, thus,

$$C_i^{(F)} = \frac{1}{2\beta_i}[-1 + \sum\limits_{l=1}^{S_R}(\frac{1}{\alpha_{il}} - 1)\frac{R_l^{(F)}}{K_{il}}]. \tag{S49}$$

Combined with **Equation S49**, and then,

$$C_i = \sum_{l=1}^{S_R} \frac{R_l^{(F)}}{2\beta_i \alpha_{il} K_{il}} [-1 + \sum_{l'=1}^{S_R} (\frac{1}{\alpha_{il'}} - 1) \frac{R_{l'}^{(F)}}{K_{il'}}]. \tag{S50}$$

We further assume that the specific function of $g_l(\{R_l\}, \{x_i\}, \{C_i\})$ satisfies **Equation 4**, that is

$$g_l(\{R_l\}, \{x_i\}, \{C_i\}) = \zeta_l(1 - R_l/\kappa_l) - \sum_{i=1}^{S_C} k_{il} x_{il}. \tag{S51}$$

By combining **Equations S48, S49 and S51**, we have:

$$\zeta_l(1 - \frac{R_l}{\kappa_l}) = \sum_{i=1}^{S_C} \frac{k_{il}}{2\beta_i K_{il}} [-1 + \sum_{l'=1}^{S_R} (\frac{1}{\alpha_{il'}} - 1) \frac{R_{l'}^{(F)}}{K_{il'}}] R_l^{(F)}. \tag{S52}$$

If the population abundance of each resource species is much more than the total population of all consumers (i.e. $R_l \gg \sum_{i=1}^{S_C} C_i (l = 1, \cdots, S_R)$), then $R_l \gg \sum_{i=1}^{S_C} x_{il}$ and $R_l^{(F)} \approx R_l$. Thus,

$$(\frac{\zeta_l}{\kappa_l} - \sum_{i=1}^{S_C} \frac{k_{il}}{2\beta_i K_{il}} + \sum_{l'=1}^{S_R} \sum_{i=1}^{S_C} \frac{k_{il}}{2\beta_i K_{il}} (\frac{1}{\alpha_{il'}} - 1) \frac{R_{l'}}{K_{il'}}) R_l = \zeta_l, \tag{S53}$$

with $l = 1, \cdots, S_R$. **Equation S53** is a set of second-order algebraic differential equations, which is clearly solvable.

In fact, when $S_R = 1, S_C \geq 1$, and $R_l \gg \sum_{i=1}^{S_C} C_i (l = 1)$, we can explicitly present the analytical solution of the steady-state species abundances. To simplify the notations, we omit the '$l$' in the sub-/super-scripts since $S_R = 1$. Then, we have:

$$\begin{cases} R = \dfrac{-\iota_1 + \sqrt{\iota_1^2 + 4\iota_2 \zeta}}{2\iota_2}, \\ C_i = \dfrac{1}{2\beta_i \alpha_i K_i} [(\dfrac{1}{\alpha_i} - 1) \dfrac{R}{K_i} - 1] R, \ i = 1, \cdots, S_C. \end{cases} \tag{S54}$$

Here $\iota_1 \equiv \frac{\zeta}{\kappa} - \sum_{i=1}^{S_C} \frac{k_i}{2\beta_i K_i}$ and $\iota_2 \equiv \sum_{i=1}^{S_C} \frac{k_i(1 - \alpha_i)}{2\beta_i \alpha_i (K_i)^2}$

## C Intuitive understanding: an underlying negative feedback loop

Intuitively, how can intraspecific predator interference promote biodiversity? Here, we solve this question by considering the case that $S_C$ types of consumers compete for one resource species. The population dynamics of the system are described in **Equations S47 and S51** with $S_R = 1$. To simplify the notations, we omit the '$l$' in the subscript since $S_R = 1$. The consumption process and interference process are supposed to be in fast equilibrium (i.e. $\dot{x}_i = 0, \dot{y}_i = 0$). Then, we have a set of equations to solve for $x_i$ and $y_i$ given the population size of each species:

$$\begin{cases} x_i = C_i^{(F)} R^{(F)} / K_i, \\ y_i = \beta_i [C_i^{(F)}]^2, \\ C_i = C_i^{(F)} + x_i + 2y_i, \\ R = R^{(F)} + \sum_{i=1}^{S_C} x_i. \end{cases} \tag{S55}$$

In the first three sub-equations of **Equation S55**, by getting rids of $C_i^{(F)}$, we have,

$$\begin{cases} 2\beta_i [\dfrac{x_i}{R^{(F)}}]^2 + x_i + \dfrac{K_i x_i}{R^{(F)}} - C_i = 0, \\ y_i = \dfrac{1}{2} [C_i - x_i - \dfrac{x_i}{R^{(F)}}]. \end{cases} \tag{S56}$$

Then, by regarding $R^{(F)}$ as a temporary parameter, we solve for $x_i$ and $y_i$:

$$\begin{cases} x_i = \dfrac{2R^{(F)}C_i}{\sqrt{(R^{(F)} + K_i)^2 + 8\beta_i K_i^2 C_i} + R^{(F)} + K_i}, \\ y_i = \dfrac{1}{2}[C_i - (1 + \dfrac{K_i}{R^{(F)}})x_i]. \end{cases}$$ (S57)

If the total population size of the resources is much larger than that of consumers (i.e. $R \gg \sum\limits_{i=1}^{S_C} C_i$), then $R \gg \sum\limits_{i=1}^{S_C} x_i$ and $R^{(F)} \approx R$, and thus we get the analytical expressions of $x_i$ and $y_i$:

$$\begin{cases} x_i \approx \dfrac{2RC_i}{\sqrt{(R + K_i)^2 + 8\beta_i K_i^2 C_i} + R + K_i}, \\ y_i \approx \dfrac{C_i}{2}[1 - \dfrac{2(R + K_i)}{\sqrt{(R + K_i)^2 + 8C_i\beta_i K_i^2} + R + K_i}]. \end{cases}$$ (S58)

Note that the fraction of $C_i$ individuals engaged in chasing pairs is $x_i/C_i$, while that for individuals trapped in intraspecific interference pairs is $y_i/C_i$. With *Equation S58*, it is straightforward to obtain these fractions:

$$\begin{cases} x_i/C_i \approx \dfrac{2R}{\sqrt{(R + K_i)^2 + 8\beta_i K_i^2 C_i} + R + K_i}, \\ y_i/C_i \approx \dfrac{1}{2}(1 - \dfrac{2(R + K_i)}{\sqrt{(R + K_i)^2 + 8\beta_i K_i^2 C_i} + R + K_i}). \end{cases}$$ (S59)

where both $x_i/C_i$ and $y_i/C_i$ are bivariate functions of $R$ and $C_i$. From *Equation S59*, it is clear that for a given population size of the resource species, $y_i/C_i$ is a monotonously increasing function of $C_i$, while $x_i/C_i$ is a monotonously decreasing function of $C_i$. In *Appendix 1—figure 15A, B*, we see that the analytical results are highly consistence with the exact numerical solutions. By definition, the functional response of $C_i$ species is $\mathcal{F} \equiv k_i x_i/C_i$, and thus,

$$\mathcal{F}(R, C_i) \approx \dfrac{2R}{\sqrt{(R + K_i)^2 + 8\beta_i K_i^2 C_i} + R + K_i},$$ (S60)

Evidently, the function response of $C_i$ species is negatively correlated with the population size of itself, which effectively constitutes a self-inhibiting negative feedback loop (*Appendix 1—figure 15C*).

Then, we have a simple intuitive understanding of species coexistence through the mechanism of intraspecific interference. In an ecological community, consumer species that of higher/lower competitiveness tend to increase/decrease their population size in the competition process. Without intraspecific interference, the increasing/decreasing trend would continue until the system obeys CEP. In the scenario involving intraspecific interference, however, for species of higher competitiveness (e.g. $C_i$), with the increase of $C_i$'s population size, a larger portion of $C_i$ individuals are then engaged in intraspecific interference pair which are temporarily absent from hunting (*Appendix 1—figure 15A, B*). Consequently, the functional response of $C_i$ drops, which prevents further increase of $C_i$'s population size, results in an overall balance among the consumer species, and thus promotes species coexistence.

## Appendix 5

### Scenario involving chasing pairs and interspecific interference

Here, we consider the scenario involving chasing pairs and interspecific interference in the case of $S_C = 2$ and $S_R = 1$, with all settings follow that depicted in Appendix 3. D. Then, $C_i \equiv C_i^{(F)} + x_i + z, R \equiv R^{(F)} + x_1 + x_2$, and the population dynamics follows (identical with **Equation S25**):

$$\begin{cases} \dot{x}_i = a_i C_i^{(F)} R^{(F)} - (k_i + d_i)x_i, i = 1, 2; \\ \dot{z} = a'_{12} C_1^{(F)} C_2^{(F)} - d'_{12} z, \\ \dot{C}_i = w_i k_i x_i - D_i C_i, \\ \dot{R} = g(R, x_1, x_2, C_1, C_2). \end{cases} \tag{S61}$$

Here, the functional form of $g(R, x_1, x_2, C_1, C_2)$ is unspecified. For convenience, we define $K_i \equiv (d_i + k_i)/a_i$, $\alpha_i \equiv D_i/(w_i k_i)$ $(i = 1, 2)$ and $\gamma \equiv a'_{12}/d'_{12}$. At steady state, from $\dot{x}_i = 0$ $(i = 1, 2)$ and $\dot{z} = 0$, we have:

$$\begin{cases} x_i = C_i^{(F)} R^{(F)}/K_i, i = 1, 2; \\ z = \gamma C_1^{(F)} C_2^{(F)}. \end{cases} \tag{S62}$$

Note that $C_i \equiv C_i^{(F)} + x_i + z$ and $R \equiv R^{(F)} + x_1 + x_2$, then,

$$\begin{cases} C_1 = C_1^{(F)} + R^{(F)} C_1^{(F)}/K_1 + \gamma C_1^{(F)} C_2^{(F)}, \\ C_2 = C_2^{(F)} + R^{(F)} C_2^{(F)}/K_2 + \gamma C_1^{(F)} C_2^{(F)}, \\ R = R^{(F)}(1 + C_1^{(F)}/K_1 + C_2^{(F)}/K_2). \end{cases} \tag{S63}$$

Then, we can express $C_1^{(F)}, C_2^{(F)}$ and $R^{(F)}$ with $C_1, C_2$ and $R$. Combined with **Equation S62**, $x_i$ and $z$ can also be expressed using $C_1, C_2$ and $R$. In particular, for $x_i$, we have:

$$x_i = u'_i(R, C_1, C_2), i = 1, 2. \tag{S64}$$

If all species coexist, by defining $\Omega'_i(R, C_1, C_2) \equiv \frac{w_i k_i}{C_i} u'_i(R, C_1, C_2)$, then, the steady-state equations of $\dot{C}_i = 0$ $(i = 1, 2)$ and $\dot{R} = 0$ are:

$$\begin{cases} \Omega'_1(R, C_1, C_2) - D_1 = 0, \\ \Omega'_2(R, C_1, C_2) - D_2 = 0, \\ G'(R, C_1, C_2) = 0, \end{cases} \tag{S65}$$

where $G'(R, C_1, C_2) \equiv g(R, u'_1(R, C_1, C_2), u'_2(R, C_1, C_2), C_1, C_2)$.

Here, **Equation S65** corresponds to three unparallel surfaces and share a common point (**Figure 1G** and **Appendix 1—figure 3A**). However, all the fixed points are unstable (**Appendix 1—figure 3F**), and hence the consumer species cannot stably coexist at steady state (**Figure 1D**).

### A Analytical results of the fixed-point solution

We proceed to investigate the unstable fixed point where $R, C_1, C_2 > 0$. From $\dot{x}_i = 0$, $\dot{z} = 0$, $\dot{C}_i = 0$, and note that $C_i \equiv C_i^{(F)} + x_i + z$, we have:

$$\begin{cases} C_i = K_i \alpha_i C_i (R^{(F)})^{-1} + \alpha_i C_i + z, i = 1, 2; \\ z = \gamma K_1 \alpha_1 K_2 \alpha_2 (R^{(F)})^{-2} C_1 C_2. \end{cases} \tag{S66}$$

Since $C_i > 0$, then:

$$\begin{cases} C_1 = \dfrac{(1-\alpha_2)[R^{(\mathrm{F})}]^2 - K_2\alpha_2 R^{(\mathrm{F})}}{\gamma K_1\alpha_1 K_2\alpha_2}, \\ C_2 = \dfrac{(1-\alpha_1)[R^{(\mathrm{F})}]^2 - K_1\alpha_1 R^{(\mathrm{F})}}{\gamma K_1\alpha_1 K_2\alpha_2}. \end{cases} \tag{S67}$$

If $R \gg C_1 + C_2$, then $R \gg x_1 + x_2$ and $R^{(\mathrm{F})} \approx R$, we have:

$$\begin{cases} C_1 = \dfrac{(1-\alpha_2)R^2 - K_2\alpha_2 R}{\gamma K_1\alpha_1 K_2\alpha_2}, \\ C_2 = \dfrac{(1-\alpha_1)R^2 - K_1\alpha_1 R}{\gamma K_1\alpha_1 K_2\alpha_2}. \end{cases} \tag{S68}$$

Still, we assume that the population dynamics of the resource species follows *Equation S42*. At the fixed point, $\dot{R} = 0$. We have:

$$\zeta\left(1 - \frac{R}{\kappa}\right) = k_1\alpha_1 C_1 + k_2\alpha_2 C_2. \tag{S69}$$

Combined with *Equation S68*, we can solve for $R$:

$$R = \frac{-\varrho_1 + \sqrt{\varrho_1^2 + 4\varrho_2\zeta}}{2\varrho_2}. \tag{S70}$$

where $\varrho_1 \equiv \frac{\zeta}{\kappa} - \frac{k_1}{\gamma K_1} - \frac{k_2}{\gamma K_2}$ and $\varrho_2 \equiv \frac{k_1(1-\alpha_2)}{\gamma K_1 K_2\alpha_2} + \frac{k_2(1-\alpha_1)}{\gamma K_1 K_2\alpha_1}$.

*Equation S68*, *Equation S70* are the analytical solutions of the fixed point when $R \gg C_1 + C_2$. As shown in *Appendix 1—figure 3E*, the analytical predictions agree well with the numerical results (the exact solutions).

## Appendix 6

### Scenario involving chasing pairs and both intra- and inter-specific interference

Here, we consider the scenario involving chasing pairs and both intra- and inter-specific interference in the simple case of $S_C = 2$ and $S_R = 1$:

$$C_i^{(F)} + R^{(F)} \underset{d_i}{\overset{a_i}{\rightleftharpoons}} C_i^{(P)} \vee R^{(P)} \xrightarrow{k_i} C_i^{(F)}(+),$$

$$C_1^{(F)} + C_2^{(F)} \underset{d'_{12}}{\overset{a'_{12}}{\rightleftharpoons}} C_1^{(P)} \vee C_2^{(P)},$$

$$C_i^{(F)} + C_i^{(F)} \underset{d'_i}{\overset{a'_i}{\rightleftharpoons}} C_i^{(P)} \vee C_i^{(P)}, \ i = 1, 2.$$

We adopt the same notations as that depicted in **Appendix 4.A** and **Appendix 5**. Then, $C_i \equiv C_i^{(F)} + x_i + 2y_i + z$ and $R \equiv R^{(F)} + x_1 + x_2$, and the population dynamics of the system can be described as follows:

$$\begin{cases} \dot{x}_i = a_i C_i^{(F)} R^{(F)} - (k_i + d_i)x_i, \\ \dot{z} = a'_{12} C_1^{(F)} C_2^{(F)} - d'_{12} z, \\ \dot{y}_i = a'_i [C_i^{(F)}]^2 - d'_i y_i, \\ \dot{C}_i = w_i k_i x_i - D_i C_i, \\ \dot{R} = g(R, x_1, x_2, C_1, C_2), i = 1, 2. \end{cases} \tag{S71}$$

Here, the functional form of $g(R, x_1, x_2, C_1, C_2)$ follows **Equation S42**. For convenience, we define $K_i \equiv (d_i + k_i)/a_i, \alpha_i \equiv D_i/(w_i k_i), \beta_i \equiv a'_i/d'_i$, and $\gamma \equiv a'_{12}/d'_{12}, (i = 1, 2)$. At steady state, from $\dot{x}_i = 0, \dot{y}_i = 0, \dot{z} = 0,$ and $\dot{C}_i = 0, (i = 1, 2)$, we have:

$$\begin{cases} x_i = \alpha_i C_i, \\ C_i^{(F)} = K_i \alpha_i C_i (R^{(F)})^{-1}, \\ y_i = \beta_i (K_i \alpha_i C_i)^2 [R^{(F)}]^{-2}, \\ z = \gamma K_1 \alpha_1 K_2 \alpha_2 [R^{(F)}]^{-2} C_1 C_2. \end{cases} \tag{S72}$$

Combined with $C_i \equiv C_i^{(F)} + x_i + 2y_i + z$, and since $C_i > 0$ $(i = 1, 2)$, then,

$$\begin{cases} (1 - \alpha_1)(R^{(F)})^2 - K_1 \alpha_1 R^{(F)} = 2\beta_1(K_1\alpha_1)^2 C_1 + \gamma K_1 \alpha_1 K_2 \alpha_2 C_2, \\ (1 - \alpha_2)(R^{(F)})^2 - K_2 \alpha_2 R^{(F)} = 2\beta_2(K_2\alpha_2)^2 C_2 + \gamma K_1 \alpha_1 K_2 \alpha_2 C_1. \end{cases} \tag{S73}$$

### A Analytical solutions of species abundances at steady state

If $R \gg C_1 + C_2$, then $R \gg x_1 + x_2$ and thus $R^{(F)} \approx R$. Combined with **Equation S73**, we obtain:

$$\begin{cases} C_1 = R\dfrac{(2\beta_2 K_2 \alpha_2(1 - \alpha_1) - \gamma K_1 \alpha_1(1 - \alpha_2))R + (\gamma - 2\beta_2)K_1 \alpha_1 K_2 \alpha_2}{K_1^2 \alpha_1^2 K_2 \alpha_2(4\beta_1\beta_2 - \gamma^2)}, \\ C_2 = R\dfrac{(2\beta_1 K_1 \alpha_1(1 - \alpha_2) - \gamma K_2 \alpha_2(1 - \alpha_1))R + (\gamma - 2\beta_1)K_1 \alpha_1 K_2 \alpha_2}{K_1 \alpha_1 K_2^2 \alpha_2^2(4\beta_1\beta_2 - \gamma^2)}. \end{cases} \tag{S74}$$

Using $\dot{R} = 0$ and $R > 0$, we have:

$$R = \frac{-\chi_2 + \sqrt{(\chi_2)^2 + 4\chi_1\zeta}}{2\chi_1}, \tag{S75}$$

where $\quad \chi_1 \equiv \frac{k_2\gamma(\alpha_1-1)}{K_1K_2\alpha_1(4\beta_1\beta_2-\gamma^2)} + \frac{k_1\gamma(\alpha_2-1)}{K_1K_2\alpha_2(4\beta_1\beta_2-\gamma^2)} - \frac{k_12\beta_2(\alpha_1-1)}{K_1^2\alpha_1(4\beta_1\beta_2-\gamma^2)} - \frac{k_22\beta_1(\alpha_2-1)}{K_2^2\alpha_2(4\beta_1\beta_2-\gamma^2)},\quad$ and

$\chi_2 \equiv \frac{k_1(\gamma-2\beta_2)}{K_1(4\beta_1\beta_2-\gamma^2)} + \frac{k_2(\gamma-2\beta_1)}{K_2(4\beta_1\beta_2-\gamma^2)} + \frac{\zeta}{\kappa}$. **Equations S74-S75** are the analytical solutions of the species abundances at steady state when $R \gg C_1 + C_2$. As shown in **Appendix 1—figure 5E**, the analytical calculations agree well with the numerical results (the exact solutions).

## B Stability analysis of the coexisting state

In the scenario involving chasing pairs and both intra- and inter-specific interference, the behavior of species coexistence is similar to that without interspecific interference. Evidently, the influence of interspecific interference would be negligible if $d'_{12}$ is extremely large, and vice versa for intraspecific interference if both $d'_1$ and $d'_2$ are tremendous. In the deterministic framework, the two-consumer species may coexist at constant population densities (**Appendix 1—figure 5B**), and the fixed points are globally attracting (**Appendix 1—figure 5C**). Furthermore, there is a non-zero measure of parameter set where both consumer species can coexist at steady state with only one type of resources (**Appendix 1—figure 5A**). In the stochastic framework, just as the scenario involving chasing pairs and intraspecific interference, the coexistence state can be maintained along with stochasticity (**Appendix 1—figure 5D**).

## Appendix 7

### Dimensional analysis for the scenario involving chasing pairs and both intra- and inter-specific interference

The population dynamics of the system involving chasing pairs and both intra- and inter-specific interference are shown in *Equations 1-4*:

$$
\begin{cases}
\dot{x}_{il} = a_{il}C_i^{(\mathrm{F})}R_l^{(\mathrm{F})} - (d_{il} + k_{il})x_{il}, \\
\dot{y}_i = a_i'[C_i^{(\mathrm{F})}]^2 - d_i'y_i, \\
\dot{z}_{ij} = a_{ij}'C_i^{(\mathrm{F})}C_j^{(\mathrm{F})} - d_{ij}'z_{ij} \\
\dot{C}_i = \sum_{l=1}^{S_R} w_{il}k_{il}x_{il} - D_iC_i, \\
\dot{R}_l = \zeta_l(1 - R_l/\kappa_l) - \sum_{i=1}^{S_C} k_{il}x_{il},
\end{cases}
\tag{S76}
$$

with $l = 1, \cdots, S_R$; $i, j = 1, \cdots, S_C$, and $i \neq j$. Here, $C_i = C_i^{(\mathrm{F})} + \sum_l x_{il} + 2y_i + \sum_{i \neq j} z_{ij}$ and $R_l = R_l^{(\mathrm{F})} + \sum_i x_{il}$ represent the population abundances of the consumers and resources in the system. In fact, there are already several dimensionless variables and parameter in *Equation S76*, namely $x_{il}$, $y_i$, $z_{ij}$, $C_i^{(\mathrm{F})}$, $R_l^{(\mathrm{F})}$, $C_i$, $R_l$, $w_{il}$, $\kappa_l$. To make all terms dimensionless, we define $\tilde{t} = t/\tau$, where $\tau = \tilde{D}_1/D_1$ and $\tilde{D}_1$ is a reducible dimensionless parameter which is freely to take any positive values. Besides, we define dimensionless parameters $\tilde{a}_{il} = a_{il}\tau$, $\tilde{d}_{il} = d_{il}\tau$, $\tilde{k}_{il} = k_{il}\tau$, $\tilde{a}_i' = a_i'\tau$, $\tilde{d}_i' = d_i'\tau$, $\tilde{a}_{ij}' = a_{ij}'\tau$, $\tilde{d}_{ij}' = d_{ij}'\tau$, $\tilde{D}_i = D_i\tau$ and $\tilde{\zeta}_l = \zeta_l\tau$. By substituting all the dimensionless terms into *Equation S76*, we have:

$$
\begin{cases}
\dot{x}_{il} = \tilde{a}_{il}C_i^{(\mathrm{F})}R_l^{(\mathrm{F})} - (\tilde{d}_{il} + \tilde{k}_{il})x_{il}, \\
\dot{y}_i = \tilde{a}_i'[C_i^{(\mathrm{F})}]^2 - \tilde{d}_i'y_i, \\
\dot{z}_{ij} = \tilde{a}_{ij}'C_i^{(\mathrm{F})}C_j^{(\mathrm{F})} - \tilde{d}_{ij}'z_{ij} \\
\dot{C}_i = \sum_{l=1}^{S_R} w_{il}\tilde{k}_{il}x_{il} - \tilde{D}_iC_i, \\
\dot{R}_l = \tilde{\zeta}_l(1 - R_l/\kappa_l) - \sum_{i=1}^{S_C} \tilde{k}_{il}x_{il}.
\end{cases}
\tag{S77}
$$

For convenience, we omit the notation '˜' and use dimensionless variables and parameters in the simulation studies unless otherwise specified.

## Appendix 8

### Approximations applied in the pairwise encounter model

For consumers within a paired state, either in a chasing pair or an interference pair, the consumer may die following the mortality rate. Thus, in the scenario involving chasing pairs and both intra- and inter-specific interference, the population dynamics of the system should be described as follows:

$$
\begin{cases}
\dot{x}_{il} = a_{il}C_i^{(F)}R_l^{(F)} - (d_{il} + k_{il} + D_i)x_{il}, \\
\dot{y}_i = a_i'[C_i^{(F)}]^2 - (d_i' + D_i)y_i, \\
\dot{z}_{ij} = a_{ij}'C_i^{(F)}C_j^{(F)} - (d_{ij}' + D_i + D_j)z_{ij} \\
\dot{C}_i = \sum_{l=1}^{S_R} w_{il}k_{il}x_{il} - D_iC_i, \ i = 1, \cdots, S_C, \\
\dot{R}_l = \zeta_l(1 - R_l/\kappa_l) - \sum_{i=1}^{S_C} k_{il}x_{il}, \ l = 1, \cdots, S_R.
\end{cases}
\tag{S78}
$$

However, since predation or interference processes are generally much faster than birth and death processes, that is $D_i << k_{il}, d_{il}, d_i', d_{ij}'$, the influence of mortality rate in a paired state is negligible. Therefore, we have used the following approximations throughout our manuscript: $(k_{il} + d_{il} + D_i) \approx (k_{il} + d_{il})$, $(d_i' + D_i) \approx d_i', (d_{ij}' + D_i + D_j) \approx d_{ij}'$. Hence, the approximated population dynamics is described as follows:

$$
\begin{cases}
\dot{x}_{il} = a_{il}C_i^{(F)}R_l^{(F)} - (d_{il} + k_{il})x_{il}, \\
\dot{y}_i = a_i'[C_i^{(F)}]^2 - d_i'y_i, \\
\dot{z}_{ij} = a_{ij}'C_i^{(F)}C_j^{(F)} - d_{ij}'z_{ij} \\
\dot{C}_i = \sum_{l=1}^{S_R} w_{il}k_{il}x_{il} - D_iC_i, \ i = 1, \cdots, S_C, \\
\dot{R}_l = \zeta_l(1 - R_l/\kappa_l) - \sum_{i=1}^{S_C} k_{il}x_{il}, \ l = 1, \cdots, S_R,
\end{cases}
\tag{S79}
$$

which is identical to those shown in the main text.

## Appendix 9

### Simulation details of the main text figures

In **Figure 1C and F**: $a_i = 0.1$, $d_i = 0.5$, $w_i = 0.1$, $k_i = 0.1$, $(i = 1, 2)$, $D_1 = 0.002$, $D_2 = 0.001$, $\kappa = 5$, $\zeta = 0.05$. In **Figure 1D and G**: $a_i = 0.02$, $a'_i = 0.021$, $d_i = 0.5$, $d'_{ij} = 0.01$, $w_i = 0.08$, $k_i = 0.03$, $i, j = 1, 2$, $i \neq j$, $D_2 = 0.001$, $D_1 = 0.0011$, $\kappa = 20$, $\zeta = 0.01$. In **Figure 1E and H**: $a_i = 0.5$, $a'_i = 0.625$, $d_i = 0.5$, $d'_i = 0.02$, $w_i = 0.2$, $k_i = 0.4$, $(i = 1, 2)$, $D_1 = 0.0286$, $D_2 = 0.022$, $\kappa = 10$, $\zeta = 0.5$. **Figure 1C and F** were calculated or simulated from **Equations 1, 4**. **Figure 1D and G** were calculated or simulated from **Equations 1, 3 and 4**. **Figure 1E and H** were calculated or simulated from **Equations 1, 2 and 4**. The analytical solutions in **Figure 1E** were calculated from **Equations S41 and S43**.

In **Figure 2A**: $a_i = 0.02$, $a'_i = 0.025$, $d_i = 0.7$, $d'_i = 0.7$, $w_i = 0.4$, $k_i = 0.05$, $(i = 1, 2)$, $D_1 = 0.0160$, $D_2 = 0.0171$, $\kappa = 2000$, $\zeta = 5.5$. In **Figure 2B–C**: $L = 100$, $r^{(C)} = 5$, $r^{(I)} = 5$, $v_{C_i} = 1$, $v_R = 0.1$, $a_i = 0.2010$, $a'_i = 0.2828$, $d_i = 0.7$, $d'_i = 0.8$, $w_i = 0.33$, $k_i = 0.2$, $(i = 1, 2)$, $D_1 = 0.0605$, $D_2 = 0.0600$, $\kappa = 1000$. In **Figure 2D**: $a_i = 0.3$, $a'_i = 0.33$, $w_i = 0.018$, $k_i = 4.8$, $d_i = 5.5$, $d'_i = 5$, $(i = 1, 2)$, $D_1 = 0.011$, $D_2 = 0.010$, $\kappa = 10000$, $\zeta = 35$. In **Figure 2E**: $a_i = 0.2$, $a'_i = 0.24$, $d_i = 4.5$, $d'_i = 4$, $w_i = 0.02$, $k_i = 4.5$, $(i = 1, 2)$, $D_1 = 0.0120$, $D_2 = 0.010$, $\kappa = 10000$, $\zeta = 35$. In **Figure 2D and E**: We set $\tau = 0.4$ Day (see Appendix 7). This results in an expected lifespan of *Drosophila serrata* in the model settings of $\tau/D_2 = 40$ days and that of *Drosophila pseudoobscura* $\tau/D_1 = 36.4$ days, which roughly agrees with experimental data showing that the average lifespan of *D. serrata* is 34 days for males and 54 days for females (**Narayan et al., 2022**), and the average lifespan of *D. pseudoobscura* is around 40 days for females (**Gowaty et al., 2010**). The time averages ($\bar{C}_i$) and standard deviations ($\delta C_i$) of the species' relative/absolute abundances for the experimental data or SSA results are as follows: $^{(R)}\bar{C}^{\mathrm{Exp\,(SSA)}}_{D.serrata\_AR\_Grp1} = 0.53\,(0.55)$, $\delta^{(R)}C^{\mathrm{Exp\,(SSA)}}_{D.serrata\_AR\_Grp1} = 0.12\,(0.09)$, $^{(R)}\bar{C}^{\mathrm{Exp\,(SSA)}}_{D.serrata\_AR\_Grp2} = 0.59\,(0.61)$, $\delta^{(R)}C^{\mathrm{Exp\,(SSA)}}_{D.serrata\_AR\_Grp2} = 0.10\,(0.12)$, $\bar{C}^{\mathrm{Exp\,(SSA)}}_{T.confusum\_24°C} = 29.1\,(28.6)$, $\delta C^{\mathrm{Exp\,(SSA)}}_{T.confusum\_24°C} = 5.4\,(5.2)$, $\bar{C}^{\mathrm{Exp\,(SSA)}}_{T.castaneuma\_24°C} = 45.9\,(54.5)$, where the superscript '(R)' represents relative abundances. A comparison of Shannon entropies in the time series between experimental data and SSA results is presented in **Appendix 1—figure 7C and D**. **Figure 2A–E** were simulated from **Equations 1, 2, and 4**. See **Appendix 1—figure 7E and G** for the long-term time series of all species in **Figure 2D and E**, respectively.

Model settings in **Figure 3A–B and D** (plankton): $a_{il} = 0.1$, $a'_{il} = 0.125$, $d_{il} = 0.5$, $d'_i = 0.2$, $w_{il} = 0.3$, $k_{il} = 0.2$, $\kappa_1 = 8 \times 10^4$, $\kappa_2 = 5 \times 10^4$, $\kappa_3 = 3 \times 10^4$, $\zeta_1 = 280$, $\zeta_2 = 200$, $\zeta_3 = 150$, $D_i = 0.03 \times \mathcal{N}(1, 0.25)$, $(i = 1, \cdots, S_C, l = 1, \cdots, S_R)$, $S_C = 140$ and $S_R = 3$. Model settings in **Figure 3C** (bird): $a_i = 0.1$, $a'_i = 0.125$, $d_i = 0.5$, $d'_i = 0.5$, $w_i = 0.3$, $k_i = 0.2$, $\kappa = 10^5$, $\zeta = 110$, $D_i = 0.02 \times \mathcal{N}(1, 0.28)$, $(i = 1, \cdots, S_C)$, $S_C = 250$ and $S_R = 1$. Model settings in **Figure 3C** (fish): $a_i = 0.1$, $a'_i = 0.14$, $d_i = 0.5$, $d'_i = 0.5$, $w_i = 0.2$, $k_i = 0.1$, $\kappa = 10^6$, $\zeta = 550$, $D_i = 0.015 \times \mathcal{N}(1, 0.32)$, $(i = 1, \cdots, 45)$, $S_C = 45$ and $S_R = 1$. Model settings in **Figure 3C** (butterfly): $a_i = 0.1$, $a'_i = 0.125$, $d_i = 0.5$, $d'_i = 0.3$, $w_i = 0.3$, $k_i = 0.2$, $\kappa = 10^5$, $\zeta = 300$, $D_i = 0.034 \times \mathcal{N}(1, 0.35)$, $(i = 1, \cdots, S_C)$, $S_C = 150$ and $S_R = 1$. Model settings in **Figure 3D** (bat): $a_i = 0.1$, $a'_i = 0.125$, $d_i = 0.5$, $d'_i = 0.5$, $w_i = 0.2$, $k_i = 0.1$, $\kappa = 10^6$, $\zeta = 250$, $D_i = 0.013 \times \mathcal{N}(1, 0.34)$, $(i = 1, \cdots, S_C)$, $S_C = 40$ and $S_R = 1$. Model settings in **Figure 3D** (lizard): $a_i = 0.1$, $a'_i = 0.125$, $d_i = 0.5$, $d'_i = 0.5$, $w_i = 0.2$, $k_i = 0.1$, $\kappa = 10^6$, $\zeta = 250$, $D_i = 0.014 \times \mathcal{N}(1, 0.34)$, $(i = 1, \cdots, S_C)$, $S_C = 55$ and $S_R = 1$. In **Figure 3A–D**, the mortality rate $D_i$ is the only parameter that varies with the consumer species, which was randomly sampled from a Gaussian distribution $\mathcal{N}(\mu, \sigma)$, where $\mu$ and $\sigma$ are the mean and standard deviation of the distribution. The coefficient of variation of the mortality rates (i.e. $\sigma/\mu$) was chosen to be around 0.3, or more precisely, the best-fit in the range of 0.15–0.43. This range was estimated from experimental results (**Menon et al., 2003**) using the two-sigma rule. These settings for the mortality rates also apply to those in **Appendix 1—figures 8–14**. **Figure 3A–D** were simulated from **Equations 1, 2 and 4**. See **Appendix 1—figures 10K, C, D, H, I, J and 14C**, **Figure 3A and B** for the time series of **Figure 3C** ($\mathrm{ODEs}^{S_R=1}_{\mathrm{bird}}$), **Figure 3C** ($\mathrm{ODEs}^{S_R=1}_{\mathrm{butterfly}}$), **Figure 3C** ($\mathrm{ODEs}^{S_R=1}_{\mathrm{fish}}$), **Figure 3D** ($\mathrm{ODEs}^{S_R=1}_{\mathrm{bat}}$), **Figure 3D** ($\mathrm{SSA}^{S_R=1}_{\mathrm{bat}}$), **Figure 3D** ($\mathrm{ODEs}^{S_R=1}_{\mathrm{lizard}}$), **Figure 3D** ($\mathrm{SSA}^{S_R=1}_{\mathrm{lizard}}$), **Figure 3D** ($\mathrm{ODEs}^{S_R=3}_{\mathrm{plankton}}$) and **Figure 3D** ($\mathrm{SSA}^{S_R=3}_{\mathrm{plankton}}$), respectively. The Shannon entropies of the experimental data and simulation results for each ecological community are: $H^{\mathrm{bird(1982)}}_{\mathrm{Exp(ODEs)}} = 5.67(6.79)$, $H^{\mathrm{bird(2018)}}_{\mathrm{Exp(ODEs)}} = 6.63(6.79)$, $H^{\mathrm{butterfly}}_{\mathrm{Exp(ODEs)}} = 4.78(4.12)$, $H^{\mathrm{fish}}_{\mathrm{Exp(ODEs)}} = 3.78(3.40)$, $H^{\mathrm{lizard}}_{\mathrm{Exp(ODEs,SSA)}} = 4.05(3.57, 3.50)$. Here the Shannon entropy $H = -\sum_{i=1}^{S_C} \mathcal{P}_i \log_2(\mathcal{P}_i)$, where $\mathcal{P}_i$ is the probability that a consumer individual belongs to species $C_i$.

