## [Editor Report · eLife assessment]

This manuscript is an **important** contribution, assessing the role of intraspecific consumer interference in maintaining diversity using a mathematical model. Consistent with long-standing ecological theory, the authors **convincingly** show that predator interference allows for the coexistence of multiple species on a single resource, beyond the competitive exclusion principle. Notably, the model matches observed rank-abundance curves in several natural ecosystems.

---

## [Referee Report · Reviewer #1 (Public Review)]

Summary:

The manuscript considers a mechanistic extension of MacArthur's consumer-resource model to include chasing down of food and potential encounters between the chasers (consumers) that lead to less efficient feeding in the form of negative feedback. After developing the model, a deterministic solution and two forms of stochastic solutions are presented, in agreement with each other. Finally, the model is applied to explain observed coexistence and rank-abundance data.

Strengths:

- The application of the theory to natural rank-abundance curves is impressive.

- The comparison with the experiments that reject the competitive exclusion principle is promising. It would be fascinating to see if in, e.g. insects, the specific interference dynamics could be observed and quantified and whether they would agree with the model.

- The results are clearly presented; the methods adequately described; the supplement is rich with details.

- There is much scope to build upon this expansion of the theory of consumer-resource models. This work can open up new avenues of research.

Weaknesses:

- Though more and better data could be used to constrain and validate the modeling, given this is a theory-driven manuscript, their results are sufficient.

---

## [Referee Report · Reviewer #3 (Public Review)]

Summary:

In this manuscript, the authors extend previous work on the role of predator interference in species coexistence. Previous theoretical work (for example, using the Beddington-DeAngelis model) has shown that predator interference allows for multiple predators to coexist on the same prey. While the Beddington-DeAngelis has been influential in theoretical ecology, it has also been criticized at times for several unusual assumptions, most critically, that predators interfere with each other regardless of whether they are already engaged in another interaction. There has been considerable work since then which has sought either to find sets of assumptions that lead to the B-D equation or to derive alternative equations from a more realistic set of assumptions (Ruxton et al. 1992; Cosner et al. 1999; Broom et al. 2010; Geritz and Gyllenberg 2012). This paper represents another effort to more rigorously derive a model of predator interference by borrowing concepts from chemical reaction kinetics (the approach is similar to previous work: Ruxton et al. 1992). The main point of difference is that the model in the current manuscript allows for 'chasing pairs', where a predator and prey engage with one another to the exclusion of other interactions, a situation Ruxton et al. (1992) do not consider. While the resulting functional response is quite complex, the authors show that under certain conditions, one can get an analytical expression for the functional response of a predator as a function of predator and resource densities. They then go on to show that including intraspecific interference allows for the coexistence of multiple species on one or a few resources, and demonstrate that this result is robust to demographic stochasticity. This work provides additional support for the idea that predator interference allows multiple predators to persist with a shared resource.

Strengths:

I appreciate the effort to rigorously derive interaction rates from models of individual behaviors. As currently applied, functional responses (FRs) are estimated by fitting equations to feeding rate data across a range of prey or predator densities. In practice, such experiments are only possible for a limited set of species. This is problematic because whether a particular FR allows stability or coexistence depends on not just its functional form, but also its parameter values. The promise of the approach taken here is that one might be able to derive the functional response parameters of a particular predator species from species traits or more readily measurable behavioural data.

Weaknesses:

The main weakness of this paper is that while it is technically sound, it doesn't change the fundamental intuition gained from more phenomenological models of predator interference: as one species becomes more common, it limits its own growth (manifested by less time spent searching for/handing resources due to interference), such that it does not exclude the existence of a competitor species. However, given the authors use a different model formulation that has been used in past studies, it suggests that predator interference will likely tend to promote coexistence regardless of some of the technical details in how it is formulated in a model.

The formulation of chasing-pair engagements assumes that prey being chased by a predator are unavailable to other predators. While this may hold in some predator-prey, it does not hold for many others, perhaps limiting some results' generality.

Summary:

The manuscript by Kang et al investigates how the consideration of pairwise encounters (consumer-resource chasing, intraspecific consumer pair, and interspecific consumer pair) influences the community assembly results. To explore this, they presented a new model that considers pairwise encounters and intraspecific interference among consumer individuals, which is an extension of the classical Beddington-DeAngelis (B-D) phenomenological model, incorporating detailed considerations of pairwise encounters and intraspecific interference among consumer individuals. Later, they connected with several experimental datasets.

Strengths:

They found that the negative feedback loop created by the intraspecific interference allows a diverse range of consumer species to coexist with only one or a few types of resources. Additionally, they showed that some patterns of their model agree with experimental data, including time-series trajectories of two small in-lab community experiments and the rank-abundance curves from several natural communities. The presented results here are interesting and present another way to explain how the community overcomes the competitive exclusion principle.

Weaknesses:

The authors did a great job of satisfactorily addressing each of my concerns raised in the previous round. I did not detect additional weaknesses.

---

## [Author Response]

The following is the authors’ response to the original reviews.

**Reviewer #1 (Public Review):**
Summary:The manuscript considers a mechanistic extension of MacArthur's consumer-resource model to include chasing down food and potential encounters between the chasers (consumers) that lead to less efficient feeding in the form of negative feedback. After developing the model, a deterministic solution and two forms of stochastic solutions are presented, in agreement with each other. Finally, the model is applied to explain observed coexistence and rank-abundance data.

We thank the reviewer for the accurate summary of our manuscript.

Strengths:The application of the theory to natural rank-abundance curves is impressive. The comparison with the experiments that reject the competitive exclusion principle is promising. It would be fascinating to see if in, e.g. insects, the specific interference dynamics could be observed and quantified and whether they would agree with the model.The results are clearly presented; the methods adequately described; the supplement is rich with details.There is much scope to build upon this expansion of the theory of consumer-resource models. This work can open up new avenues of research.

We appreciate the reviewer for the very positive comments. We have followed many of the suggestions raised by the reviewer, and the manuscript is much improved as a result.

Following the reviewer’s suggestions, we have now used Shannon entropies to quantify the model comparison with experiments that reject the Competitive Exclusion Principle (CEP). Specifically, for each time point of each experimental or model-simulated community, we calculated the Shannon entropies using the formula:

H(t)=−∑i=1ScPi(t)log2⁡(Pi(t)), where Pi(t) is the probability that a consumer individual belongs to species C_i_ at the time stamp of t. The comparison of Shannon entropies in the time series between those of the experimental data and SSA results shown in Fig. 2D-E is presented in Appendix-fig. 7C-D. The time averages (H¯) and standard deviations (δH) of the Shannon entropies for these experimental or SSA model-simulated communities are as follows:

H¯Exp(SSA)DrosplaARGip1=0.95(0.97), δHExp(SSA)DrasophidaARGipl=0.06(0.04); H¯Exp(SSA)−DrasospGip2=0.94(0.92),

δHExp(SSA)2Tribolimu =0.02(0.05), ,

Meanwhile, we have calculated the time averages (C¯i) and standard deviations (δ*C_i_)* of the species’ relative/absolute abundances for the experimental or SSA model-simulated communities shown in Fig. 2D-E, which are as follows:

C(R)CD.serrataExp(SR_Cip11)¯=0.53(0.55), δ(R)CD serrataAR_Grpl Exp(SA) =0.12(0.09); C(R)CD.errata_AR_Gip2Exp(SA)¯=0.59(0.61), δ(R)CD serrata AR_Gip2 2Exp(SSA =0.10(0.12); CT.casstaneuma 24∘CExp(SSA)¯=45.9(54.5), δCT.castaneuma_24 Exp⁡C=7.2(8.6), where the superscript “(R)” represents relative abundances.

From the results of Shannon entropies shown in Author response image 1 (which are identical to those of Appendix-fig. 7C-D) and the quantitative comparison of the time average and standard deviation between the model and experiments presented above, it is evident that the model results in Fig. 2D-E exhibit good consistency with the experimental data. They share roughly identical time averages and standard deviations in both Shannon entropies and the species' relative/absolute abundances for most of the comparisons. All these analyses are included in the appendices and mentioned in the main text.

**Author response image 1. sa3fig1:** Shannon Entropies of the experimental data and SSA results in Fig. 2D-E, redrawn from Appendix-fig. 7C-D.

Weaknesses:I am questioning the use of carrying capacity (Eq. 4) instead of using nutrient limitation directly through Monod consumption (e.g. Posfai et al. who the authors cite). I am curious to see how these results hold or are changed when Monod consumption is used.

We thank the reviewer for raising this question. To explain it more clearly, the equation combining the third equation in Eq. 1 and Eq. 4 of our manuscript is presented below as Eq. R1:

R˙l={ηlRl(1−Rl/κl)−∑i=1sckllxil (for biotic resources) ζl(1−Rl/κl)−∑i=1sckilxil (for abiotic resources) 

where x_il_ represents the population abundance of the chasing pair C_i_^(P)^ ∨ R_l_^(P)^, κ_l_ stands for the steady-state population abundance of species R_l_ (the carrying capacity) in the absence of consumer species. In the case with no consumer species, then x_il_ = 0 since C_i_ = 0 (*i*=1,…,S_C_), thus R_l_ = κ_l_ when R_l_ = 0.

Eq. R1 for the case of abiotic resources is comparable to Eq. (1) in Posfai et al., which we present below as Eq. R2:

dcidt=si−(∑σnσ(t)ασi)ri(ci)−μici(t),

where c_i_ represents the concentration of nutrient *i*, and thus corresponds to our R_l_ ; n_σ_(t) is the population of species σ, which corresponds to our C_i_ ; s_i_ stands for the nutrient supply rate, which corresponds to our *ζl ; µi* denotes the nutrient loss rate, corresponding to our ζl/κl;ασi is the coefficient of the rate of species σ for consuming nutrient i, which corresponds to our kil;nσ(t)⋅ri(ci) in Posfai et al. is the consumption rate of nutrient *i* by the population of species σ, which corresponds to our *x_il_.*

In Posfai et al., ri(ci) is the Monod function: ri(ci)=ci/(Ki+ci) and thus

nσ(t)⋅ri(ci)=nσ(t)ci/(Ki+ci).

In our model, however, since predator interference is not involved in Posfai et al., we need to analyze the form of x_il_ presented in the functional form of x_il_ ({R_l_},{C_i_}) in the case involving only chasing pairs. Specifically, for the case of abiotic resources, the population dynamics can be described by Eq. 1 combined with Eq. R1:

{x˙il=aiiCi(F)Rl(F)−(kil+dil)xil,C˙i=∑l=1sRwilkilxil−DiCi,i=1,⋯,SC,R˙l=ζl(1−Rl/κl)−∑i=1sckilxil,l=1,⋯,SR,

where Ci=Ci(F)+∑Ixil and Rl=Rl(F)+∑ixil. For convenience, we consider the case of S_R_ = 1 where the Monod form was derived (Monod, J. (1949). Annu. Rev. Microbiol., 3, 371-394.). From dxi/dt=0, we have

xil=Ci(F)Ri(F)/Ku,

where Kil=kil+dilail, and *l* = 1. If the population abundance of the resource species is much larger than that of all consumer species (i.e., Rl≫∑i=1SCCi), then,

Rl≫∑i=1SCCi>∑i=1SCxi and *R_l_^(F)^ ≈ R_l_.* Combined with R5, and noting that *C_i_ = C_i_(F) + xil* we can solve for *x_il_ :*

xil=CiRl/(Kil+Rl), 

with *l* = 1 since S_R_ = 1. Comparing Eq. R6 with Eq. R3, and considering the symbol correspondence explained in the text above, it is now clear that our model can be reduced to the Monod consumption form in the case of S_R_ = 1 where the Monod form was derived from.

Following on the previous comment, I am confused by the fact that the nutrient consumption term in Eq. 1 and how growth is modeled (Eq. 4) are not obviously compatible and would be hard to match directly to experimentally accessible quantities such as yield (nutrient to biomass conversion ratio). Ultimately, there is a conservation of mass ("flux balance"), and therefore the dynamics must obey it. I don't quite see how conservation of mass is imposed in this work.

We thank the reviewer for raising this question. Indeed, the population dynamics of our model must adhere to flux balance, with the most pertinent equation restated here as Eq. R7:

{C˙i=∑l=1SRwilkilxil−DiCi,i=1,⋯,SC,R˙l=ζl(1−Rl/κl)−∑i=1sCkilxil,l=1,⋯,SR

Below is the explanation of how Eq. R7, and thus Eqs. 1 and 4 of our manuscript, adhere to the constraint of flux balance. The interactions and fluxes between consumer and resource species occur solely through chasing pairs. At the population level, the scenario of chasing pairs among consumer species C_i_ and resource species R_l_ is presented in the follow expression:Ci(F)+Rl(F)⇄auduCi(P)∨Rl(P)→ku⟶Ci(F)(+)

where the superscripts "(F)" and "(P)" represent the freely wandering individuals and those involved in chasing pairs, respectively, "(+)" stands for the gaining biomass of consumer C_i_ from resource R_l_. In our manuscript, we use x_l_ to represent the population abundance (or equivalently, the concentration, for a well-mixed system with a given size) of the chasing pair C_i_^(P)^ ∨ R_l_^(P)^, and thus, the net flow from resource species R_l_ to consumer species C_i_ per unit time is k_il_x_il_. Noting that there is only one R_l_ individual within the chasing pair C_i_^(P)^ ∨ R_l_^(P)^, then the net effect on the population dynamics of species .dRl/dt is −k_il_x_il_. However, since a consumer individual from species C_i_ could be much heavier than a species R_l_ individual, and energy dissipation would be involved from nutrient conversion into biomass, we introduce a mass conversion ratio w_l_ in our manuscript. For example, if a species C_i_ individual is ten times the weight of a species R_l_ individual, without energy dissipation, the mass conversion ratio wil should be 1/10 (i.e., wil = 0.1), however, if half of the chemical energy is dissipated into heat from nutrient conversion into biomass, then w_l_ = 0.1 0.5× = 0.05. Consequently, the net effect of the flux from resource species _R_l to consumer species C_i_ per unit time on the population dynamics dCi/dt is wilkilxil , and flux balance is clearly satisfied.

For the population dynamics of a consumer species C_i_, we need to consider all the biomass influx from different resource species, and thus there is a summation over all species of resources, which leads to the term of “∑l=1SRwilkilxil” in Eq. R7. Similarly, for the population dynamics of a resource species R_l_, we need to lump sum all the biomass outflow into different consumer species, resulting in the term of “−∑i=1SCkilxil” in Eq. R7.

Consequently, Eq. R7 and our model satisfy the constraint of flux balance.

These models could be better constrained by more data, in principle, thereby potential exists for a more compelling case of the relevance of this interference mechanism to natural systems.

We thank the reviewer for raising this question. Indeed, our model could benefit from the inclusion of more experimental data. In our manuscript, we primarily set the parameters by estimating their reasonable range. Following the reviewer's suggestions, we have now specified the data we used to set the parameters. For example, in Fig. 2D, we set 𝐷_2_=0.01 with τ=0.4 days, resulting in an expected lifespan of *Drosophila* serrata in our model setting of 𝜏⁄𝐷_2_ = 40 days, which roughly agrees with experimental data showing that the average lifespan of D. serrata is 34 days for males and 54 days for females (lines 321-325 in the appendices; reference: Narayan et al. J Evol Biol. 35: 657–663 (2022)). To explain biodiversity and quantitatively illustrate the rank-abundance curves across diverse communities, the competitive differences across consumer species, exemplified by the coefficient of variation of the mortality rates - a key parameter influencing the rank-abundance curve, were estimated from experimental data in the reference article (Patricia Menon et al., Water Research (2003) 37, 4151) using the two-sigma rule (lines 344-347 in the appendices).

Still, we admit that many factors other than intraspecific interference, such as temporal variation, spatial heterogeneity, etc., are involved in breaking the limits of CEP in natural systems, and it is still challenging to differentiate each contribution in wild systems. However, for the two classical experiments that break CEP (Francisco Ayala, 1969; Thomas Park, 1954), intraspecific interference could probably be the most relevant mechanism, since factors such as temporal variation, spatial heterogeneity, cross-feeding, and metabolic tradeoffs are not involved in those two experimental systems.

The underlying frameworks, B-D and MacArthur are not properly exposed in the introduction, and as a result, it is not obvious what is the specific contribution in this work as opposed to existing literature. One needs to dig into the literature a bit for that.The specific contribution exists, but it might be more clearly separated and better explained. In the process, the introduction could be expanded a bit to make the paper more accessible, by reviewing key features from the literature that are used in this manuscript.

We thank the reviewer for these very insightful suggestions. Following these suggestions, we have now added a new paragraph and revised the introduction part of our manuscript (lines 51-67 in the main text) to address the relevant issues. Our paper is much improved as a result.

**Reviewer #2 (Public Review):**
Summary:The manuscript by Kang et al investigates how the consideration of pairwise encounters (consumer-resource chasing, intraspecific consumer pair, and interspecific consumer pair) influences the community assembly results. To explore this, they presented a new model that considers pairwise encounters and intraspecific interference among consumer individuals, which is an extension of the classical Beddington-DeAngelis (BD) phenomenological model, incorporating detailed considerations of pairwise encounters and intraspecific interference among consumer individuals. Later, they connected with several experimental datasets.Strengths:They found that the negative feedback loop created by the intraspecific interference allows a diverse range of consumer species to coexist with only one or a few types of resources. Additionally, they showed that some patterns of their model agree with experimental data, including time-series trajectories of two small in-lab community experiments and the rank-abundance curves from several natural communities. The presented results here are interesting and present another way to explain how the community overcomes the competitive exclusion principle.

We appreciate the reviewer for the positive comments and the accurate summary of our manuscript.

Weaknesses:The authors only explore the case with interspecific interference or intraspecific interference exists. I believe they need to systematically investigate the case when both interspecific and intraspecific interference exists. In addition, the text description, figures, and mathematical notations have to be improved to enhance the article's readability. I believe this manuscript can be improved by addressing my comments, which I describe in more detail below.

We thank the reviewer for these valuable suggestions. We have followed many of the suggestions raised by the reviewer, and the manuscript is much improved as a result.

(1) In nature, it is really hard for me to believe that only interspecific interference or intraspecific interference exists. I think a hybrid between interspecific interference and intraspecific interference is very likely. What would happen if both the interspecific and intraspecific interference existed at the same time but with different encounter rates? Maybe the authors can systematically explore the hybrid between the two mechanisms by changing their encounter rates. I would appreciate it if the authors could explore this route.

We thank the reviewer for raising this question. Indeed, interspecific interference and intraspecific interference simultaneously exist in real cases. To differentiate the separate contributions of inter- and intra-specific interference on biodiversity, we considered different scenarios involving inter- or intra-specific interference. In fact, we have also considered the scenario involving both inter- and intra-specific interference in our old version for the case of S_C_ = 2 and S_R_ = 1, where two consumer species compete for one resource species (Appendix-fig. 5, and lines 147-148, 162-163 in the main text of the old version, or lines 160-161, 175-177 in the new version).

Following the reviewer’s suggestions, we have now systematically investigated the cases of S_C_ = 6, S_R_ = 1, and S_C_ = 20, S_R_ = 1, where six or twenty consumer species compete for one resource species in scenarios involving chasing pairs and both inter- and intra-specific interference using both ordinary differential equations (ODEs) and stochastic simulation algorithm (SSA). These newly added ODE and SSA results are shown in Appendix-fig. 5 F-H, and we have added a new paragraph to describe these results in our manuscript (lines 212-215 in the main text). Consistent with our findings in the case of S_C_ = 2 and S_R_ = 1, the species coexistence behavior in the cases of both S_C_ = 6, S_R_ = 1, and S_C_ = 20, S_R_ = 1 is very similar to those without interspecific interference: all consumer species coexist with one type of resources at constant population densities in the ODE studies, and the SSA results fluctuate around the population dynamics of the ODEs.

As for the encounter rates of interspecific and intraspecific interference, in fact, in a well-mixed system, these encounter rates can be derived from the mobility rates of the consumer species using the mean field method. For a system with a size of *L2*, the interspecific encounter rate between consumer species C_i_ and C_j_ (i ≠ j) is aij′=2r(1)L−2vCi2+vCJ2 please refer to lines 100-102, 293-317 in the main text, and see also Appendix-fig. 1, where r^(I)^ is the upper distance for interference, while v_Ci_ and v_Cj_ represent the mobility rates of species C_i_ and C_j_, respectively. Meanwhile, the intraspecific encounter rates within species C_i_ and species C_j_ are aii′=22vCir(1)L−2 and aij′=22vCjr(1)L−2, respectively.

Thus, once the intraspecific encounter rates a’_ii_ are a’_jj_ given, the interspecific encounter rate between species C_i_ and C_j_ is determined. Consequently, we could not tune the encounter rates of interspecific and intraspecific interference at will in our study, especially noting that for clarity reasons, we have used the mortality rate as the only parameter that varies among the consumer species throughout this study. Alternatively, we have made a systematic study on analyzing the influence of varying the separate rate and escape rate on species coexistence in the case of two consumers competing for a single type of resources (see Appendix-fig. 5A).

(2) In the first two paragraphs of the introduction, the authors describe the competitive exclusion principle (CEP) and past attempts to overcome the CEP. Moving on from the first two paragraphs to the third paragraph, I think there is a gap that needs to be filled to make the transition smoother and help readers understand the motivations. More specifically, I think the authors need to add one more paragraph dedicated to explaining why predator interference is important, how considering the mechanism of predator interference may help overcome the CEP, and whether predator interference has been investigated or under-investigated in the past. Then building upon the more detailed introduction and movement of predator interference, the authors may briefly introduce the classical B-D phenomenological model and what are the conventional results derived from the classical B-D model as well as how they intend to extend the B-D model to consider the pairwise encounters.

We thank the reviewer for these very insightful suggestions. Following these suggestions, we have added a new paragraph and revised the introduction part of our paper (lines 51-67 in the main text). Our manuscript is significantly improved as a result.

(3) The notations for the species abundances are not very informative. I believe some improvements can be made to make them more meaningful. For example, I think using Greek letters for consumers and English letters for resources might improve readability. Some sub-scripts are not necessary. For instance, R^(l)_0 can be simplified to g_l to denote the intrinsic growth rate of resource l. Similarly, K^(l)_0 can be simplified to K_l. Another example is R^(l)_a, which can be simplified to s_l to denote the supply rate. In addition, right now, it is hard to find all definitions across the text. I would suggest adding a separate illustrative box with all mathematical equations and explanations of symbols.

We thank the reviewer for these very useful suggestions. We have now followed many of the suggestions to improve the readability of our manuscript. Given that we have used many English letters for consumers and there are already many symbols of English and Greek letters for different variables and parameters in the appendices, we have opted to use Greek letters for parameters specific to resource species and English letters for those specific to consumer species. Additionally, we have now added Appendix-tables 1-2 in the appendices (pages 16-17 in the appendices) to illustrate the symbols used throughout our manuscript.

(4) What is the f_i(R^(F)) on line 131? Does it refer to the growth rate of C_i? I noticed that f_i(R^(F)) is defined in the supplementary information. But please ensure that readers can understand it even without reading the supplementary information. Otherwise, please directly refer to the supplementary information when f_i(R^(F)) occurs for the first time. Similarly, I don't think the readers can understand \Omega^\prime_i and G^\prime_i on lines 135-136.

We thank the reviewer for raising these questions. We apologize for not illustrating those symbols and functions clearly enough in our previous version of the manuscript. *f*_*i*_*⟮R*^*(F)*^*⟯* is a function of the variable *R*^*(F)*^ with the index *i,* which is defined as f1(R(F))=R(F)R(F)+K1 and f2(R(F))=R(F)R(F)+K2
*for i=2.* Following the reviewer’s suggestions, we have now added clear definitions for symbols and functions and resolved these issues. The definitions of \Omega_i, \Omega^\prime_i, G, and G^\prime are overly complex, and hence we directly refer to the Appendices when they occur for the first time in the main text.

**Reviewer #3 (Public Review):**
Summary:A central question in ecology is: Why are there so many species? This question gained heightened interest after the development of influential models in theoretical ecology in the 1960s, demonstrating that under certain conditions, two consumer species cannot coexist on the same resource. Since then, several mechanisms have been shown to be capable of breaking the competitive exclusion principle (although, we still lack a general understanding of the relative importance of the various mechanisms in promoting biodiversity).One mechanism that allows for breaking the competitive exclusion principle is predator interference. The Beddington-DeAngelis is a simple model that accounts for predator interference in the functional response of a predator. The B-D model is based on the idea that when two predators encounter one another, they waste some time engaging with one another which could otherwise be used to search for resources. While the model has been influential in theoretical ecology, it has also been criticized at times for several unusual assumptions, most critically, that predators interfere with each other regardless of whether they are already engaged in another interaction. However, there has been considerable work since then which has sought either to find sets of assumptions that lead to the B-D equation or to derive alternative equations from a more realistic set of assumptions (Ruxton et al. 1992; Cosner et al. 1999; Broom et al. 2010; Geritz and Gyllenberg 2012). This paper represents another attempt to more rigorously derive a model of predator interference by borrowing concepts from chemical reaction kinetics (the approach is similar to previous work: Ruxton et al. 1992). The main point of difference is that the model in the current manuscript allows for 'chasing pairs', where a predator and prey engage with one another to the exclusion of other interactions, a situation Ruxton et al. (1992) do not consider. While the resulting functional response is quite complex, the authors show that under certain conditions, one can get an analytical expression for the functional response of a predator as a function of predator and resource densities. They then go on to show that including intraspecific interference allows for the coexistence of multiple species on one or a few resources, and demonstrate that this result is robust to demographic stochasticity.

We thank the reviewer for carefully reading our manuscript and for the positive comments on the rigorously derived model of predator interference presented in our paper. We also appreciate the reviewer for providing a thorough introduction to the research background of our study, especially the studies related to the BeddingtonDeAngelis model. We apologize for our oversight in not fully appreciating the related study by Ruxton et al. (1992) at the time of our first submission. Indeed, as suggested by the reviewer, Ruxton et al. (1992) is relevant to our study in that we both borrowed concepts from chemical reaction kinetics. Now, we have reworked the introduction and discussion sections of our manuscript, cited, and acknowledged the contributions of related works, including Ruxton et al. (1992).

Strengths:I appreciate the effort to rigorously derive interaction rates from models of individual behaviors. As currently applied, functional responses (FRs) are estimated by fitting equations to feeding rate data across a range of prey or predator densities. In practice, such experiments are only possible for a limited set of species. This is problematic because whether a particular FR allows stability or coexistence depends on not just its functional form, but also its parameter values. The promise of the approach taken here is that one might be able to derive the functional response parameters of a particular predator species from species traits or more readily measurable behavioral data.

We appreciate the reviewer's positive comments regarding the rigorous derivation of our model. Indeed, all parameters of our model can be derived from measurable behavioral data for a specific set of predator species.

Weaknesses:The main weakness of this paper is that it devotes the vast majority of its length to demonstrating results that are already widely known in ecology. We have known for some time that predator interference can relax the CEP (e.g., Cantrell, R. S., Cosner, C., & Ruan, S. 2004).While the model presented in this paper differs from the functional form of the B-D in some cases, it would be difficult to formulate a model that includes intraspecific interference (that increases with predator density) that does not allow for coexistence under some parameter range. Thus, I find it strange that most of the main text of the paper deals with demonstrating that predator interference allows for coexistence, given that this result is already well known. A more useful contribution would focus on the extent to which the dynamics of this model differ from those of the B-D model.

We appreciate the reviewer for raising this question and apologize for not sufficiently clarifying the contribution of our manuscript in the context of existing knowledge upon our initial submission. We have now significantly revised the introduction part of our manuscript (lines 51-67 in the main text) to make this clearer. Indeed, with the application of the Beddington-DeAngelis (B-D) model, several studies (e.g., Cantrell, R. S., Cosner, C., & Ruan, S. 2004) have already shown that intraspecific interference promotes species coexistence, and it is certain that the mechanism of intraspecific interference could lead to species coexistence if modeled correctly. However, while we acknowledge that the B-D model is a brilliant phenomenological model of intraspecific interference, for the specific research topic of our manuscript on breaking the CEP and explaining the paradox of the plankton, it is highly questionable regarding the validity of applying the B-D model to obtain compelling results.

Specifically, the functional response in the B-D model of intraspecific interference can be formally derived from the scenario involving only chasing pairs without consideration of pairwise encounters between consumer individuals (Eq. S8 in Appendices; related references: Gert Huisman, Rob J De Boer, J. Theor. Biol. 185, 389 (1997) and Xin Wang and Yang-Yu Liu, iScience 23, 101009 (2020)). Since we have demonstrated that the scenario involving only chasing pairs is under the constraint of CEP (see lines 139-144 in the main text and Appendix-fig. 3A-C; related references: Xin Wang and Yang-Yu Liu, iScience 23, 101009 (2020)), and given the identical functional response mentioned above, it is thus highly questionable regarding the validity of the studies relying on the B-D model to break CEP or explain the paradox of the plankton.

Consequently, one of the major objectives of our manuscript is to resolve whether the mechanism of intraspecific interference can truly break CEP and explain the paradox of the plankton in a rigorous manner. By modeling intraspecific predator interference from a mechanistic perspective and applying rigorous mathematical analysis and numerical simulations, our work resolves these issues and demonstrates that intraspecific interference enables a wide range of consumer species to coexist with only one or a handful of resource species. This naturally breaks CEP, explains the paradox of plankton, and quantitatively illustrates a broad spectrum of experimental results.

For intuitive understanding, we introduced a functional response in our model (presented as Eq. 5 in the main text), which indeed involves approximations. However, to rigorously break the CEP or explain the paradox of plankton, all simulation results in our study were directly derived from equations 1 to 4 (main text), without relying on the approximate functional response presented in Eq. 5.

The formulation of chasing-pair engagements assumes that prey being chased by a predator are unavailable to other predators. For one, this seems inconsistent with the ecology of most predator-prey systems. In the system in which I work (coral reef fishes), prey under attack by one predator are much more likely to be attacked by other predators (whether it be a predator of the same species or otherwise). I find it challenging to think of a mechanism that would give rise to chased prey being unavailable to other predators. The authors also critique the B-D model: "However, the functional response of the B-D model involving intraspecific interference can be formally derived from the scenario involving only chasing pairs without predator interference (Wang and Liu, 2020; Huisman and De Boer, 1997) (see Eqs. S8 and S24). Therefore, the validity of applying the B-D model to break the CEP is questionable.".

We appreciate the reviewer for raising this question. We fully agree with the reviewer that in many predator-prey systems (e.g., coral reef fishes as mentioned by the reviewer, wolves, and even microbial species such as Myxococcus xanthus; related references: Berleman et al., FEMS Microbiol. Rev. 33, 942-957 (2009)), prey under attack by one predator can be targeted by another predator (which we term as a chasing triplet) or even by additional predator individuals (which we define as higher-order terms). However, since we have already demonstrated in a previous study (Xin Wang, Yang-Yu Liu, iScience 23, 101009 (2020)) from a mechanistic perspective that a scenario involving chasing triplets or higher-order terms can naturally break the CEP, while our manuscript focuses on whether pairwise encounters between individuals can break the CEP and explain the paradox of plankton, we deliberately excluded confounding factors that are already known to promote biodiversity, just as we excluded prevalent factors such as cross-feeding and temporal variations in our model.

However, the way "chasing pairs" are formulated does result in predator interference because a predator attacking prey interferes with the ability of other predators to encounter the prey. I don't follow the author's logic that B-D isn't a valid explanation for coexistence because a model incorporating chasing pairs engagements results in the same functional form as B-D.

We thank the reviewer for raising this question, and we apologize for not making this point clear enough at the time of our initial submission. We have now revised the related part of our manuscript (lines 56-62 in the main text) to make this clearer.

In our definition, predator interference means the pairwise encounter between consumer individuals, while a chasing pair is formed by a pairwise encounter between a consumer individual and a resource individual. Thus, in these definitions, a scenario involving only chasing pairs does not involve pairwise encounters between consumer individuals (which is our definition of predator interference).

We acknowledge that there can be different definitions of predator interference, and the reviewer's interpretation is based on a definition of predator interference that incorporates indirect interference without pairwise encounters between consumer individuals. We do not wish to argue about the appropriateness of definitions. However, since we have proven that scenarios involving only chasing pairs are under the constraint of CEP (see lines 139-144 in the main text and Appendix-fig. 3A-C; related references: Xin Wang and Yang-Yu Liu, iScience 23, 101009 (2020)), while the functional response of the B-D model can be derived from the scenario involving only chasing pairs without consideration of pairwise encounters between consumer individuals (Eq. S8 in Appendices; related references: Gert Huisman, Rob J De Boer, J. Theor. Biol. 185, 389 (1997) and Xin Wang and Yang-Yu Liu, iScience 23, 101009 (2020)), it is thus highly questionable regarding the validity of applying the B-D model to break CEP.

More broadly, the specific functional form used to model predator interference is of secondary importance to the general insight that intraspecific interference (however it is modeled) can allow for coexistence. Mechanisms of predator interference are complex and vary substantially across species. Thus it is unlikely that any one specific functional form is generally applicable.

We thank the reviewer for raising this issue. We agree that the general insight that intraspecific predator interference can facilitate species coexistence is of great importance. We also acknowledge that any functional form of a functional response is unlikely to be universally applicable, as explicit functional responses inevitably involve approximations. However, we must reemphasize the importance of verifying whether intraspecific predator interference can truly break CEP and explain the paradox of plankton, which is one of the primary objectives of our study. As mentioned above, since the B-D model can be derived from the scenario involving only chasing pairs (Eq. S8 in Appendices; related references: Gert Huisman, Rob J De Boer, J. Theor. Biol. 185, 389 (1997) and Xin Wang and Yang-Yu Liu, iScience 23, 101009 (2020)), while we have demonstrated that scenarios involving only chasing pairs are subject to the constraint of CEP (see lines 139-144 in the main text and Appendix-fig. 3A-C; related references: Xin Wang and Yang-Yu Liu, iScience 23, 101009 (2020)), it is highly questionable regarding the validity of applying the B-D model to break CEP.

**Recommendations for the authors:**

**Reviewer #1 (Recommendations For The Authors):**
I do not see any code or data sharing. They should exist in a prominent place. The authors should make their simulations and the analysis scripts freely available to download, e.g. by GitHub. This is always true but especially so in a journal like eLife.

We appreciate the reviewer for these recommendations. We apologize for our oversight regarding the unsuccessful upload of the data in our initial submission, as the data size was considerable and we neglected to double-check for this issue. Following the reviewer’s recommendation, we have now uploaded the code and dataset to GitHub (accessible at https://github.com/SchordK/Intraspecific-predator-interference-promotesbiodiversity-in-ecosystems), where they are freely available for download.

The introduction section should include more background, including about BD but also about consumer-resource models. Part of the results section could be moved/edited to the introduction. You should try that the results section should contain only "new" stuff whereas the "old" stuff should go in the introduction.

We thank the reviewer for these recommendations. Following these suggestions, we have now reorganized our manuscript by adding a new paragraph to the introduction section (lines 51-62 in the main text) and revising related content in both the introduction and results sections (lines 63-67, 81-83 in the main text).

I found myself getting a little bogged down in the general/formal description of the model before you go to specific cases. I found the most interesting part of the paper to be its second half. This is a dangerous strategy, a casual reader may miss out on the most interesting part of the paper. It's your paper and do what you think is best, but my opinion is that you could improve the presentation of the model and background to get to the specific contribution and specific use case quickly and easily, then immediately to the data. You can leave the more general formulation and the details to later in the paper or even the appendix. Ultimately, you have a simple idea and a beautiful application on interesting data-that is your strength I think, and so, I would focus on that.

We appreciate the reviewer for the positive comments and valuable suggestions. Following these recommendations, we have revised the presentation of the background information to clarify the contribution of our manuscript, and we have refined our model presentation to enhance clarity. Meanwhile, as we need to address the concerns raised by other reviewers, we continue to maintain systematic investigations for scenarios involving different forms of pairwise encounters in the case of S_C_ = 2 and S_R_ = 1 before applying our model to the experimental data.

**Reviewer #2 (Recommendations For The Authors):**
(1) I believe the surfaces in Figs. 1F-H corresponds to the zero-growth isoclines. The authors should directly point it out in the figure captions and text descriptions.

We thank the reviewer for this suggestion, and we have followed it to address the issue.

(2) After showing equations 1 or 2, I believe it will help readers understand the mechanism of equations by adding text such as "(see Fig. 1B)" to the sentences following the equations.

We appreciate the reviewer's suggestion, and we have implemented it to address the issue.

(3) Lines 12, 129 143 & 188: "at steady state" -> "at a steady state"(4) Line 138: "is doom to extinct" -> "is doomed to extinct"(5) Line 170: "intraspecific interference promotes species coexistence along with stochasticity" -> "intraspecific interference still robustly promotes species coexistence when stochasticity is considered"(6) Line 190: "The long-term coexistence behavior are exemplified" -> "The long-term coexistence behavior is exemplified"(7) Line 227: "the coefficient of variation was taken round 0.3" -> "the coefficient of variation was taken around 0.3"?(8) Line 235: "tend to extinct" -> "tend to be extinct"

We thank the reviewer for all these suggestions, and we have implemented each of them to revise our manuscript.

**Reviewer #3 (Recommendations For The Authors):**
I think this would be a much more useful paper if the authors focused on how the behavior of this model differs from existing models rather than showing that the new formation also generates the same dynamics as the existing theory.

We thank the reviewers for this suggestion, and we apologize for not explaining the limitations of the B-D model and the related studies on the topic of CEP clearly enough at the time of our initial submission. As we have explained in the responses above, we have now revised the introduction part of our manuscript (lines 5167 in the main text) to make it clear that since the functional response in the B-D model can be derived from the scenario involving only chasing pairs without consideration of pairwise encounters between consumer individuals, while we have demonstrated that a scenario involving only chasing pairs is under the constraint of CEP, it is thus highly questionable regarding the validity of the studies relying on the B-D model to break CEP or explain the paradox of the plankton. Consequently, one of the major objectives of our manuscript is to resolve whether the mechanism of intraspecific interference can truly break CEP and explain the paradox of the plankton in a rigorous manner. By modeling from a mechanistic perspective, we resolve the above issues and quantitatively illustrate a broad spectrum of experimental results, including two classical experiments that violate CEP and the rank-abundance curves across diverse ecological communities.

Things that would be of interest:What are the conditions for coexistence in this model? Presumably, it depends heavily on the equilibrium abundances of the consumers and resources as well as the engagement times/rates.

We thank the reviewer for raising this question. We have shown that there is a wide range of parameter space for species coexistence in our model. Specifically, for the case involving two consumer species and one resource species (S_C_ = 2 and S_R_ = 1), we have conducted a systematic study on the parameter region for promoting species coexistence. For clarity, we set the mortality rate 𝐷_i_ (*i* = 1, 2) as the only parameter that varies with the consumer species, and the order of magnitude of all model parameters was estimated from behavioral data. The results for scenarios involving intraspecific predator interference are shown in Appendix-figs. 4B-D, 5A, 6C-D and we redraw some of them here as Fig. R2, including both ODEs and SSA results, wherein Δ = (𝐷_1_-𝐷_2_)/ 𝐷_2_ represents the competitive difference between the two consumer species. For example, Δ = 1 means that species C2 is twice the competitiveness of species C_1_. In Fig. R2 (see also Appendix-figs. 4B-D, 5A, 6C-D), we see that the two consumer species can coexist with a large competitive difference in either ODEs and SSA simulation studies.

**Author response image 2. sa3fig2:** The parameter region for two consumer species coexisting with one type of abiotic resource species (S_C_ = 2 and S_R_ = 1). (A) The region below the blue surface and above the red surface represents stable coexistence of the three species at constant population densities. (B) The blue region represents stable coexistence at a steady state for the three species. (C) The color indicates (refer to the color bar) the coexisting fraction for long-term coexistence of the three species. Figure redrawn from Appendixfigs. 4B, 6C-D.

For systems shown in Fig. 3A-D, where the number of consumer species is much larger than that of the resource species, we set each consumer species with unique competitiveness through a distinctive 𝐷_i_ (*i* = 1,…, S_C_). In Fig. 3A-D (see also Appendix fig. 10), we see that hundreds of consumer species may coexist with one or three types of resources when the coefficient of variation (CV) of the consumer species’ competitiveness was taken around 0.3, which indicates a large parameter region for promoting species coexistence.

Is there existing data to estimate the parameters in the model directly from behavioral data? Do these parameter ranges support the hypothesis that predator interference is significant enough to allow for the coexistence of natural predator populations?

We appreciate the reviewer for raising this question. Indeed, the parameters in our model were primarily determined by estimating their reasonable range from behavioral data. Following the reviewer's suggestions, we have now specified the data we used to set the parameters. For instance, in Fig. 2D, we set 𝐷_2_=0.01 with τ=0.4 Day, resulting in an expected lifespan of *Drosophila* serrata in our model setting of 𝜏⁄𝐷_2_ = 40 days, which roughly agrees with experimental behavioral data showing that the average lifespan of D. serrata is 34 days for males and 54 days for females (lines 321325 in the appendices; reference: Narayan et al. J Evol Biol. 35: 657–663 (2022)). To account for competitive differences, we set the mortality rate as the only parameter that varies among the consumer species. As specified in the Appendices, the CV of the mortality rate is the only parameter that was used to fit the experiments within the range of 0.15-0.43. This parameter range (i.e., 0.15-0.43) was directly estimated from experimental data in the reference article (Patricia Menon et al., Water Research 37, 4151(2003)) using the two-sigma rule (lines 344-347 in the appendices).

Given the high consistency between the model results and experiments shown in Figs. 2D-E and 3C-D, where all the key model parameters were estimated from experimental data in references, and considering that the rank-abundance curves shown in Fig. 3C-D include a wide range of ecological communities, there is no doubt that predator interference is significant enough to allow for the coexistence of natural predator populations within the parameter ranges estimated from experimental references.

Bifurcation analyses for the novel parameters of this model. Does the fact that prey can escape lead to qualitatively different model behaviors?

**Author response image 3. sa3fig3:** Bifurcation analyses for the separate rate d’_i_ and escape rate d_i_ (*i* = 1, 2) of our model in the case of two consumer species competing for one abiotic resource species (S_C_ = 2 and S_R_ = 1). (A) A 3D representation: the region above the blue surface signifies competitive exclusion where C_1_ species extinct, while the region below the blue surface and above the red surface represents stable coexistence of the three species at constant population densities. (B) A 2D representation: the blue region represents stable coexistence at a steady state for the three species. Figure redrawn from Appendix-fig. 4C-D.

We appreciate the reviewer for this suggestion. Following this suggestion, we have conducted bifurcation analyses for the separate rate d’_i_ and escape rate d_i_ of our model in the case where two consumer species compete for one resource species (S_C_ = 2 and S_R_ = 1). Both 2D and 3D representations of these results have been included in Appendix-fig. 4, and we redraw them here as Fig. R3. In Fig. R3, we set the mortality rate *Di (i = 1, 2)* as the only parameter that varies between the consumer species, and thus *Δ = (D1-D2)/D2* represents the competitive difference between the two species*.*

As shown in Fig. R3A-B, the smaller the escape rate d_i_, the larger the competitive difference Δ tolerated for species coexistence at steady state. A similar trend is observed for the separate rate d’_i_. However, there is an abrupt change for both 2D and 3D representations at the area where d’_i_ = 0, since if d’_i_ = 0, all consumer individuals would be trapped in interference pairs, and then no consumer species could exist. On the contrary, there is no abrupt change for both 2D and 3D representations at the area where d_i_=0, since even if d_i_=0, the consumer individuals could still leave the chasing pair through the capture process.

Figures: I found the 3D plots especially Appendix Figure 2 very difficult to interpret. I think 2D plots with multiple lines to represent predator densities would be more clear.

We thank the reviewer for this suggestion. Following this suggestion, we have added a 2D diagram to Appendix-fig. 2.